# Navigating the MIL Trade-Off: Flexible Pooling for Whole Slide Image Classification

**Hossein Jafarinia**[1], **Danial Hamdi**[1,*] **Amirhossein Alamdar**[1,*] **Elahe Zahiri**[2]
**Soroush Vafaie Tabar**[1], **Alireza Alipanah**[1], **Nahal Mirzaie**[1], **Saeed Razavi**[1]
**Amir Najafi**[1], **Mohammad Hossein Rohban**[1]

[1]Computer Engineering Department, Sharif University of Technology
[2]Department of Mathematical Sciences, Sharif University of Technology

`{jafarinia, amirhossein.alamdar, elahe.zahiri`
`soroush.vafaie96, alireza.alipanah46, nahal.mirzaie`
`saeed.razavi, amir.najafi, rohban}@sharif.edu`
`danial.hamdi@outlook.com`

## Abstract

Multiple Instance Learning (MIL) is a standard weakly supervised approach for Whole Slide Image (WSI) classification, where performance hinges on both feature representation and MIL pooling strategies. Recent research has predominantly focused on Transformer-based architectures adapted for WSIs. However, we argue that this trend faces a fundamental limitation: data scarcity. In typical settings, Transformer models yield only marginal gains without access to large-scale datasets—resources that are virtually inaccessible to all but a few well-funded research labs. Motivated by this, we revisit simple, non-attention MIL with unsupervised slide features and analyze temperature-$\beta$-controlled log-sum-exp (LSE) pooling. For slides partitioned into $N$ patches, we theoretically show that LSE has a smooth transition at a critical $\beta_{\mathrm{crit}} = \mathcal{O}(\log N)$ threshold, interpolating between mean-like aggregation (stable, better generalization but less sensitive) and max-like aggregation (more sensitive but looser generalization bounds). Grounded in this analysis, we introduce Maxsoft—a novel MIL pooling function that enables flexible control over this trade-off, allowing adaptation to specific tasks and datasets. To further tackle real-world deployment challenges such as specimen heterogeneity, we propose PerPatch augmentation—a simple yet effective technique that enhances model robustness. Empirically, Maxsoft achieves state-of-the-art performance in low-data regimes across four major benchmarks (CAMELYON16, CAMELYON17, TCGA-Lung, and SICAP-MIL), often matching or surpassing large-scale foundation models. When combined with PerPatch augmentation, this performance is further improved through increased robustness. Code is available at `https://github.com/jafarinia/maxsoft`.

## 1 Introduction

Whole Slide Image (WSI) analysis using machine learning holds significant promise for supporting the complex and labor-intensive workflow of pathologists [1–3]. However, the extremely large and variable size of WSIs—typically on the order of $150{,}000 \times 150{,}000$ pixels—renders the direct application of standard computer vision models, such as Vision Transformers (ViTs) [4], infeasible. The prevailing solution is to divide WSIs into smaller patches and apply a weakly supervised framework known as Multiple Instance Learning (MIL). MIL enables joint modeling of slide-level and patch-level predictions by decomposing the task into a patch-level representation encoder followed by a pooling function that aggregates patch features for WSI-level classification [5, 6].

---

*Equal contribution.

Following the success of the attention mechanism [7] and the introduction of the Transformer architecture [8], attention-based methods such as ABMIL were adopted in the context of pathology MIL, leading to modest performance gains [5]. In parallel, the introduction of self-supervised learning (SSL)—beginning with SimCLR [9] and its application in DSMIL [10]—enabled training of encoders from scratch on domain-specific patches, resulting in substantial improvements. Since then, much of the research has centered on pairing powerful SSL-trained encoders with increasingly complex Transformer-based architectures to push state-of-the-art (SoTA). However, these works often lack thorough analysis of performance attribution and frequently credit improvements to architectural complexity without sufficient empirical justification [11–16].

In this paper, we demonstrate that Transformer-based MIL methods yield only marginal gains in low-data settings (Figure 3), and once a strong representation encoder is in place, Transformer architectures offer little to no added benefit (see Tables 1 and 3 when the Encoder is Prov-GigaPath [17]). Drawing on our comprehensive encoder experiments in Tables 1 and 3 and the data-quality analysis in Appendix N, we attribute this effect to their well-documented reliance on large-scale training data [4, 15, 18–31]. Unfortunately, most publicly available WSI datasets are relatively small [18, 19], comprising only a few hundred to a few tens of thousands of slides—insufficient for Transformer training (WSI-level, not patch-level). To our knowledge, the largest existing dataset contains approximately 200,000 WSIs [17], is not publicly accessible, and is still arguably too small for training Transformer models. Moreover, the computational resources required to process such datasets are concentrated in a few well-resourced laboratories, making this direction impractical for most groups working on WSI classification.

This motivates a reconsideration of classical (non-attention-based) MIL pooling strategies such as **max pooling** and **mean pooling**. In particular, we analyze **log-sum-exp (LSE) pooling**, a temperature–parameterized function:

$$\mathsf{LSE}_\beta(q_1, \ldots, q_N) \triangleq \frac{1}{\beta} \log \left( \frac{1}{N} \sum_{i=1}^{N} e^{\beta q_i} \right),$$

defined for real-valued inputs $q_1, \ldots, q_N$ with an adjustable parameter $\beta \geq 0$ (inverse temperature). The LSE function smoothly interpolates between mean and max behavior: when $\beta \ll 1$, it approximates the mean; when $\beta \gg 1$, it approaches the maximum. Through a combination of theoretical and empirical analyses, we observe smooth phase transitions in model behavior as $\beta$ crosses a critical threshold of order $\mathcal{O}(\log N)$, where $N$ denotes the number of patches. In the *small-$\beta$ regime* (i.e., $\beta \ll \mathcal{O}(\log N)$), the model exhibits improved generalization but reduced sensitivity (see Theorem 1). In contrast, in the *large-$\beta$ regime* (i.e., $\beta \gg \mathcal{O}(\log N)$), sensitivity increases at the cost of looser generalization bounds (see Theorem 2) and less stable training. This trade-off reveals a form of *Pareto optimality* in the temperature parameter, enabling task-specific tuning to balance generalization and sensitivity. Additionally, while max pooling aligns well with the MIL inductive bias, its non-differentiability and high gradient variance make it challenging to optimize, often resulting in unwanted test-time performance fluctuations (see Tables 1 and 3).

Motivated by these observations and insights from Backward Pass Differentiable Approximation (BPDA) [32], we propose **Maxsoft pooling**—a novel MIL pooling strategy that applies LSE with $\beta = \infty$ (i.e., max) during the forward pass, and the gradient of LSE with a moderate $\beta$ during the backward pass. Conceptually, this hard-forward/soft-backward design is closely related to straight-through estimators [33] (e.g., Straight-Through Gumbel-Softmax) [34]. This design improves optimization stability, generalization, and inference-time sensitivity. Across four major pathology datasets and five standard MIL benchmarks, Maxsoft consistently demonstrates strong performance. On the challenging CAMELYON17 dataset [35], we achieve a WSI-level AUC of $1.0$ and a patch-level AUC of $0.93$—to our knowledge, the best reported results to date in WSI classification.

From a different perspective, real-world deployment presents challenges such as staining variability, artifacts introduced during slide preparation, and acquisition noise. To enhance robustness under these conditions, we explored data augmentation strategies that reflect such clinically relevant variations. We found that previously proposed WSI-specific augmentations offer negligible or no benefit, as they fail to introduce meaningful or diverse transformations. To address this, we also introduce **PerPatch augmentation**—a simple yet previously unexplored technique that leverages the inherent patch-level granularity of WSIs to increase training diversity in MIL. We observed that combining PerPatch with existing methodologies further improves results in a measurable and consistent manner, and also reduces performance fluctuations.

In summary, our main contributions are:

- A theoretical and empirical analysis of LSE pooling, motivated by its ability to interpolate between max and mean behaviors through a tunable temperature parameter. In particular, we establish several smooth and competing phase transitions in model behavior, revealing trade-offs between generalization and sensitivity.

- The introduction of **Maxsoft**, a novel MIL pooling method derived from our analysis, which demonstrates strong performance, especially in low-data regimes.

- The proposal of **PerPatch augmentation**, a simple yet effective data augmentation strategy that leverages the patch-level structure of WSIs to substantially increase data diversity and yield measurable performance improvements.

## 2 Related Work

### 2.1 Transformer-based Architectures

Motivated by the success of the attention mechanism [7], the Transformer architecture [8], and biological insights suggesting that spatial relationships between regions in a WSI influence diagnostic outcomes, recent work has predominantly focused on Transformer-based MIL pooling architectures to capture inter-patch dependencies. WSIs typically consist of a large number of patches (e.g., around 20,000 per slide), making the naive application of self-attention—with its $\mathcal{O}(N^2)$ complexity—infeasible due to prohibitive GPU memory requirements. Among these approaches, ABMIL [5] was the first to introduce attention mechanisms in this context, while DSMIL [10] and Snuffy [15] proposed *sparse* self-attention variants. Notably, Snuffy provided theoretical justification that their sparse attention mechanism can approximate full self-attention under certain conditions [15].

Beyond Transformers, there are graph-based methods, which are hampered by graph construction and nucleus segmentation, and many graph–ViT hybrids collapse to self-attention—effectively Transformer MIL (e.g., GTP) [13, 36–39] and prototype-based approaches, which are also emerging and are sometimes not strictly MIL (e.g., PANTHER) [40, 41].

### 2.2 Classical MIL Pooling Functions

In addition to max and mean, several other pooling functions have been used, such as noisy-or [42] (any single cause), noisy-and [43] (sufficient proportion of causes), Integrated Segmentation and Recognition (ISR) [44] (a smooth OR with evidence accumulation), and smooth approximations to max/mean such as LSE [45], generalized mean (GM), and smoothmax [46].

### 2.3 Pathology Foundation Models

Recent efforts to develop Foundation Models for pathology [17, 47–49] have largely focused on training representation encoders using massive numbers of patches extracted from as many WSIs as possible. For instance, UNI [47] is trained on 100 million patches, while Prov-GigaPath [17] uses 1.3 billion patches from 200,000 WSIs—both based on DINOv2 [50]. These approaches yield highly robust encoders that can boost performance across various MIL pooling strategies. However, building such models faces key challenges: limited public data, high computational cost, lack of flexibility compared to generative foundation models like LLMs [51, 52], and sensitivity to training errors such as poor hyperparameter choices. Consequently, progress remains confined to a few well-resourced labs. Complementing these efforts, the recently proposed $R^2$T-MIL targets foundation model-level performance in low-data regimes via online feature re-embedding during MIL pooling training.

### 2.4 Augmentation Methods

**WSI-level Augmentation** modifies slide-level representations by selecting a fixed number of representative patches, inspired by Mixup [53] and Mask Augmentation [54]. Methods such as ReMix [55], RankMix [56], PseMix [19], MixupMIL [57], and Attention-Guided Mixup [58] apply various mixing strategies after aggressive patch downsampling (e.g., using centroids, ranked attention scores, or pseudo-bags). This leads to substantial information loss and degraded performance, often

requiring teacher-student distillation for limited gains (see Tables 2 and 4). Additionally, many such augmentations lack realism.

**Patch-level Augmentation** methods apply standard image transformations—such as random rotation, color jitter, and Hematoxylin-Eosin-DAB (HED) jitter—directly to patches prior to feature extraction. These augmentations are typically inspired by patch classification research [59, 60]. To increase efficiency, EMBAUGMENTER [61] and AugDiff [62] employ latent generative models to simulate augmented patch representations during MIL training. SSRDL [63] builds on this by introducing a DINO-based [64] framework that performs online sampling from a learned distribution of augmented features and reuses parts of the model during MIL pooling training. While these methods improve efficiency and robustness, they often suffer from limited diversity—e.g., applying the same augmentation uniformly across patches (EMBAUGMENTER, AugDiff)—and reduced interpretability, especially in SSRDL, where the specific form of augmentation is not directly observable.

## 3 Background: MIL Formulation

In a binary image classification setting, the dataset $D = \{(I_1, y_1), \ldots, (I_n, y_n)\}$ consists of images (or equivalently *bags of patches*) $I_i$, where each bag $I_i = \{\boldsymbol{X}_1^{(i)}, \ldots, \boldsymbol{X}_N^{(i)}\}$ contains $N$ corresponding instances. Each bag label $y_i \in \{0, 1\}$ is determined by its individual instance labels $\{y_1^{(i)}, \ldots, y_N^{(i)}\}$, where $y_j^{(i)} \in \{0, 1\}$ are unknown during training. Under the standard MIL assumption, we have

$$y_i = \left( \begin{cases} 0, & \text{if } \sum_{j=1}^N y_j^{(i)} = 0, \\ 1, & \text{otherwise,} \end{cases} \right) = \max_{j=1,\ldots,N} \{y_j^{(i)}\} \quad \forall i = 1, \ldots, n.$$

Thus, a bag $I_i$ is labeled positive if at least one of its instances is labeled positive; otherwise, it is labeled negative. The MIL model is trained by optimizing the log-likelihood function:

$$P(y|I) = \phi(I)^y (1 - \phi(I))^{1-y},$$

where $\phi(I) = \phi(\boldsymbol{X}_1, \ldots, \boldsymbol{X}_N) \in [0, 1]$ represents the predicted probability of $(y = 1)$ given the bag $I$. Since MIL assumes no ordering or dependency among instances, $\phi(I)$ must be a permutation-invariant function. This is ensured by the *fundamental theorem of symmetric functions with monomials* [65] and a similar result by [66], which states that for any permutation-invariant function $\phi$ satisfying Hausdorff continuity, there exist functions $\psi$ and $\Phi : \mathbb{R} \to \mathbb{R}$, and a permutation-invariant function $\pi : \mathbb{R}^N \to \mathbb{R}$ such that

$$\phi(I) = \Phi(\pi(\psi(\boldsymbol{X}_1), \ldots, \psi(\boldsymbol{X}_N))). \tag{1}$$

Here, $\psi$ and $\Phi$ are continuous transformations, and $\pi$ is a permutation-invariant function (such as sum or max). MIL methods typically follow two primary approaches: i) **Instance-level approach**, where $\psi$ is an instance classifier and $\Phi$ is the identity function. ii) **Embedding-level approach**, where $\psi$ is a feature extractor, and $\Phi$ maps the extracted features to a bag classification score.

In Deep MIL, $\psi$ typically uses pre-trained vision backbones to extract features from bag instances [10–12, 15, 18, 67, 68]. The aggregation function $\pi$ ranges from non-parametric methods like max pooling to parametric ones like attention mechanisms, as detailed in Section 2.1. For multi-class and multi-task classification, $\Phi$'s output dimension is adjusted to the number of classes [69–71].

## 4 Our Method

While SoTA MIL pooling architectures focus on embedding-level methods, this work emphasizes instance-level approaches. In clinical practice, pathologists prioritize patch-level predictions as they allow validation of the model's reasoning. As a result, instance-level methods offer full interpretability. Although max pooling and mean pooling can be applied at both embedding and instance levels, we focus exclusively on their instance-level variants throughout this work (i.e., $\Phi$ in (1) is identity).

We begin by following the standard MIL pooling procedure: each image in $D$ is divided into $N$ non-overlapping patches $\boldsymbol{X}_1, \ldots, \boldsymbol{X}_N$. We model the function $\psi(\cdot)$ in (1) by passing each patch through a fixed, pre-trained ViT, followed by a trainable fully-connected MLP, and concluding with a sigmoid function in order to compute $q_i \triangleq \psi(\boldsymbol{X}_i) \in [0, 1]$ for $i = 1, \ldots, N$ (see Section 5 for more details). The next subsection establishes a theoretical foundation for the trade-offs associated with the choice of aggregation function $\pi(\cdot)$.

## 4.1 Theoretical Analysis

Assume, instead of the standard max/mean pooling, we choose a soft $\mathsf{LSE}_\beta$ function for $\pi(\cdot)$ in (1). We theoretically demonstrate that smaller values of $\beta$ generally lead to better generalization and a more stable training process. Conversely, larger values of $\beta$ are more suitable for reducing training error and increasing the model's sensitivity in real-world scenarios, where only a few patches may contain malignant patterns, and the model must detect and respond to them effectively. A full version of this section is provided in Appendix A, with a summary of informal results here:

**Theorem 1** ((Informal) Effect of $\beta$ on Generalization Error). *Assuming mild conditions—such as Lipschitz continuity of the loss function, i.i.d. sampling of images in dataset $D$, and local statistical dependence among image patches—we show that the generalization error undergoes a smooth phase transition as $\beta$ increases from below $\mathcal{O}(\log N)$ to above. In the small-$\beta$ regime, the generalization gap can be bounded as $\mathcal{O}\left((nN)^{-1/2} + n^{-1}\right)$, while in the large-$\beta$ regime, it can be as large as $\Omega\left(n^{-1/2}\right)$.*

The formal version of this theorem (Theorem 4), including precise bounds, threshold conditions, and constants, is provided in Appendix A, along with a complete proof. Next, we focus on sensitivity analysis. Notably, in contrast to generalization—which favors smaller values of $\beta$—high sensitivity in noisy regimes requires sufficiently large $\beta$.

**Theorem 2** ((Informal) Effect of $\beta$ on True Positive / False Negative Rate). *Assume a general statistical model for the distribution of $\psi(\boldsymbol{X}_i)$s on each patch, and suppose the number of cancer-indicative patches (i.e., patches exhibiting malignant patterns) is, with high probability, substantially smaller than $N$. Then, for any $\alpha \in (0, 1)$, if $\beta \geq \mathcal{O}\left(\log\left(N/\alpha\right)\right)$, the true positive rate of classifier $\mathsf{LSE}_\beta(\psi(\boldsymbol{X}_1), \ldots, \psi(\boldsymbol{X}_N))$ is at least $1 - \alpha$. Conversely, if $\beta$ remains $\mathcal{O}(1)$ with respect to $N$, the false negative rate is at least $1/2$.*

Again, the formal version of this theorem (Theorem 6), along with a complete proof and further explanations, is provided in Appendix A. Specifically, we show that choosing $\beta \gg \mathcal{O}(\log N)$ is both necessary and sufficient to achieve a high true positive rate and to mitigate false negatives. The key conclusion of this section is that all values of $\beta \in (0, +\infty)$ are *Pareto-optimal*: depending on which aspect—generalization or sensitivity—is prioritized, and how the trade-off is weighted, the optimal choice of $\beta$ will vary. This insight supports the central idea behind our method: using smaller values of $\beta$ during training to promote generalization, while selecting larger values (possibly even $\infty$) at inference time to enhance sensitivity.

## 4.2 Maxsoft pooling

High gradient variance can lead to instability during training, slowing convergence and requiring more iterations to reach a desired accuracy [72–74]. Thus, reducing variance improves stability, enhances generalization, and promotes more reliable learning dynamics [75–77]. This underscores the need to regulate gradient variance to balance stability with effective optimization. max pooling and mean pooling, the two dominant MIL pooling strategies, represent opposite ends of this spectrum: max pooling excels in instance discrimination but suffers from instability and worse generalization due to its non-smooth, non-differentiable nature, while mean pooling is stable, fully differentiable, and robust for bag classification but weaker at distinguishing instances. We provide a unified theoretical explanation for these trade-offs in Section 4.1.

Leveraging this insight, we introduce Maxsoft pooling (see Figure 1), a novel strategy that combines the stability and smoothness of mean pooling with the discriminative power of max pooling. In the forward pass, we use $\mathsf{LSE}_\infty(q_{1:N}) = \max_i q_i$, while in the backward pass we approximate the gradient of the max operator by $\nabla\mathsf{LSE}_\beta(q_{1:N})$ for some moderate $\beta < \infty$. For more details on the learning procedure, see Algorithm 1. Maxsoft pooling retains the exact behavior of max pooling during the forward pass, but it employs a differentiable function in the backward pass by replacing the max operator with the LSE. This ensures smoother gradient computation, improved stability, and reduced generalization gap (see [32], and Theorem 1).

## 4.3 PerPatch Augmentation

Existing WSI-level augmentation methods, as outlined in Section 2.4, often yield marginal gains or even degrade performance [19, 55, 62, 63]. Therefore, we shift focus to patch-level augmentation,

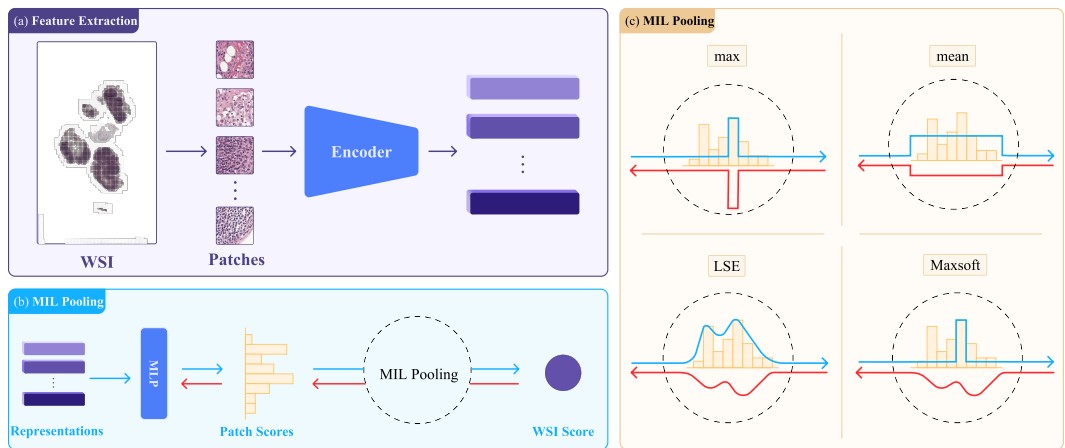

Figure 1: Overview of MIL (a) The WSIs are segmented into patches, followed by embedding extraction via a pre-trained encoder. (b) Embeddings are fed into MIL pooling for patch. (c) mean, max, LSE, and Maxsoft with their respective forward and backward behaviors.

which introduces more meaningful variability. Latent generative models [61, 62] add computational overhead without consistent benefits (Tables 2, 4). Instead, we precompute multiple augmented versions of each patch and employ the PerPatch sampling strategy, enabling a substantial increase in WSI diversity during MIL pooling training.

**PerPatch** augmentation differs from PerSlide by sampling each patch independently—either from its original version or one of its augmented variants—instead of applying the same augmentation transformation across all patches in a single WSI (see Figure 5). Specifically, given a WSI $I$, we generate multiple augmented variants $I^{(k)} = \{\boldsymbol{X}_1^{(k)}, ..., \boldsymbol{X}_N^{(k)}\}$, $k = 0, 1, \ldots, m$, where $I^{(0)} = I$ is the original WSI and $m$ denotes the number of augmentations. We then construct an augmented WSI $\widehat{I}$ by independently sampling each patch from its available versions: $\widehat{I} = \{\widehat{\boldsymbol{X}}_1, \ldots, \widehat{\boldsymbol{X}}_N\}$ with $\widehat{\boldsymbol{X}}_j = \boldsymbol{X}_j^{(k)}$, $k \sim \text{Uniform}(\{0, 1, 2, ..., m\})$. Thus, each patch $\widehat{\boldsymbol{X}}_j$ is independently sampled from either the original version ($k = 0$) or one of its $m$ augmented variants ($1 \leq k \leq m$). For a visual and procedural comparison between PerSlide and PerPatch augmentations, refer to Figure 5 and Algorithms 2 and 3. We argue that PerPatch creates significantly more diverse augmentations during MIL pooling training compared to PerSlide, and this may explain why the method consistently improves results, as shown in Tables 2 and 4.

## 5  Experiments and Results

Our major experiments are on major WSI classification datasets. Analyses of classical MIL pooling functions are in Appendix O. Although out-of-distribution (OOD) generalization is beyond our scope, we include such experiments in Appendix Q. We also evaluate on five canonical MIL datasets, reported in Appendix R.

### 5.1  Datasets

We evaluate on four large-scale pathology WSI datasets—CAMELYON16, CAMELYON17, TCGA-Lung and SICAP-MIL [78, 35, 79, 80]. CAMELYON16 contains 270 training / 129 test slides; CAMELYON17 has 1,000 slides across four tumor-type labels (normal, macro, micro, ITC); TCGA-Lung spans 1,042 slides split between LUAD and LUSC; SICAP-MIL comprises 349 slides with normal/abnormal labels and primary/secondary Gleason grades. Further dataset statistics and split protocols are given in Appendix E.

### 5.2  Experimental Setup

All WSIs are tiled into $256 \times 256$ patches at $20\times$ magnification, discarding background. For CAME-LYON16 we use the official split; CAMELYON17, TCGA-Lung and SICAP-MIL are each partitioned

into 60% train, 15% val, 25% test (with SICAP-MIL's official split). Details on train/val/test allocations and tiling thresholds appear in Appendix F.

## 5.3 Evaluation Metrics

We report slide-level AUC and accuracy, patch-level AUC and F1 (omitting patch accuracy due to class imbalance), plus F1 and Expected Calibration Error (ECE) to assess confidence calibration in cancer detection. Formal definitions and details for ECE are in Appendix G.

## 5.4 Models and Implementation Details

**Encoders:** We compare domain-specific and natural-image backbones including DINO Domain (ViT-S/16) trained on around a few million in-domain patches, DINO Natural (ViT-S/16) [64] pre-trained on ImageNet-1K, UNI (ViT-L/16) [47] with 100 million pathology patches, Prov-GigaPath (ViT-G/14) [17] with 1.3 billion patches, ResNet-50 [81] pre-trained on ImageNet-1K, PLIP [82], a CLIP-based model fine-tuned on around 208 thousand text-annotated pathology patches, and SSRDL-trained ViTs [63] via its Representation Augmentation Module on domain patches.

**MIL-Pooling:** We evaluate max, mean, LSE, ABMIL [5], DSMIL [10], Snuffy [15], $R^2$T-MIL [16] and our Maxsoft.

**Miscellaneous:** For completeness, we also evaluate GTP [37] and PANTHER [41] as non-MIL WSI classifiers.

**Augmentations:** We compare standard WSI-level baselines—with basic augmentations and simple per-slide transforms (Random Rotation, Gaussian Blur, and Color Jitter; see Appendix I)—and prior augmentation strategies (ReMix [55], RankMix [56], AugDiff [62], and SSRDL's latent augmentations [63]) against PerPatch, our patch-level method. Hyperparameters, optimizer settings, learning rates, and batch sizes are provided in Appendix H.

## 5.5 Results and Analysis

Across all datasets, stronger encoders yield better performance: DINO Natural < DINO Domain < UNI < Prov-GigaPath. With Prov-GigaPath, every MIL pooling strategy attains high scores (Tables 1 and 3). For visualizations of the resulting representations, see Appendix K.

Max pooling shows high variability, whereas mean pooling offers lower variance and on average better generalization. Guided by our theoretical analysis, we select $\beta$ via validation: $\beta = 5$ for CAMELYON16/17, $\beta = 10$ for TCGA-Lung, and $\beta = 3.5$ for SICAP-MIL. This choice reflects the datasets' characteristics—sparser positives in CAMELYON demand higher sensitivity, while more frequent positives in TCGA-Lung and SICAP-MIL favor generalization.

LSE matches or outperforms both Transformer-based and non-Transformer-based methods (ABMIL, DSMIL, Snuffy, $R^2$T-MIL, GTP, and PANTHER) despite its simplicity. Maxsoft attains the highest overall performance and substantially lower variance than max pooling, validating the use of $\beta = \infty$ in the forward pass with a moderate $\beta$ for gradients. These results indicate that, at current WSI dataset sizes, encoder quality dominates architectural complexity and attention-based pooling yields limited gains.

On augmentation (Tables 2 and 4), PerSlide yields inconsistent or negative effects, whereas PerPatch uniformly reduces variance and often improves accuracy, indicating robustness. WSI-level methods like ReMix and RankMix underperform due to aggressive patch downsampling and because DINO [64] embeddings lack augmentation–interpolation consistency between the image and feature spaces. Among prior approaches, only AugDiff and, at times, SSRDL show improvements—likely because, despite the issues above, they still produce diverse, non-redundant feature augmentations. SSRDL [63] offers some gains on TCGA-Lung but at high cost, since each new augmentation requires retraining a self-supervised model from scratch. Overall, PerPatch delivers the largest gains through richer, more varied augmentations. For analysis of special cases, see Appendix L.2.

Notably, Maxsoft with DINO Domain surpasses Prov-GigaPath in accuracy while requiring $0.001\times$ less computation and having $0.0026\times$ fewer parameters. In terms of augmentation speed, AugDiff is

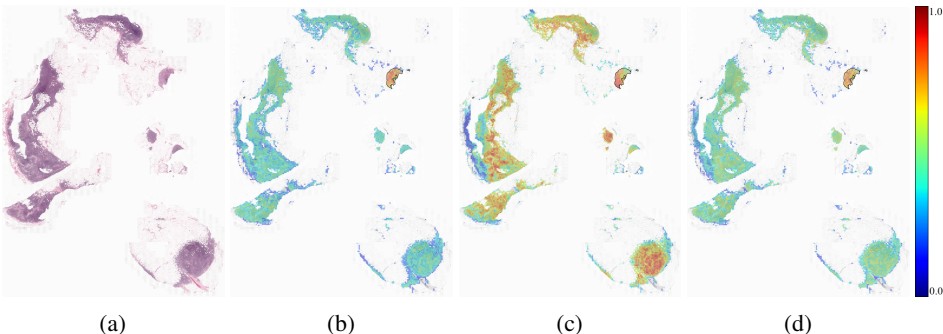

Figure 2: Overview of ROIs through patch classification on (a) a sample WSI from CAMELYON17 by (b) max pooling, (c) mean pooling, and (d) Maxsoft pooling. The cancerous area is specified with a black border. Mean pooling is classifying most of the WSI as cancerous incorrectly.

prohibitively slow due to its multi-step diffusion, a limitation not addressed in the original work. For a detailed analysis of special cases, see Appendix L.1.

Finally, for patch-level classification and localization, simple instance-level pooling methods outperform SoTA embedding-level architectures—except Snuffy—underscoring the advantage of direct patch-wise aggregation. For qualitative comparison, see Figures 2 and 8.

Table 1: Results of MIL pooling methods on CAMELYON16 and CAMELYON17 [78, 35], reported as mean$_{std}$ for performance metrics. Times are given as pre-training (days) / MIL training (minutes).

| Encoder | Method | CAMELYON16 | | | | | | | | CAMELYON17 | | | | | | | |
|---|---|---|---|---|---|---|---|---|---|---|---|---|---|---|---|---|---|
| | | Slide | | | | Patch | | Resource | | Slide | | | | Patch | | Resource | |
| | | AUC | ACC | F1 | ECE | AUC | F1 | Time | #Params | AUC | ACC | F1 | ECE | AUC | F1 | Time | #Params |
| DINO Natural | max pooling | $0.588_{029}$ | $0.620_{015}$ | $0.483_{077}$ | $0.287_{010}$ | $0.586_{216}$ | $0.001_{002}$ | 16.66/1.56 | 385 | $0.513_{216}$ | $0.433_{189}$ | $0.514_{288}$ | $0.244_{129}$ | $0.527_{350}$ | $0.206_{357}$ | 16.66/2.35 | 385 |
| | mean pooling | $0.569_{015}$ | $0.585_{008}$ | $0.513_{039}$ | $\mathbf{0.057}_{014}$ | $0.688_{016}$ | $0.175_{001}$ | 16.66/1.452 | 385 | $0.510_{000}$ | $0.450_{000}$ | $0.642_{000}$ | $\mathbf{0.157}_{000}$ | $0.513_{001}$ | $0.027_{000}$ | 16.66/2.179 | 385 |
| | ABMIL | $0.710_{129}$ | $0.716_{137}$ | $0.649_{085}$ | $0.251_{106}$ | $0.697_{151}$ | $0.224_{175}$ | 16.66/121.06 | 99074 | $0.673_{111}$ | $0.600_{100}$ | $0.622_{143}$ | $0.348_{086}$ | $\mathbf{0.822}_{040}$ | $\mathbf{0.347}_{125}$ | 16.66/181.596 | 99074 |
| | DSMIL | $0.594_{024}$ | $0.646_{004}$ | $0.531_{027}$ | $0.187_{021}$ | $0.440_{065}$ | $0.096_{026}$ | 16.66/124.3 | 66562 | $0.580_{026}$ | $0.517_{029}$ | $0.546_{073}$ | $0.455_{047}$ | $0.767_{127}$ | $0.184_{153}$ | 16.66/186.45 | 66562 |
| | Snuffy | $0.550_{072}$ | $0.550_{022}$ | $0.420_{035}$ | $0.196_{004}$ | $0.534_{456}$ | $0.097_{138}$ | 16.66/3.1 | 1776002 | $0.540_{014}$ | $0.550_{071}$ | $0.492_{090}$ | $0.246_{045}$ | $0.273_{090}$ | $0.000_{000}$ | 16.66/4.654 | 1776002 |
| | GTP | $0.690_{046}$ | $0.620_{021}$ | $0.576_{061}$ | $0.342_{019}$ | / | / | 16.66/80.0 | 131365 | $\mathbf{0.773}_{067}$ | $0.583_{104}$ | $0.799_{033}$ | $0.409_{090}$ | / | / | 16.66/85.4 | 131365 |
| | LSE pooling | $0.710_{080}$ | $0.733_{104}$ | $0.627_{123}$ | $0.202_{068}$ | $0.835_{000}$ | $0.210_{148}$ | 16.66/1.691 | 74113 | $0.683_{093}$ | $0.600_{087}$ | $0.654_{070}$ | $0.361_{075}$ | $0.168_{291}$ | $0.040_{069}$ | 16.66/2.537 | 74113 |
| | Maxsoft pooling | $\mathbf{0.754}_{017}$ | $\mathbf{0.822}_{000}$ | $\mathbf{0.715}_{006}$ | $0.167_{004}$ | $\mathbf{0.839}_{028}$ | $\mathbf{0.265}_{026}$ | 16.66/3.27 | 74113 | $0.710_{010}$ | $\mathbf{0.650}_{050}$ | $0.658_{083}$ | $0.345_{040}$ | $0.312_{024}$ | $0.000_{000}$ | 16.66/4.908 | 74113 |
| DINO Domain | max pooling | $0.873_{178}$ | $0.844_{173}$ | $0.864_{186}$ | $0.089_{102}$ | $0.888_{131}$ | $0.394_{220}$ | 1.10/1.508 | 385 | $0.767_{361}$ | $0.800_{260}$ | $0.862_{149}$ | $0.242_{093}$ | $0.838_{106}$ | $0.426_{369}$ | 1.57/2.262 | 385 |
| | mean pooling | $0.511_{012}$ | $0.513_{011}$ | $0.248_{064}$ | $0.189_{019}$ | $0.754_{033}$ | $0.193_{007}$ | 1.10/1.39 | 385 | $0.897_{006}$ | $0.700_{000}$ | $0.892_{030}$ | $0.237_{011}$ | $0.695_{002}$ | $0.041_{000}$ | 1.57/2.09 | 385 |
| | ABMIL | $0.819_{059}$ | $0.762_{060}$ | $0.715_{064}$ | $0.215_{064}$ | $0.817_{046}$ | $0.302_{033}$ | 1.10/136.16 | 99074 | $0.750_{151}$ | $0.683_{116}$ | $0.737_{041}$ | $0.308_{117}$ | $0.689_{215}$ | $0.133_{177}$ | 1.57/204.24 | 99074 |
| | DSMIL | $0.910_{071}$ | $0.780_{078}$ | $0.824_{064}$ | $0.114_{043}$ | $0.278_{185}$ | $0.058_{031}$ | 1.10/140.09 | 66562 | $0.833_{108}$ | $0.800_{132}$ | $0.838_{088}$ | $0.204_{124}$ | $0.834_{084}$ | $0.160_{085}$ | 1.57/210.14 | 66562 |
| | Snuffy | $0.754_{001}$ | $0.589_{022}$ | $0.636_{080}$ | $0.195_{028}$ | $0.334_{006}$ | $0.142_{006}$ | 1.10/4.55 | 1776002 | $0.775_{191}$ | $0.650_{000}$ | $0.839_{086}$ | $0.260_{083}$ | $0.765_{115}$ | $0.013_{018}$ | 1.57/6.832 | 1776002 |
| | GTP | $0.599_{177}$ | $0.615_{135}$ | $0.559_{162}$ | $0.358_{114}$ | / | / | 1.57/80.0 | 131365 | $0.627_{049}$ | $0.583_{076}$ | $0.679_{115}$ | $0.385_{082}$ | / | / | 1.57/85.4 | 131365 |
| | LSE pooling | $0.919_{019}$ | $0.853_{012}$ | $0.848_{009}$ | $0.093_{002}$ | $\mathbf{0.926}_{005}$ | $\mathbf{0.455}_{023}$ | 1.10d/1.629 | 74113 | $0.850_{070}$ | $0.800_{050}$ | $0.833_{021}$ | $0.185_{066}$ | $0.836_{026}$ | $\mathbf{0.499}_{113}$ | 1.57/2.444 | 74113 |
| | Maxsoft pooling | $\mathbf{0.934}_{012}$ | $\mathbf{0.863}_{004}$ | $\mathbf{0.876}_{039}$ | $\mathbf{0.088}_{021}$ | $0.919_{011}$ | $0.321_{004}$ | 1.10/3.368 | 74113 | $\mathbf{0.983}_{055}$ | $\mathbf{0.867}_{076}$ | $\mathbf{0.935}_{072}$ | $0.121_{051}$ | $\mathbf{0.839}_{019}$ | $0.386_{237}$ | 1.57/5.052 | 74113 |
| UNI | max pooling | $0.903_{197}$ | $0.890_{173}$ | $0.898_{183}$ | $0.090_{146}$ | $0.853_{266}$ | $0.376_{212}$ | −/1.81 | 1025 | $0.633_{233}$ | $0.583_{153}$ | $0.609_{263}$ | $0.372_{154}$ | $0.761_{173}$ | $0.053_{091}$ | −/2.72 | 1025 |
| | mean pooling | $0.557_{053}$ | $0.564_{010}$ | $0.429_{073}$ | $0.367_{012}$ | $0.702_{071}$ | $0.181_{023}$ | −/1.68 | 1025 | $0.733_{071}$ | $0.730_{000}$ | $0.766_{036}$ | $0.262_{061}$ | $0.656_{134}$ | $0.036_{010}$ | −/2.52 | 1025 |
| | ABMIL | $0.935_{072}$ | $0.886_{043}$ | $0.884_{088}$ | $0.102_{035}$ | $0.959_{011}$ | $0.548_{264}$ | −/125.2 | 263554 | $0.707_{058}$ | $0.633_{116}$ | $0.777_{025}$ | $0.341_{079}$ | $0.475_{058}$ | $0.005_{007}$ | −/187.81 | 263554 |
| | DSMIL | $0.910_{016}$ | $0.822_{056}$ | $0.802_{043}$ | $0.060_{046}$ | $0.500_{000}$ | $0.125_{000}$ | −/128.42 | 149762 | $0.597_{249}$ | $0.633_{202}$ | $0.624_{240}$ | $0.353_{183}$ | $0.525_{362}$ | $0.220_{371}$ | −/192.63 | 149762 |
| | Snuffy | $0.787_{230}$ | $0.779_{137}$ | $0.682_{286}$ | $0.148_{094}$ | $0.911_{074}$ | $0.274_{386}$ | −/5.98 | 12600322 | $0.685_{035}$ | $0.675_{035}$ | $0.718_{026}$ | $\mathbf{0.192}_{037}$ | $0.664_{051}$ | $0.000_{000}$ | −/8.98 | 12600322 |
| | GTP | $0.644_{250}$ | $0.721_{174}$ | $0.300_{520}$ | $0.056_{018}$ | / | / | −/90.2 | 172325 | $\mathbf{0.763}_{021}$ | $0.733_{058}$ | $0.746_{038}$ | $0.252_{058}$ | / | / | −/92.4 | 172325 |
| | LSE pooling | $0.947_{077}$ | $0.933_{063}$ | $0.935_{087}$ | $0.069_{064}$ | $0.970_{003}$ | $0.604_{139}$ | −/1.96 | 525313 | $0.603_{351}$ | $0.617_{247}$ | $0.628_{400}$ | $0.386_{230}$ | $0.709_{213}$ | $0.210_{364}$ | −/2.942 | 525313 |
| | Maxsoft pooling | $\mathbf{0.992}_{003}$ | $\mathbf{0.966}_{025}$ | $\mathbf{0.980}_{000}$ | $\mathbf{0.028}_{018}$ | $\mathbf{0.974}_{004}$ | $\mathbf{0.800}_{106}$ | −/3.384 | 525313 | $0.753_{071}$ | $\mathbf{0.750}_{205}$ | $\mathbf{0.779}_{040}$ | $0.238_{172}$ | $\mathbf{0.786}_{255}$ | $\mathbf{0.476}_{429}$ | −/5.076 | 525313 |
| Prov-GigaPath | max pooling | $0.930_{043}$ | $0.876_{067}$ | $0.860_{085}$ | $0.098_{062}$ | $0.935_{014}$ | $0.416_{125}$ | 162.56/1.87 | 1537 | $0.860_{052}$ | $0.750_{132}$ | $0.836_{075}$ | $0.185_{072}$ | $0.920_{030}$ | $0.430_{318}$ | 162.56/2.814 | 1537 |
| | mean pooling | $0.542_{022}$ | $0.526_{017}$ | $0.399_{143}$ | $0.407_{012}$ | $0.860_{009}$ | $0.244_{002}$ | 162.56/1.74 | 1537 | $0.880_{087}$ | $0.767_{104}$ | $0.833_{030}$ | $0.181_{058}$ | $0.678_{155}$ | $0.037_{011}$ | 162.56/2.6 | 1537 |
| | ABMIL | $0.952_{034}$ | $0.894_{076}$ | $0.890_{054}$ | $0.103_{075}$ | $0.939_{013}$ | $\mathbf{0.632}_{139}$ | 162.56/161.5 | 395138 | $0.877_{035}$ | $0.850_{087}$ | $0.862_{101}$ | $0.149_{087}$ | $0.940_{020}$ | $0.490_{370}$ | 162.56/242.35 | 395138 |
| | DSMIL | $0.940_{049}$ | $0.845_{075}$ | $0.886_{064}$ | $0.043_{035}$ | $0.500_{000}$ | $0.125_{000}$ | 162.56/249.59 | 216322 | $0.917_{029}$ | $0.928_{034}$ | $0.090_{034}$ | | $0.928_{025}$ | $0.738_{108}$ | 162.56/249.59 | 216322 |
| | Snuffy | $0.896_{050}$ | $0.806_{066}$ | $0.798_{068}$ | $0.073_{012}$ | $0.962_{013}$ | $0.490_{193}$ | 162.56/10.21 | 28337666 | $0.885_{035}$ | $0.750_{071}$ | $0.856_{019}$ | $0.158_{039}$ | $0.940_{002}$ | $0.694_{135}$ | 162.56/15.326 | 28337666 |
| | GTP | $0.787_{092}$ | $0.721_{064}$ | $0.680_{075}$ | $0.253_{061}$ | / | / | $NA/NA$ | $NA$ | $0.442_{170}$ | $0.500_{000}$ | $0.232_{402}$ | $0.000_{000}$ | / | / | $NA/NA$ | $NA$ |
| | LSE pooling | $0.948_{024}$ | $0.956_{022}$ | $0.944_{024}$ | $0.046_{021}$ | $0.797_{108}$ | $0.540_{093}$ | 162.56/2.02 | 1181185 | $0.933_{031}$ | $0.900_{050}$ | $0.932_{027}$ | $0.095_{048}$ | $0.943_{004}$ | $0.741_{108}$ | 162.56/3.031 | 1181185 |
| | Maxsoft pooling | $\mathbf{0.985}_{004}$ | $\mathbf{0.966}_{004}$ | $\mathbf{0.976}_{006}$ | $\mathbf{0.035}_{002}$ | $\mathbf{0.962}_{002}$ | $0.437_{019}$ | 162.56/4.36 | 1181185 | $\mathbf{1.000}_{000}$ | $\mathbf{0.933}_{029}$ | $\mathbf{1.000}_{000}$ | $\mathbf{0.062}_{022}$ | $\mathbf{0.948}_{009}$ | $\mathbf{0.744}_{034}$ | 162.56/6.551 | 1181185 |
| IN | R²T-MIL | $0.607_{015}$ | $0.639_{093}$ | $0.509_{114}$ | $0.351_{088}$ | / | / | −/5.68 | 2696961 | $0.473_{035}$ | $0.467_{058}$ | $0.352_{295}$ | $0.166_{086}$ | / | / | −/8.52 | 2696961 |
| UNI | R²T-MIL | $0.886_{007}$ | $0.798_{000}$ | $0.791_{007}$ | $0.178_{002}$ | / | / | −/4.76 | 2434817 | $0.677_{067}$ | $0.650_{050}$ | $0.598_{152}$ | $0.352_{056}$ | / | / | −/7.14 | 2434817 |
| UNI | PANTHER | $0.832_{002}$ | $0.757_{013}$ | $0.735_{018}$ | $0.110_{009}$ | / | / | −/34.44 | 65570 | $0.583_{053}$ | $0.616_{023}$ | $0.590_{033}$ | $0.253_{038}$ | / | / | −/31.50 | 65570 |

## 6 Ablation Study

We first compare representative Transformer-based and classical methods across varying dataset sizes, then analyze Maxsoft by isolating the contribution of its key components to WSI- and patch-level performance.

**Dataset size.** Figure 3 reports results on SICAP-MIL for both Transformer-based and classical MIL pooling under different numbers of training WSIs. These findings indicate that, given current data scales, Transformer architectures offer no clear benefit. Notably, fine-tuning the Transformer-based Prov-GigaPath [17]—pretrained on roughly 200,000 WSIs—does not provide meaningful gains, highlighting the data inefficiency and limited practicality of complex, data-hungry approaches in this setting.

Table 2: Results of augmentation methods on CAMELYON16 and CAMELYON17 [78, 35], reported as mean$_{std}$ for performance metrics. Times are given as MIL training (hours).

| Method | Augmentation | CAMELYON16 | | | | | | | CAMELYON17 | | | | | | |
|---|---|---|---|---|---|---|---|---|---|---|---|---|---|---|---|
| | | Slide | | | | Patch | | Resource | Slide | | | | Patch | | Resource |
| | | AUC | ACC | F1 | ECE | AUC | F1 | Time | AUC | ACC | F1 | ECE | AUC | F1 | Time |
| Maxsoft pooling | No Aug | $0.934_{012}$ | $0.863_{004}$ | $0.876_{039}$ | $0.088_{021}$ | $0.919_{011}$ | $0.321_{004}$ | $0.06h$ | $\mathbf{0.983}_{055}$ | $0.867_{076}$ | $\mathbf{0.935}_{072}$ | $0.121_{051}$ | $0.839_{019}$ | $0.386_{237}$ | $0.08h$ |
| | ReMix | $0.883_{023}$ | $0.806_{008}$ | $0.807_{023}$ | $0.150_{012}$ | $0.875_{012}$ | $0.204_{007}$ | $0.52h$ | $0.840_{020}$ | $0.733_{029}$ | $0.798_{060}$ | $0.242_{017}$ | $0.808_{016}$ | $0.431_{016}$ | $0.78h$ |
| | RankMix | $0.913_{001}$ | $0.830_{000}$ | $0.826_{031}$ | $0.092_{060}$ | $0.894_{029}$ | $0.246_{032}$ | $0.24h$ | $0.855_{191}$ | $0.775_{247}$ | $0.849_{140}$ | $0.197_{156}$ | $0.881_{035}$ | $0.359_{366}$ | $0.32h$ |
| | AugDiff | $0.801_{078}$ | $0.820_{036}$ | $0.81_{071}$ | $0.182_{021}$ | $0.73_{035}$ | $0.211_{084}$ | $89h$ | $0.894_{069}$ | $0.840_{103}$ | $0.830_{030}$ | $0.161_{110}$ | $0.882_{012}$ | $0.441_{023}$ | $126h$ |
| | SSRDL | $0.810_{035}$ | $0.717_{076}$ | $0.775_{060}$ | $0.231_{031}$ | $0.689_{035}$ | $0.097_{052}$ | $4.5h$ | $0.859_{040}$ | $0.806_{013}$ | $0.790_{030}$ | $\mathbf{0.061}_{030}$ | $0.894_{008}$ | $0.462_{060}$ | $7h$ |
| | Base PerSlide | $0.865_{007}$ | $0.840_{012}$ | $0.797_{007}$ | $0.141_{011}$ | $0.869_{014}$ | $0.234_{032}$ | $0.9h$ | $0.907_{021}$ | $0.800_{087}$ | $0.846_{084}$ | $0.176_{060}$ | $\mathbf{0.904}_{015}$ | $\mathbf{0.544}_{030}$ | $1.31h$ |
| | Base PerPatch | $\mathbf{0.938}_{002}$ | $\mathbf{0.873}_{009}$ | $\mathbf{0.883}_{014}$ | $\mathbf{0.087}_{005}$ | $\mathbf{0.938}_{003}$ | $\mathbf{0.364}_{017}$ | $0.88h$ | $0.950_{011}$ | $\mathbf{0.910}_{055}$ | $0.908_{039}$ | $0.122_{016}$ | $0.862_{017}$ | $0.489_{035}$ | $1.40h$ |

Table 3: Results of MIL pooling methods on TCGA-Lung and SICAP-MIL [79, 80], reported as mean$_{std}$ for performance metrics. Times are given as pre-training (days) / MIL training (minutes).

| Encoder | Method | TCGA-Lung | | | | | | SICAP-MIL | | | | | |
|---|---|---|---|---|---|---|---|---|---|---|---|---|---|
| | | Slide | | | | Resource | | Slide | | | | Resource | |
| | | AUC | ACC | F1 | ECE | Time | #Params | AUC | ACC | F1 | ECE | Time | #Params |
| DINO Natural | max pooling | $0.881_{008}$ | $0.794_{010}$ | $0.813_{008}$ | $0.107_{010}$ | 16.66/9.22 | 385 | $0.829_{005}$ | $0.798_{016}$ | $0.770_{013}$ | $\mathbf{0.101}_{013}$ | 16.66/1.156 | 1155 |
| | mean pooling | $0.867_{000}$ | $0.795_{000}$ | $0.793_{000}$ | $0.076_{000}$ | 16.66/7.135 | 385 | $0.820_{001}$ | $0.745_{000}$ | $0.757_{012}$ | $0.131_{001}$ | 16.66/1.24 | 1155 |
| | ABMIL | $0.855_{050}$ | $0.796_{085}$ | $\mathbf{0.831}_{042}$ | $0.185_{068}$ | 16.66/619.64 | 99074 | $0.770_{056}$ | $0.779_{026}$ | $0.732_{038}$ | $0.230_{011}$ | 16.66/31.76 | 99332 |
| | DSMIL | $0.879_{014}$ | $0.784_{006}$ | $0.801_{011}$ | $0.069_{013}$ | 16.66/652.48 | 66562 | $0.829_{020}$ | $0.789_{013}$ | $0.776_{019}$ | $0.127_{021}$ | 16.66/34.14 | 70406 |
| | Snuffy | $0.885_{013}$ | $0.785_{021}$ | $0.782_{009}$ | $0.172_{012}$ | 16.66/23.328 | 1776002 | $0.812_{013}$ | $0.745_{011}$ | $0.781_{022}$ | $0.105_{015}$ | 16.66/3.361 | 1777542 |
| | GTP | $0.805_{008}$ | $0.747_{022}$ | $0.774_{022}$ | $0.252_{017}$ | 16.66/300.1 | 131365 | $0.513_{106}$ | $0.554_{075}$ | $0.508_{141}$ | $0.390_{048}$ | 16.66/25.0 | 131495 |
| | LSE pooling | $0.881_{012}$ | $0.801_{018}$ | $0.806_{016}$ | $0.139_{011}$ | 16.66/10.873 | 74499 | $0.825_{001}$ | $0.785_{010}$ | $0.781_{029}$ | $0.123_{011}$ | 16.66/1.721 | 74499 |
| | Maxsoft pooling | $\mathbf{0.892}_{004}$ | $\mathbf{0.809}_{009}$ | $0.809_{018}$ | $\mathbf{0.066}_{006}$ | 16.66/17.177 | 74113 | $\mathbf{0.834}_{007}$ | $\mathbf{0.802}_{009}$ | $\mathbf{0.781}_{014}$ | $0.158_{009}$ | 16.66/2.868 | 74499 |
| DINO Domain | max pooling | $0.937_{002}$ | $0.879_{014}$ | $0.891_{009}$ | $0.080_{000}$ | 7.13/9.202 | 385 | $0.775_{078}$ | $0.770_{073}$ | $0.740_{073}$ | $0.219_{044}$ | 0.54/1.195 | 1155 |
| | mean pooling | $0.933_{001}$ | $0.849_{006}$ | $0.861_{010}$ | $0.244_{006}$ | 7.13/8.593 | 385 | $0.800_{002}$ | $0.747_{006}$ | $0.767_{007}$ | $0.179_{012}$ | 0.54/1.189 | 1155 |
| | ABMIL | $0.910_{002}$ | $0.870_{002}$ | $0.886_{003}$ | $0.090_{002}$ | 7.13/631.5 | 99074 | $0.806_{032}$ | $0.803_{035}$ | $0.762_{034}$ | $0.213_{028}$ | 0.54/31.92 | 99332 |
| | DSMIL | $0.937_{004}$ | $0.864_{010}$ | $0.883_{006}$ | $\mathbf{0.033}_{013}$ | 7.13/650.92 | 66562 | $0.835_{002}$ | $0.797_{011}$ | $0.787_{032}$ | $\mathbf{0.162}_{014}$ | 0.54/34.26 | 70406 |
| | Snuffy | $0.921_{007}$ | $0.868_{000}$ | $0.882_{005}$ | $0.108_{011}$ | 7.13/23.204 | 1776002 | $0.827_{035}$ | $0.750_{004}$ | $0.792_{029}$ | $0.191_{011}$ | 0.54/3.375 | 1777542 |
| | GTP | $0.727_{134}$ | $0.669_{115}$ | $0.687_{122}$ | $0.321_{110}$ | 7.13/300.1 | 131365 | $0.557_{134}$ | $0.566_{114}$ | $0.635_{014}$ | $0.357_{077}$ | 0.54/25.0 | 131495 |
| | LSE pooling | $0.930_{006}$ | $0.883_{005}$ | $0.894_{008}$ | $0.113_{006}$ | 7.13/10.034 | 74113 | $0.840_{006}$ | $0.806_{005}$ | $0.783_{027}$ | $0.223_{012}$ | 0.54/1.667 | 74499 |
| | Maxsoft pooling | $\mathbf{0.940}_{004}$ | $\mathbf{0.885}_{007}$ | $\mathbf{0.894}_{013}$ | $0.094_{002}$ | 7.13/17.145 | 74113 | $\mathbf{0.850}_{011}$ | $\mathbf{0.808}_{005}$ | $\mathbf{0.796}_{014}$ | $0.211_{011}$ | 0.54/2.919 | 74499 |
| UNI | max pooling | $0.931_{006}$ | $0.868_{024}$ | $0.883_{013}$ | $0.123_{017}$ | −/11.915 | 1025 | $0.852_{053}$ | $0.841_{050}$ | $0.809_{043}$ | $0.163_{029}$ | −/1.3 | 3075 |
| | mean pooling | $0.930_{004}$ | $0.855_{007}$ | $0.882_{009}$ | $0.140_{004}$ | −/12.162 | 1025 | $0.825_{001}$ | $0.744_{002}$ | $0.742_{006}$ | $0.255_{001}$ | −/1.279 | 3075 |
| | ABMIL | $0.864_{042}$ | $0.864_{043}$ | $0.864_{043}$ | $0.137_{043}$ | −/1686.16 | 263554 | $0.807_{044}$ | $0.823_{032}$ | $0.770_{049}$ | $0.229_{034}$ | −/63.18 | 263812 |
| | DSMIL | $0.941_{008}$ | $0.876_{016}$ | $0.872_{017}$ | $0.068_{012}$ | −/1635.3 | 149762 | $0.877_{002}$ | $0.827_{014}$ | $0.829_{001}$ | $0.174_{005}$ | −/65.78 | 160006 |
| | Snuffy | $0.927_{004}$ | $0.878_{009}$ | $0.845_{007}$ | $\mathbf{0.067}_{015}$ | −/41.466 | 12600322 | $0.844_{004}$ | $0.814_{007}$ | $0.796_{028}$ | $0.181_{019}$ | −/4.181 | 12604422 |
| | GTP | $0.926_{038}$ | $0.834_{044}$ | $0.876_{039}$ | $0.168_{069}$ | −/380.1 | 172325 | $0.719_{021}$ | $0.700_{024}$ | $0.676_{021}$ | $0.271_{031}$ | −/26.1 | 172455 |
| | LSE pooling | $0.878_{011}$ | $0.868_{015}$ | $0.873_{018}$ | $0.132_{015}$ | −/13.622 | 525313 | $0.879_{011}$ | $0.874_{013}$ | $0.838_{009}$ | $0.145_{006}$ | −/1.749 | 526339 |
| | Maxsoft pooling | $\mathbf{0.942}_{002}$ | $\mathbf{0.882}_{006}$ | $\mathbf{0.883}_{004}$ | $0.110_{005}$ | −/24.281 | 525313 | $\mathbf{0.891}_{002}$ | $\mathbf{0.877}_{011}$ | $\mathbf{0.841}_{008}$ | $\mathbf{0.142}_{004}$ | −/3.123 | 526339 |
| Prov-GigaPath | max pooling | $0.950_{005}$ | $0.900_{007}$ | $0.905_{004}$ | $0.062_{008}$ | 162.56/18.696 | 1537 | $0.790_{052}$ | $0.767_{046}$ | $0.743_{059}$ | $0.227_{036}$ | 162.56/1.348 | 4611 |
| | mean pooling | $0.947_{002}$ | $0.886_{002}$ | $0.897_{004}$ | $0.110_{003}$ | 162.56/19.393 | 1537 | $0.805_{001}$ | $0.767_{002}$ | $0.725_{004}$ | $0.258_{001}$ | 162.56/2.48 | 1537 |
| | ABMIL | $0.956_{001}$ | $0.892_{008}$ | $0.907_{016}$ | $0.106_{006}$ | 162.56/2537.52 | 395138 | $0.701_{113}$ | $0.722_{029}$ | $0.663_{122}$ | $0.283_{023}$ | 162.56/88.44 | 395396 |
| | DSMIL | $0.957_{002}$ | $0.900_{008}$ | $0.913_{009}$ | $0.056_{011}$ | 162.56/2629.860 | 216322 | $0.844_{014}$ | $0.803_{008}$ | $0.802_{006}$ | $0.175_{003}$ | 162.56/93.24 | 231686 |
| | Snuffy | $0.954_{005}$ | $0.895_{008}$ | $0.904_{007}$ | $0.087_{009}$ | 162.56/61.587 | 28337666 | $0.786_{018}$ | $0.779_{002}$ | $0.725_{034}$ | $0.223_{001}$ | 162.56/7.442 | 28343814 |
| | GTP | $0.891_{052}$ | $0.786_{049}$ | $0.841_{031}$ | $0.209_{046}$ | 162.56/390.8 | 205093 | $0.621_{197}$ | $0.601_{115}$ | $0.509_{298}$ | $0.358_{094}$ | 162.56/27.9 | 205223 |
| | LSE pooling | $0.950_{005}$ | $0.896_{006}$ | $0.913_{007}$ | $0.103_{001}$ | 162.56/21.672 | 1181185 | $0.830_{063}$ | $0.790_{065}$ | $0.788_{050}$ | $0.218_{053}$ | 162.56/1.812 | 1182723 |
| | Maxsoft pooling | $\mathbf{0.966}_{004}$ | $\mathbf{0.905}_{008}$ | $\mathbf{0.924}_{008}$ | $\mathbf{0.054}_{013}$ | 162.56/29.919 | 1181185 | $\mathbf{0.872}_{028}$ | $\mathbf{0.821}_{032}$ | $\mathbf{0.805}_{037}$ | $0.172_{014}$ | 162.56/1.812 | 1182723 |
| IN | R²T-MIL | $0.928_{000}$ | $0.855_{006}$ | $0.869_{015}$ | $0.138_{006}$ | −/28.937 | 2696961 | $0.854_{008}$ | $0.823_{009}$ | $0.798_{012}$ | $0.168_{004}$ | −/4.778 | 2697987 |
| PLIP | R²T-MIL | $0.945_{006}$ | $0.881_{003}$ | $0.886_{004}$ | $0.100_{005}$ | −/24.252 | 2434817 | $0.832_{011}$ | $0.793_{007}$ | $0.794_{004}$ | $0.222_{021}$ | −/4.601 | 2435843 |
| UNI | PANTHER | $0.902_{013}$ | $0.884_{005}$ | $0.884_{005}$ | $0.115_{005}$ | −/50.13 | 65570 | $0.716_{026}$ | $0.734_{014}$ | $0.689_{065}$ | $0.148_{024}$ | −/26.01 | 98355 |

Table 4: Results of augmentation methods on TCGA-Lung and SICAP-MIL [79, 80], reported as mean$_{std}$ for performance metrics. Times are given as MIL training (hours).

| Method | Augmentation | TCGA-Lung | | | | | SICAP-MIL | | | | |
|---|---|---|---|---|---|---|---|---|---|---|---|
| | | Slide | | | | Resource | Slide | | | | Resource |
| | | AUC | ACC | F1 | ECE | Time | AUC | ACC | F1 | ECE | Time |
| Maxsoft pooling | No Aug | $0.940_{004}$ | $\mathbf{0.885}_{007}$ | $0.894_{013}$ | $0.094_{002}$ | $0.29h$ | $0.850_{011}$ | $0.808_{005}$ | $0.796_{014}$ | $0.211_{011}$ | $0.05h$ |
| | ReMix | $0.930_{003}$ | $0.865_{006}$ | $0.890_{001}$ | $0.081_{002}$ | $8.25h$ | $0.849_{008}$ | $0.814_{014}$ | $0.772_{008}$ | $0.180_{017}$ | $0.13h$ |
| | RankMix | $0.935_{003}$ | $0.855_{006}$ | $0.886_{001}$ | $0.098_{002}$ | $3.5h$ | $0.845_{008}$ | $\mathbf{0.816}_{014}$ | $0.782_{008}$ | $0.280_{017}$ | $0.07h$ |
| | AugDiff | $0.934_{007}$ | $0.812_{025}$ | $0.856_{005}$ | $0.126_{014}$ | $527h$ | $0.818_{019}$ | $0.787_{011}$ | $0.778_{017}$ | $0.180_{005}$ | $5h$ |
| | SSRDL | $0.944_{004}$ | $0.834_{015}$ | $0.896_{005}$ | $0.116_{014}$ | $84.5h$ | $0.838_{009}$ | $0.798_{009}$ | $0.788_{016}$ | $\mathbf{0.161}_{005}$ | $4h$ |
| | PerSlide | $0.938_{002}$ | $0.877_{006}$ | $0.895_{010}$ | $0.062_{005}$ | $14h$ | $\mathbf{0.857}_{008}$ | $0.795_{006}$ | $\mathbf{0.802}_{019}$ | $0.198_{012}$ | $0.23h$ |
| | PerPatch | $\mathbf{0.946}_{001}$ | $0.876_{005}$ | $\mathbf{0.904}_{001}$ | $\mathbf{0.051}_{012}$ | $15h$ | $0.827_{002}$ | $0.795_{004}$ | $0.767_{011}$ | $0.223_{018}$ | $0.21h$ |

**Effect of $\beta$ in LSE pooling.** We assess the sensitivity of the LSE pooling operation to the temperature parameter $\beta$, evaluating values in the range $\{0.5, 1, 2, 3.5, 5, 7.5, 10\}$. As shown in Figure 4, lower $\beta$ values approximate mean pooling behavior, while higher values approach max pooling. The results indicate a trade-off: smaller $\beta$ values reduce variance but underemphasize salient features, whereas larger values overly amplify them. We find that $\beta = 5$ offers the best compromise, yielding the highest performance on both WSI and patch classification.

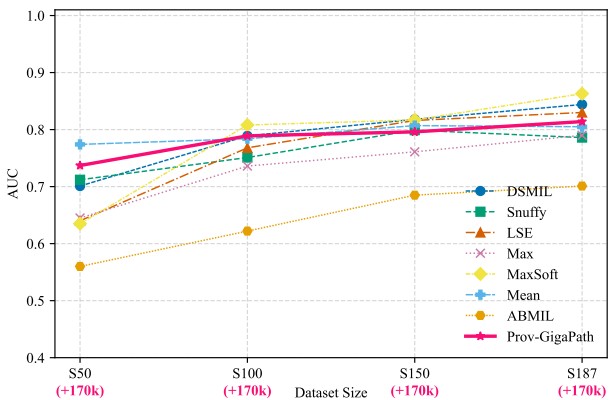

Figure 3: Performance of major Transformer-based and classical MIL pooling methods on the SICAP-MIL dataset across different training set sizes.

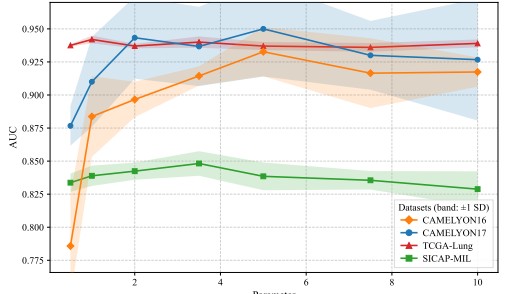

Figure 4: The impact of $\beta$ in Maxsoft architecture on CAMELYON16, CAMELYON17, TCGA-Lung, and SICAP-MIL datasets.

| Augmentation | Slide | | Patch |
|---|---|---|---|
| | AUC | ECE | AUC |
| No Aug | $0.983_{.055}$ | $0.121_{.051}$ | $0.839_{.019}$ |
| Random Rotation | $0.965_{.021}$ | $0.133_{.060}$ | $0.854_{.020}$ |
| Random Elastic Deformation | $0.890_{.028}$ | $0.193_{.003}$ | $0.861_{.031}$ |
| Random Affine Transformation | $0.900_{.113}$ | $0.193_{.102}$ | $0.822_{.020}$ |
| Random Gaussian Blurring | $0.995_{.007}$ | $0.081_{.002}$ | $0.882_{.011}$ |
| Random Color Jitter | $0.955_{.064}$ | $0.084_{.001}$ | $0.876_{.045}$ |
| Random HED Jitter | $0.955_{.050}$ | $0.156_{.047}$ | $0.874_{.039}$ |

Table 5: Result of each base augmentation on the CAMELYON17 dataset, reported as mean$_{.std}$.

**Impact of Base Augmentations.** To evaluate the augmentations from Section 5.4, we ablated them individually. We also examined the spatial transforms Random Elastic Deformation and Random Affine Transformation. As shown in Table 5, these spatial operations consistently reduced performance. We hypothesize that such transformations fail to capture clinical variability and primarily inject noise rather than useful diversity.

# 7   Conclusion and Discussion

This work highlights the value of instance-level representation learning under current data constraints in WSI diagnosis. In response, we present a theoretical and empirical analysis of classical MIL pooling and introduce Maxsoft—a simple, resource-efficient pooling function with strong, adaptable performance. A limitation is that, despite clinically acceptable visualizations, patch-level performance lags behind WSI-level accuracy. Another weakness is that, while our theory offers numerical guidance for selecting $\beta$ in Maxsoft, a small dataset-specific search is still required; future work should make this selection tuning-free and integrated into training. Moreover, standard augmentations (e.g., Random Affine, Elastic Deformation) do not capture real distribution shifts, underscoring the need for more realistic techniques. Finally, we advocate studying cancer detection via anomaly detection [83–88], especially in data-scarce regimes, since it naturally targets rare or novel abnormalities without large labeled cancer sets [89–92].

## Acknowledgments

We thank Mehrab Moradzadeh, Mohammad Mosayyebi, Arian Komaei Koma, Amir Hossein Saberi, Mohammad Azizmalayeri, Hossein Mirzaei, Hosein Hasani, and the anonymous reviewers for their helpful discussions and feedback on this work.

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

# A  Theoretical Analysis

In this section, we first provide a mathematical formulation of the multiple instance learning (MIL) setting considered in this work. We begin by establishing the notation and then describe the main problem setup, along with the data generation model. Specifically, we introduce a statistical model that captures the generation process of images in the dataset, as well as the set (or bag) of patches associated with each image. Our modeling approach is intentionally kept general, incorporating only standard and widely accepted assumptions such as the local dependence of image patches and the i.i.d. nature of images in the training set. Additionally, although our experiments use the cross-entropy loss, the theoretical analysis accommodates any proper, continuous, Lipschitz loss function.

We then proceed to the theoretical analysis, starting with uniform convergence bounds that provide generalization guarantees for our method. In particular, we study the role of the hyperparameter $\beta$ in the log-sum-exp formulation and its influence on generalization. Our main result in this part is stated in Theorem 4.

Subsequently, we shift focus to the sensitivity analysis in Section A.4. Notably, in contrast to generalization—which favors smaller values of $\beta$—high sensitivity in noisy regimes requires sufficiently large $\beta$. In Theorem 6, we show that choosing $\beta \gg \mathcal{O}(\log N)$ is both necessary and sufficient to achieve a high true positive rate and to mitigate false negatives.

The key conclusion of this section is that all values of $\beta \in (0, +\infty)$ are *Pareto-optimal*: depending on which aspect—generalization or sensitivity—is prioritized and how the trade-off is weighted, the optimal choice of $\beta$ will vary. This insight supports the central idea behind our method: using smaller values of $\beta$ during training to promote generalization, while selecting larger values (possibly even $\infty$) at inference time to enhance sensitivity.

## A.1  Data Generation Model

Let $n, N \in \mathbb{N}$. Assume we observe $n$ i.i.d. images $I_1, \ldots, I_n \overset{i.i.d.}{\sim} P_0$, where $P_0 \in \mathcal{M}(\mathbb{R}^{D_1 \times D_2})$ is an unknown distribution over images of size $D_1 \times D_2$, with $D_1, D_2 \in \mathbb{N}$. Assume each image $I_j$ for $j \in [n]$ is decomposed into $N$ non-overlapping (or possibly overlapping) patches:

$$\mathsf{Patch}(I_j) \triangleq \left( \boldsymbol{X}_1^{(j)}, \ldots, \boldsymbol{X}_N^{(j)} \right).$$

Each image patch $\boldsymbol{X}_i^{(j)}$ (for $j \in [n]$, $i \in [N]$) is passed through a fixed vision transformer model network $\mathsf{ViT}(\cdot)$ to produce a feature vector $\boldsymbol{x}_i^{(j)}$:

$$\boldsymbol{x}_i^{(j)} \triangleq \mathsf{ViT}\left( \boldsymbol{X}_i^{(j)} \right) \in \mathcal{X}, \tag{2}$$

where $\mathcal{X} \subseteq \mathbb{R}^d$ is the feature space, and $d$ denotes the feature dimension. Define the *bag of features* for image $I_j$ as

$$\mathsf{Bag}(I_j) \triangleq \left( \boldsymbol{x}_1^{(j)}, \ldots, \boldsymbol{x}_N^{(j)} \right), \quad j \in [n].$$

**Assumption 1** (Local Dependence of Image Patches)**.** *Let $I \sim P_0$, and consider the $N$ feature vectors $(\boldsymbol{x}_1, \ldots, \boldsymbol{x}_N)$ in $\mathsf{Bag}(I)$. Then, we assume they form a locally dependent stochastic process such that:*

$$\boldsymbol{x}_i \perp \boldsymbol{x}_j \quad \forall i, j \in [N] \text{ with } |i - j| > B, \tag{3}$$

*where $\perp$ indicates statistical independence according to $P_0$, and $1 \le B \ll N$ is a fixed parameter independent of $N$.*

This assumption has been widely validated empirically across a broad range of natural and medical image datasets. Indeed, the statistical correlations among different patches of such images are predominantly local: patches that are spatially distant—i.e., whose patch indices differ by more than a certain *bandwidth* $B = \mathcal{O}(1)$—are typically almost independent, even though they might be identically or non-identically distributed.

## A.2 Model Training via SGD

Consider the class of Multi-Layer Perceptrons (MLPs) defined as

$$\mathcal{F} \triangleq \left\{ f_{\boldsymbol{\theta}} : \mathcal{X} \to \mathbb{R} \mid \boldsymbol{\theta} \in \Theta \right\},$$

where $\Theta$ denotes the parameter space, i.e., the set of all possible weight configurations of the neural networks. For each $\boldsymbol{\theta} \in \Theta$, the function $f_{\boldsymbol{\theta}}$ represents an MLP that maps input feature vectors from $\mathcal{X}$ to real-valued outputs.

**Definition 3** (LSE Loss). Fix $\beta \in (0, +\infty)$ and a Lipschitz loss function $\mathcal{L}(y\|y') : [0,1]^2 \to \mathbb{R}_+$ for (soft) labels $y, y' \in [0,1]$. For $\boldsymbol{x} \in \mathcal{X}$, define the *log-sum-exp aggregation function*:

$$A\left(\boldsymbol{x}_1, \ldots, \boldsymbol{x}_N\right) \triangleq \frac{1}{\beta} \log \left( \frac{1}{N} \sum_{i=1}^{N} \exp \left( \beta \cdot \sigma \circ f_{\boldsymbol{\theta}}(\boldsymbol{x}_i) \right) \right),$$

where $\sigma \circ f_{\boldsymbol{\theta}}$ is the composition of a learned function $f_{\boldsymbol{\theta}} : \mathcal{X} \to \mathbb{R}$ and an activation $\sigma : \mathbb{R} \to [0,1]$. The LSE-based loss function for parameters $\boldsymbol{\theta} \in \Theta$ for the empirical dataset $\{(I_j, y_j) \mid j = 1, \ldots, n\}$ is then given by:

$$\mathsf{L}_{\mathrm{LSE}}(\boldsymbol{\theta}) \triangleq \frac{1}{n} \sum_{j=1}^{n} \mathcal{L}\left( y_j \,\Big\|\, A\left( \boldsymbol{x}_1^{(j)}, \ldots, \boldsymbol{x}_N^{(j)} \right) \right). \tag{4}$$

Note that we have

$$\lim_{\beta \to 0} A\left(\boldsymbol{x}_1, \ldots, \boldsymbol{x}_N\right) = \frac{1}{N} \sum_{i=1}^{N} \sigma \circ f_{\boldsymbol{\theta}}(\boldsymbol{x}_i),$$

$$\text{while} \quad \lim_{\beta \to \infty} A\left(\boldsymbol{x}_1, \ldots, \boldsymbol{x}_N\right) = \max_{i \in [N]} \sigma \circ f_{\boldsymbol{\theta}}(\boldsymbol{x}_i). \tag{5}$$

Consider the loss function defined in Definition 3, and suppose that an empirical risk minimization (ERM) procedure yields the estimator $\widehat{\boldsymbol{\theta}}_{\mathrm{ERM}}$ defined as

$$\widehat{\boldsymbol{\theta}}_{\mathrm{ERM}} \triangleq \operatorname*{argmin}_{\boldsymbol{\theta} \in \Theta} \mathsf{L}_{\mathrm{LSE}}(\boldsymbol{\theta}). \tag{6}$$

The minimization problem is assumed to be solved using a standard Stochastic Gradient Descent (SGD) algorithm with an arbitrary batch size. It is important to note that such optimization problems are generally non-convex and, therefore, are not expected to be solved to global optimality. In practice, due to early stopping and the inherent complexity of the loss landscape, the algorithm typically converges to a local minimum or a stationary point. However, this does not impact the validity of our subsequent analysis. Specifically, our generalization guarantees are stated as high-probability uniform convergence bounds that hold for all $\boldsymbol{\theta} \in \Theta$, regardless of the particular solution returned by the optimization algorithm.

## A.3 Generalization Bounds

Define the generalization error as

$$\mathsf{GE}\left(\widehat{\boldsymbol{\theta}}_{\mathrm{ERM}}\right) \triangleq \mathbb{E}_{P_0}\left[ \mathsf{L}_{\mathrm{LSE}}\left(\widehat{\boldsymbol{\theta}}_{\mathrm{ERM}}\right) \right] - \mathsf{L}_{\mathrm{LSE}}\left(\widehat{\boldsymbol{\theta}}_{\mathrm{ERM}}\right).$$

**Theorem 4** (Effect of $\beta$ on Generalization Error). *Let Assumption 1 hold for some $B$, and define $q_{i,j} \triangleq \sigma \circ f_{\widehat{\boldsymbol{\theta}}_{\mathrm{ERM}}}(\boldsymbol{x}_i^j)$ for all images $j \in [n]$ and patch ids $i \in [N]$. Also let $U \triangleq \frac{1}{n} \sum_{j=1}^{n} \max_i q_{i,j}$ and $M \triangleq \frac{1}{n} \sum_{j=1}^{n} \operatorname{median}_i q_{i,j}$. Then, having $\beta \ll \frac{\log(N/2)}{U-M}$, with high probability results into the following bound:*

$$\mathsf{GE}(\widehat{\boldsymbol{\theta}}_{\mathrm{ERM}}) \leq \mathcal{O}\left( (nN)^{-1/2} + n^{-1} \right).$$

*On the other hand, if $\beta \gg \frac{\log(N/2)}{U-M}$, then, depending on the tail behavior of the $q_{i,j}$'s, the generalization error may (again with high probability) satisfy*

$$\mathsf{GE}(\widehat{\boldsymbol{\theta}}_{\mathrm{ERM}}) \asymp \mathcal{O}(n^{-1/2}).$$

*Proof of Theorem 4.* We first show that the LSE behaves similarly to an exponentially weighted average, exhibiting a smooth phase transition as $\beta$ increases from below to above the threshold

$$\frac{\log \frac{N}{2}}{U - M}.$$

In the small-$\beta$ regime, the LSE behaves roughly like a simple average, and thus the variance of the loss decreases with both $n$ and $N$. In contrast, in the large-$\beta$ regime, the LSE approximates the maximum operator, which may exhibit weaker concentration properties depending on the tail behavior of the variables $\sigma \circ f_{\boldsymbol{\theta}}(\boldsymbol{x}_i^{(j)})$.

**Lemma 1.** *For $\beta \geq 0$, the log-sum-exp (LSE) and the exponentially weighted average of $N$ values $q_1, \ldots, q_N$ bound each other as follows:*

$$\left| \frac{1}{\beta} \log \left( \frac{1}{N} \sum_{i=1}^{N} e^{\beta q_i} \right) - \sum_{i=1}^{N} q_i \cdot \frac{e^{\beta q_i}}{\sum_{j=1}^{N} e^{\beta q_j}} \right| \leq C \cdot \min \left\{ \beta, \frac{\log N}{\beta} \right\}, \tag{7}$$

*where $C$ is a constant independent of $\beta$, but dependent on the empirical distribution of the $q_i$s.*

*Proof.* For $\beta \ll 1$, we use the Taylor expansion of the exponential function: $e^x = 1 + x + \frac{x^2}{2} + \mathcal{O}(x^3)$. Applying this to $e^{\beta q_i}$, we get

$$\frac{1}{\beta} \log \left( \frac{1}{N} \sum_{i=1}^{N} e^{\beta q_i} \right) = \frac{1}{\beta} \log \left( \frac{1}{N} \sum_{i=1}^{N} \left( 1 + \beta q_i + \frac{\beta^2 q_i^2}{2} + \mathcal{O}(\beta^3) \right) \right)$$

$$= \frac{1}{\beta} \log \left( 1 + \beta \bar{q} + \frac{\beta^2}{2} \bar{q^2} + \mathcal{O}(\beta^3) \right)$$

$$= \bar{q} + \frac{\beta}{2} \left( \bar{q^2} - \bar{q}^2 \right) + \mathcal{O}(\beta^2), \tag{8}$$

where $\bar{q} = \frac{1}{N} \sum_{i=1}^{N} q_i$ and $\bar{q^2} = \frac{1}{N} \sum_{i=1}^{N} q_i^2$. For the exponentially weighted average, we expand the weights similarly:

$$\sum_{i=1}^{N} q_i \cdot \frac{e^{\beta q_i}}{\sum_{j=1}^{N} e^{\beta q_j}} = \sum_{i=1}^{N} q_i \cdot \frac{1 + \beta q_i + \mathcal{O}(\beta^2)}{N \left( 1 + \beta \bar{q} + \mathcal{O}(\beta^2) \right)}$$

$$= \bar{q} + \beta \left( \bar{q^2} - \bar{q}^2 \right) + \mathcal{O}(\beta^2). \tag{9}$$

Thus, the absolute difference between the two expressions is

$$|A - B| \leq \frac{\beta}{2} \cdot \mathrm{Var}(q_1, \ldots, q_N) + \mathcal{O}(\beta^2).$$

For $\beta \gg 1$, denote $q_{\max} = \max_i q_i$ and let $i^*$ be the index where this maximum is achieved. We write

$$\frac{1}{\beta} \log \left( \frac{1}{N} \sum_{i=1}^{N} e^{\beta q_i} \right) = \frac{1}{\beta} \log \left( \frac{1}{N} e^{\beta q_{\max}} \left[ 1 + \sum_{i \neq i^*} e^{-\beta(q_{\max} - q_i)} \right] \right)$$

$$= q_{\max} - \frac{\log N}{\beta} + \frac{1}{\beta} \log \left( 1 + \sum_{i \neq i^*} e^{-\beta(q_{\max} - q_i)} \right)$$

$$= q_{\max} - \frac{\log N}{\beta} + \mathcal{O}(\beta^{-2}). \tag{10}$$

For the exponentially weighted average,

$$\sum_{i=1}^{N} q_i \cdot \frac{e^{\beta q_i}}{\sum_{j=1}^{N} e^{\beta q_j}} = \frac{q_{\max} + \sum_{i \neq i^*} q_i e^{-\beta(q_{\max} - q_i)}}{1 + \sum_{i \neq i^*} e^{-\beta(q_{\max} - q_i)}}$$

$$= q_{\max} + \mathcal{O}(\beta^{-2}). \tag{11}$$

Therefore, the difference in this regime satisfies $|A - B| \leq \frac{\log N}{\beta} + \mathcal{O}(\beta^{-2})$. Combining both cases, we obtain the claimed bound:

$$\left| \frac{1}{\beta} \log \left( \frac{1}{N} \sum_{i=1}^{N} e^{\beta q_i} \right) - \sum_{i=1}^{N} q_i \cdot \frac{e^{\beta q_i}}{\sum_{j=1}^{N} e^{\beta q_j}} \right| \leq C \cdot \min \left\{ \beta, \frac{\log N}{\beta} \right\}.$$

$\square$

Hence, the two formulations become asymptotically equivalent as $\beta \to 0^+$ and as $\beta \to \infty$. In these limiting regimes—which are of primary interest in this analysis—we may treat the two expressions interchangeably.

**Lemma 2.** *Assume real values satisfying $q_1 \leq q_2 \leq \ldots \leq q_N$ and let $\beta \geq 0$. Define:*

$$\Delta \triangleq q_N - q_1, \quad \delta \triangleq q_{\lceil N/2 \rceil} - q_1.$$

*Then, for all $j = 1, \ldots, N$, the exponentially weighted probabilities satisfy:*

$$\frac{1}{N} \cdot \frac{2}{e^{\beta \delta} + e^{\beta \Delta}} \leq p_j \triangleq e^{\beta q_j} \left( \sum_{i=1}^{N} e^{\beta q_i} \right)^{-1} \leq \frac{1}{N} \cdot \frac{2e^{\beta \Delta}}{1 + e^{\beta \delta}}. \tag{12}$$

*Proof.* To prove the upper-bound, we can write

$$e^{\beta q_j} \left( \sum_{i=1}^{N} e^{\beta q_i} \right)^{-1} \leq e^{\beta q_{\max}} \left( \sum_{i=1}^{N} e^{\beta q_i} \right)^{-1}$$

$$= e^{\beta \Delta} \left( \sum_{i=1}^{N} e^{\beta(q_i - q_1)} \right)^{-1}$$

$$\leq e^{\beta \Delta} \left( \frac{N}{2} + \frac{N}{2} e^{\beta \delta} \right)^{-1}$$

$$= \frac{1}{N} \cdot \frac{2e^{\beta \Delta}}{1 + e^{\beta \delta}}. \tag{13}$$

The lower-bound can be achieved in a similar fashion:

$$e^{\beta q_j} \left( \sum_{i=1}^{N} e^{\beta q_i} \right)^{-1} \geq e^{\beta q_1} \left( \sum_{i=1}^{N} e^{\beta q_i} \right)^{-1}$$

$$= \left( \sum_{i=1}^{N} e^{\beta(q_i - q_1)} \right)^{-1}$$

$$\geq \left( \frac{N}{2} e^{\beta \Delta} + \frac{N}{2} e^{\beta \delta} \right)^{-1}$$

$$= \frac{1}{N} \cdot \frac{2}{e^{\beta \delta} + e^{\beta \Delta}}. \tag{14}$$

This completes the proof. $\square$

**Corollary 1.** *Under the setting of Lemma 2, for*

$$\beta \ll \frac{\log(N/2)}{\Delta - \delta},$$

*we have $p_j \leq c(\beta)/N$ for all $j$, where $c(\beta)$ is a constant independent of $N$. Conversely, when*

$$\beta \gg \frac{\log(N/2)}{\Delta - \delta},$$

*there exists an index $j^* \in [N]$ such that $p_{j^*} \geq 1 - c'(\beta)$ and $p_j \leq c'(\beta)$ for all $j \neq j^*$, with $c'(\beta)$ also independent of $N$. Moreover,*

$$\lim_{\beta \to 0^+} c(\beta) = 1, \quad \lim_{\beta \to \infty} c'(\beta) = 0.$$

*Proof.* The proof is straightforward. We only need to show that the quantity

$$\beta^* \triangleq \frac{\log(N/2)}{\Delta - \delta}$$

serves as the critical threshold such that, for all $\beta < \beta^*$, the upper bound in Lemma 2 drops below 1. Specifically, we verify that

$$\frac{1}{N} \cdot \frac{2e^{\beta^* \Delta}}{1 + e^{\beta^* \delta}} \leq 1 \quad \implies \quad \beta^* \simeq \frac{\log \frac{N}{2}}{\Delta - \delta}.$$

For $\beta \ll \beta^*$, the resulting distribution over the $p_j$ values is nearly uniform. In contrast, when $\beta \gg \beta^*$, the probability mass concentrates sharply on the index $j$ corresponding to $q_j = q_{\max}$, with $p_j \approx 1$ and all other $p_j$ values approaching zero. $\square$

**Lemma 3** (Uniform Convergence Bound in Eq. (2) of [93]). *Let $\mathcal{H} = \{h_{\boldsymbol{\theta}} : \mathcal{X} \to \mathbb{R} \mid \boldsymbol{\theta} \in \Theta\}$ be a learnable class of functions, and let $P_0$ be a probability measure over $\mathcal{X}$. Suppose $\ell : \mathcal{X} \times \Theta \to \mathbb{R}$ is a loss function satisfying mild regularity conditions. Then, with high probability and uniformly over all $\boldsymbol{\theta} \in \Theta$, the following generalization bound holds:*

$$\mathbb{E}_{P_0}\left[\ell(\boldsymbol{X}; \boldsymbol{\theta})\right] \leq \frac{1}{n} \sum_{i=1}^{n} \ell(\boldsymbol{X}_i; \boldsymbol{\theta}) + C_1 \sqrt{\frac{\mathsf{Var}\left(\ell(\boldsymbol{X}; \boldsymbol{\theta})\right)}{n}} + \frac{C_2}{n}, \tag{15}$$

*where $\boldsymbol{X}_1, \ldots, \boldsymbol{X}_n \overset{i.i.d.}{\sim} P_0$, and $C_1$, $C_2$ are constants depending on the model and confidence parameters.*

The proof can be found in [93], as well as in several other related works cited therein. To bound the generalization gap, we must bound the variance $\mathsf{Var}\left(\ell(I, y; \boldsymbol{\theta})\right)$, where

$$\ell(I, y; \boldsymbol{\theta}) = \mathcal{L}\left(y \left\| \frac{1}{\beta} \log\left(\frac{1}{N} \sum_{i=1}^{N} e^{\beta \sigma \circ f_{\boldsymbol{\theta}}(\boldsymbol{x}_i)}\right)\right.\right),$$

and $(I, y)$ is a sample drawn from $P_0$, with $\mathsf{Bag}(I) = (\boldsymbol{x}_1, \ldots, \boldsymbol{x}_N)$.

**Lemma 4.** *Assume that the function $h \mapsto \mathcal{L}(y \| h)$ is $L$-Lipschitz for some $L \geq 0$ and all $y \in [0, 1]$. Then, by the Efron-Stein inequality,*

$$\mathsf{Var}\left(\ell(I, y; \boldsymbol{\theta})\right) \leq L^2 \cdot \mathsf{Var}\left(\frac{1}{\beta} \log\left(\frac{1}{N} \sum_{i=1}^{N} e^{\beta \sigma \circ f_{\boldsymbol{\theta}}(\boldsymbol{x}_i)}\right)\right).$$

The proof follows directly from the Lipschitz continuity of $\mathcal{L}$. Recall that due to the statement of the theorem we assume $\mathcal{L} : [0, 1]^2 \to \mathbb{R}$ is smooth and Lipschitz. Define

$$U \triangleq \frac{1}{n} \sum_{j=1}^{n} \max_{i=1,\ldots,N} \sigma\left(f_{\boldsymbol{\theta}}(x_i^{(j)})\right), \quad M \triangleq \frac{1}{n} \sum_{j=1}^{n} \underset{i=1,\ldots,N}{\mathrm{median}} \, \sigma\left(f_{\boldsymbol{\theta}}(x_i^{(j)})\right).$$

**Analysis for small $\beta$:** Assume that

$$\beta \ll \frac{\log(N/2)}{U - M}.$$

Then, using Lemma 2 and related results, we obtain:

$$\begin{aligned}
\mathsf{Var}\left(\ell(I, y; \boldsymbol{\theta})\right) &\leq L^2 c^2(\beta) \cdot \mathsf{Var}\left(\frac{1}{N} \sum_{i=1}^{N} \sigma \circ f_{\boldsymbol{\theta}}(\boldsymbol{x}_i)\right) \\
&= \frac{L^2 c^2(\beta)}{N^2} \sum_{i,j=1}^{N} \mathsf{Cov}\left(\sigma \circ f_{\boldsymbol{\theta}}(\boldsymbol{x}_i), \sigma \circ f_{\boldsymbol{\theta}}(\boldsymbol{x}_j)\right) \\
&\overset{(i)}{=} \frac{L^2 c^2(\beta)}{N^2} \sum_{\substack{i,j=1 \\ |i-j| \leq B}}^{N} \mathsf{Cov}\left(\sigma \circ f_{\boldsymbol{\theta}}(\boldsymbol{x}_i), \sigma \circ f_{\boldsymbol{\theta}}(\boldsymbol{x}_j)\right) \\
&\overset{(ii)}{\leq} \frac{B L^2 c^2(\beta)}{4N},
\end{aligned} \tag{16}$$

where (i) follows from a mixing or local-dependence assumption, and (ii) uses the fact that any variable bounded in $[0, 1]$ has variance at most $1/4$.

Therefore, the generalization gap is bounded by:

$$C_1 \sqrt{\frac{\mathsf{Var}\left(\ell(\boldsymbol{X}; \boldsymbol{\theta})\right)}{n}} + \frac{C_2}{n} \leq \mathcal{O}\left(\frac{1}{\sqrt{Nn}} + \frac{1}{n}\right),$$

where the constants are $\mathcal{O}(1)$ in the limit as $n, N \to \infty$, assuming $\beta$ is sufficiently small. These constants may also depend on other complexity measures (e.g., Rademacher complexity or VC-dimension in binary classification settings).

**Analysis for large $\beta$:**  Assume that

$$\beta \gg \frac{\log(N/2)}{U - M}.$$

Then, based on Corollary 1, we have

$$\mathsf{Var}\left(\ell(I, y; \boldsymbol{\theta})\right) \asymp L^2(1 - c'(\beta)) \cdot \mathsf{Var}\left(\max_{i=1,\ldots,N} \sigma \circ f_{\boldsymbol{\theta}}(\boldsymbol{x}_i)\right).$$

The term above—i.e., the variance of the maximum of $N$ weakly-dependent random variables—does not, in general, admit a closed-form analytical expression. Moreover, depending on the tail behavior of the variables $\sigma \circ f_{\boldsymbol{\theta}}(\boldsymbol{x}_i)$, the variance of their maximum may not even decrease with $N$.

**Lemma 5** (Var($\max_i Z_i$) for exponentials). *Assume $Z_1, \ldots, Z_N \overset{i.i.d.}{\sim} \lambda e^{-\lambda x}$ for $x \geq 0$ and some $\lambda > 0$, and define $M_N \triangleq \max_{i=1,\ldots,N} Z_i$. Then:*

$$\mathsf{Var}\left(M_N\right) = \frac{1}{\lambda^2} \sum_{k=1}^{N} \frac{1}{k^2},$$

*which converges to $\pi^2/(6\lambda^2)$ as $N \to \infty$, and hence does not vanish as $N$ grows.*

The proof can be found in Chapter 8 of [94]. In general, unless one is dealing with degenerate cases with sharply truncated support (e.g., a uniform distribution on $[0, 1]$), the variance of the *mean* decays faster than that of the *maximum* of $\lceil N/B \rceil$ random variables [94, 95]. This completes the proof. $\quad \square$

## A.4  Selection Sensitivity Analysis

In this section, we aim to theoretically analyze the effect of the parameter $\beta$ on the *sensitivity* of the inference-time performance of the model $\mathsf{LSE}_\beta\left(\sigma \circ f_{\boldsymbol{\theta}}(\boldsymbol{x}_{1:N})\right)$, where $\mathsf{LSE} * \beta$ is defined as:

$$\mathsf{LSE}_\beta(q_1, \ldots, q_N) \triangleq \frac{1}{\beta} \log\left(\frac{1}{N} \sum_{i=1}^{N} e^{\beta q_i}\right). \tag{17}$$

A commonly accepted assumption in the analysis of whole-slide histopathology images is that often only a small fraction of image patches may contain cancerous patterns—yet this is sufficient to label the entire slide as cancerous. The remaining patches may contain entirely normal tissue.

Another widely accepted assumption is that the values of $\sigma \circ f_{\boldsymbol{\theta}}(\boldsymbol{x}_i)$ across different patches of an image $I$—i.e., for $\mathsf{Bag}(I) = (\boldsymbol{x}_1, \ldots, \boldsymbol{x}_N)$ and some learned parameters $\boldsymbol{\theta} \in \Theta$—exhibit statistical variation. For instance, for "normal tissue" patches $\boldsymbol{x}_i \mid i \in \mathcal{N}$, the outputs are distributed around a mean value $\mu_0$, whereas for "abnormal" or "critical tissue" patches $\boldsymbol{x}_i \mid i \in \mathcal{C}$, they are distributed around a higher mean $\mu_c$, with $\mu_c > \mu_0$. Here, $\mathcal{N}, \mathcal{C} \subseteq [N]$ form a bipartition of the patch indices corresponding to normal and abnormal regions, respectively (so that $|\mathcal{N}| = N - |\mathcal{C}|$). In real-world scenarios, it is typically the case that $|\mathcal{C}| \ll N$.

The variances of the aforementioned class-conditional distributions, as well as the mean gap $\mu_c - \mu_0$, depend on how well the parameters $\boldsymbol{\theta}$ have been learned during training.

In what follows, we argue that—independent of the training quality (as captured by the class-conditional variances and the mean difference $\mu_c - \mu_0$)—choosing a larger value of $\beta$ leads to

improved accuracy bias and higher sensitivity, especially in the regime where $|\mathcal{C}| \ll N$. Conversely, when $|\mathcal{C}| \geq |\mathcal{N}|$, selecting a smaller value of $\beta$ is preferable. However, since the former condition ($|\mathcal{C}| \ll N$) is the one that arises in practice, our analysis supports choosing larger values of $\beta$ for better sensitivity.

**Assumption 2.** *Let $I \sim P_0$ denote a random image, and let $\mathsf{Bag}(I) = (\boldsymbol{x}_1, \ldots, \boldsymbol{x}_N)$ denote its random patches. Let $Q_1$ be a distribution supported over $[N] \cup \{0\}$. Then, suppose that $N_c \sim Q_1$ denotes the number of abnormal patches in the image, which can be zero upon being a "normal" image. Then, there exist $N_c$ unknown patches containing cancerous patterns, indexed by $\mathcal{C} \subseteq [N]$ if $N_c > 0$, and the remaining patches are normal, indexed by $\mathcal{N} = [N] \setminus \mathcal{C}$. For these patches, we assume:*

$$\sigma \circ f_{\boldsymbol{\theta}}(\boldsymbol{x}_i) \sim Q_n \quad for \ i \in \mathcal{N}, \tag{18}$$

$$\sigma \circ f_{\boldsymbol{\theta}}(\boldsymbol{x}_i) \sim Q_c \quad for \ i \in \mathcal{C}, \tag{19}$$

*where $Q_n$ and $Q_c$ are unknown distributions satisfying the following:*

- $\mathbb{E}_{Q_c}(X) - \mathbb{E}_{Q_n}(X) \geq \Delta$ *for some known* $\Delta > 0$,

- $\mathsf{Var}(Q_n), \mathsf{Var}(Q_c) \leq V$ *for some known* $V \geq 0$.

*No further assumptions are made about the distributions $Q_1$, $Q_n$, or $Q_c$, since they all depend on the learned parameter $\boldsymbol{\theta} \in \Theta$.*

For simplicity, we threshold the value of $\mathsf{LSE}_\beta(\sigma \circ f_{\boldsymbol{\theta}}(\boldsymbol{x}_{1:N}))$ for an image $I$ at

$$T \triangleq \frac{\mu_c + \mu_n}{2},$$

where values above this threshold are considered indicative of cancer, and values below it indicate a normal image. While this threshold can be optimized in more complex settings, such refinements do not affect the core analysis presented in this section. Also, consider the following standard definition for performance measure:

**Definition 5** (TP/FP/TN/FN Rates of $\boldsymbol{\theta}$). For a random image $I \sim P_0$ and corresponding bag of patch features $\boldsymbol{x}_1, \ldots, \boldsymbol{x}_N$, and under Assumption 2, we define the true/false positive (TP/FP) and true/false negative error rates of a given parameter $\boldsymbol{\theta} \in \Theta$ as follows:

- True Positive rate is the probability of $\mathsf{LSE}_\beta (\sigma \circ f_{\boldsymbol{\theta}}(\boldsymbol{x}_{1:N})) \geq T$ given $N_c \geq 1$,

- False Positive rate is the probability of $\mathsf{LSE}_\beta (\sigma \circ f_{\boldsymbol{\theta}}(\boldsymbol{x}_{1:N})) \geq T$ given $N_c = 0$,

- True Negative rate is the probability of $\mathsf{LSE}_\beta (\sigma \circ f_{\boldsymbol{\theta}}(\boldsymbol{x}_{1:N})) < T$ given $N_c = 0$,

- False Negative rate is the probability of $\mathsf{LSE}_\beta (\sigma \circ f_{\boldsymbol{\theta}}(\boldsymbol{x}_{1:N})) < T$ given $N_c \geq 1$.

**Theorem 6** (Effect of $\beta$ on Sensitivity). *Let Assumption 2 hold for some unknown sub-Gaussian distributions $Q_0, Q_1$, and $Q_2$. For $\alpha \in (0, 1)$, assume*

$$\beta \geq \frac{2C_\alpha \, \mathbb{E}\left[\log \frac{N}{N_c} \,\middle|\, N_c \geq 1\right]}{\Delta - 2C'_\alpha \sqrt{V} N_c^{-\gamma}} = \mathcal{O}\left(\frac{1}{\Delta} \mathbb{E}\left[\log \frac{N}{N_c} \,\middle|\, N_c \geq 1\right]\right),$$

*where $C_\alpha$ and $C'_\alpha$ have polylogarithmic dependence on $\alpha^{-1}$, and $\gamma \in (0, 1)$ is a constant depending on the tail properties of $Q_1$ and $Q_2$. Then, we have $\mathrm{TP} \geq 1 - \alpha$. Conversely, conditioned on $1 \leq N_c < N/2$, assume that*

$$\beta \leq \mathcal{O}\left(\Delta\left(1 - \frac{2N_c}{N}\right)\right),$$

*where the constants only depend on $Q_1$ and $Q_2$. Then, we have $\mathrm{FN} \geq 1/2$ conditionally.*

*Proof of Theorem 6.* According to Assumption 2 and based on the definition of log-sum-exp (LSE), we have

$$\mathsf{LSE}_\beta(\sigma \circ f_{\boldsymbol{\theta}}(\boldsymbol{x}_{1:N})) = \frac{1}{\beta} \log\left(\frac{1}{N} \sum_{i=1}^N \exp\left(\beta \, \sigma \circ f_{\boldsymbol{\theta}}(\boldsymbol{x}_i)\right)\right)$$

$$= \frac{1}{\beta} \log\left(\frac{N_c}{N} e^{\beta \Gamma_c} + \frac{N - N_c}{N} e^{\beta \Gamma_n}\right), \tag{20}$$

where

$$\Gamma_c \overset{\mathrm{d}}{=} \frac{1}{\beta} \log \left( \frac{1}{N_c} \sum_{i \in \mathcal{C}} e^{\beta q_i} \right), \quad \Gamma_n \overset{\mathrm{d}}{=} \frac{1}{\beta} \log \left( \frac{1}{N - N_c} \sum_{i \in \mathcal{N}} e^{\beta q_i'} \right), \tag{21}$$

with $q_1, \ldots, q_{N_c} \overset{i.i.d.}{\sim} Q_c$ and $q_1', \ldots, q_{N-N_c}' \overset{i.i.d.}{\sim} Q_n$. Moreover,

$$\Gamma_c = \mu_c + \frac{1}{\beta} \log \left( \frac{1}{N_c} \sum_{i=1}^{N_c} e^{\beta(q_i - \mu_c)} \right) \triangleq \mu_c + \Delta q_c,$$

$$\Gamma_n = \mu_n + \frac{1}{\beta} \log \left( \frac{1}{N - N_c} \sum_{i=1}^{N-N_c} e^{\beta(q_i' - \mu_n)} \right) \triangleq \mu_n + \Delta q_n, \tag{22}$$

where $\Delta q_c$ and $\Delta q_n$ are the $\beta$-LSE of $N_c$ and $N - N_c$ zero-mean sub-Gaussian random variables, respectively. It is known that as $\beta$ ranges from 0 to $+\infty$, the expected values $\mathbb{E}[\Delta q_c]$, $\mathbb{E}[\Delta q_n]$ grow from 0 to $\mathcal{O}(\sqrt{V} \log N_c)$ and $\mathcal{O}(\sqrt{V} \log(N - N_c))$, respectively [96]. Additionally, their variances vanish at the rate $\{N_c, N - N_c\}^{-\gamma}$ for some $\gamma \in (1/2, 1)$ depending on the tail behavior of $Q_c$ and $Q_n$. Therefore,

$$\mathbb{P}\left( \Delta q_c \le C_\alpha \sqrt{V} (\log N_c + N_c^{-\gamma}), \quad \Delta q_n \le C_\alpha \sqrt{V} (\log(N - N_c) + (N - N_c)^{-\gamma}) \right) \ge 1 - \alpha/2,$$

and

$$\mathbb{P}\left( \Delta q_c \ge -C_\alpha \sqrt{V} N_c^{-\gamma}, \quad \Delta q_n \ge -C_\alpha \sqrt{V} (N - N_c)^{-\gamma} \right) \ge 1 - \alpha/2,$$

for all $\alpha \in (0, 1)$, where $C_\alpha$ has polylogarithmic dependence on $\alpha^{-1}$.

**Guarantee on TP rate:** With probability at least $1 - \alpha$ given $N_c$, we have

$$\mathsf{LSE}_\beta(\sigma \circ f_{\boldsymbol{\theta}}(\boldsymbol{x}_{1:N})) = \mu_n + \Delta - \frac{\log(N/N_c)}{\beta} + \frac{1}{\beta} \log \left( e^{\beta \Delta q_c} + \left( \frac{N - N_c}{N_c} \right) e^{-\beta \Delta} e^{\beta \Delta q_n} \right)$$

$$\ge \mu_n + \Delta - \frac{\log(N/N_c)}{\beta} + \Delta q_c$$

$$\ge \mu_n + \Delta - \frac{\log(N/N_c)}{\beta} - C_\alpha \sqrt{V} N_c^{-\gamma}. \tag{23}$$

Thus, to ensure $\mathsf{LSE}_\beta(\sigma \circ f_{\boldsymbol{\theta}}(\boldsymbol{x}_{1:N})) \ge T$, it suffices that

$$\beta \ge \frac{\log(N/N_c)}{\frac{\Delta}{2} - C_\alpha \sqrt{V} N_c^{-\gamma}}.$$

Taking expectation over $N_c$, it suffices that

$$\beta \ge \frac{2C_\alpha' \, \mathbb{E}\left[ \log(N/N_c) \mid N_c \ge 1 \right]}{\Delta - 2C_\alpha \sqrt{V} N_c^{-\gamma}},$$

which implies $\mathrm{TP} \ge 1 - \alpha$. This proves the first part.

**Guarantee on FN rate:** Next, we show that small values of $\beta$ can sharply degrade the TP rate, leading to an increase in FN rate. When $\beta$ is small, the distributions of $\Delta q_c$ and $\Delta q_n$ are nearly symmetric. This directly implies that $\beta$-LSE value is symmetrically distributed around

$$\frac{1}{\beta} \log \left( \frac{N_c}{N} e^{\beta \mu_c} + \frac{N - N_c}{N} e^{\beta \mu_n} \right).$$

Therefore, conditioned on $N_c \ge 1$, we consider the case where

$$\frac{1}{\beta} \log \left( \frac{N_c}{N} e^{\beta \mu_c} + \frac{N - N_c}{N} e^{\beta \mu_n} \right) \le \frac{\mu_c + \mu_n}{2},$$

which implies FN $\geq 1/2$. Using Taylor expansion for small $\beta$:

$$\frac{\mu_c + \mu_n}{2} \geq \frac{1}{\beta} \log \left( \frac{N_c}{N} e^{\beta \mu_c} + \frac{N - N_c}{N} e^{\beta \mu_n} \right)$$

$$= \frac{N_c}{N} \mu_c + \frac{N_c}{2N} \beta \mu_c^2 + \frac{N - N_c}{N} \mu_n + \frac{N - N_c}{2N} \beta \mu_n^2 + \mathcal{O}(\beta^2). \tag{24}$$

Rearranging yields the sufficient condition:

$$\beta \leq \frac{\Delta \left( 1 - \frac{2 N_c}{N} \right)}{\mu_n^2 + \frac{N_c}{N} (\Delta^2 + 2 \mu_n \Delta)},$$

which guarantees a conditional FN rate of at least $1/2$. This completes the proof. $\qquad\square$

Theorem 6 shows that selecting a sufficiently large value of $\beta$ ensures an arbitrarily high true positive (TP) rate. Specifically, setting $\beta \gg \mathcal{O}(\frac{1}{\Delta} \log N)$—with constants depending only polylogarithmically on the target confidence level—is sufficient to achieve any desired TP rate. Conversely, in the presence of noise, choosing a small $\beta$ leads to a substantial false negative (FN) rate, significantly degrading performance. Therefore, the theorem recommends using large values of $\beta$ to ensure robust sensitivity.

## B  Algorithms

### B.1  Maxsoft

To avoid using complex notation, we explain our algorithm on a binary classification problem. With simple modifications, our method can be extended to other MIL problems such as multiclass classification or regression.

As defined, $\mathsf{LSE}_\beta$ is an aggregation function in form of

$$\mathsf{LSE}_\beta(q_1, \ldots, q_N) \triangleq \frac{1}{\beta} \log \left( \frac{1}{N} \sum_{i=1}^{N} e^{\beta q_i} \right), \tag{25}$$

represents a scalar instance value in the range $[0, 1]$. For simplicity, we denote $\mathsf{LSE}_\beta(q_{1:N})$ as $\mathsf{LSE}_\beta(q_1, \ldots, q_N)$. We can see that

$$\frac{\partial \mathsf{LSE}_\beta}{\partial q_i}(q_{1:N}) = \frac{e^{\beta q_i}}{\sum_{j=1}^{N} e^{\beta q_j}}.$$

which is simply the Softmax function when applied to $q_1, \ldots, q_N$.

In multiple instance learning of WSI we aim to train a model to reduce $\mathsf{L}_{\mathrm{LSE}}(\boldsymbol{\theta})$ with respect to $\boldsymbol{\theta}$ (see Definition 3). Let us introduce likelihood probability with

$$p_j = P(y_j = 1 | I_j) = P(y_j = 1 | q_1^{(j)}, \ldots, q_N^{(j)}), \tag{26}$$

rephrasing $\mathsf{L}_{\mathrm{LSE}}(\boldsymbol{\theta})$ as $\frac{1}{n} \sum_{j=1}^{n} \mathcal{L}\left(y_j \| p_j\right)$. We specifically use the cross entropy loss function for $\mathcal{L}$.

In Maxsoft, we perform the forward pass with $p_j = \max_i(q_1^{(j)}, \ldots q_N^{(j)})$. Afterward, in the backward step, we approximate $\mathsf{L}_{\mathrm{LSE}}(\theta)$ by replacing the $\max$ aggregation operator with $\mathsf{LSE}_\beta$. More precisely, we approximate $\nabla_\theta \mathsf{L}_{\mathrm{LSE}}$ as:

$$\nabla_\theta \mathsf{L}_{\mathrm{LSE}} \approx \frac{1}{n} \sum_{j=1}^{n} \frac{\partial \mathcal{L}}{\partial p_j} \left( \max(q_{1:N}^j) \right) \cdot \mathsf{Softmax}(q_{1:N}^j) \cdot J_{q_{1:N}^j}(\theta). \tag{27}$$

where $J$ is the Jacobian matrix. Notice that in case we calculated the original gradients, the Softmax term would be replaced with the max gradient (which due to non-differentiability has instable behavior) and in case we initially used LSE in the forward pass, $\frac{\partial \mathcal{L}}{\partial p_j} \left( \max \left( q_{1:N}^{(j)} \right) \right)$ would be replaced with $\frac{\partial L}{\partial p_j} \left( \mathsf{LSE}_\beta \left( q_{1:N}^{(j)} \right) \right)$.

**Algorithm 1** Multiple Instance Learning of WSI with Maxsoft

---

1: **Input:** WSI Dataset $D = \left\{ (I_1, y_1), \ldots, (I_n, y_n) \right\}$
2: **Input:** $\beta$: Hyperparameter for $\mathsf{LSE}_\beta$
3: **Input:** ViT: Frozen Vision Transformer feature extractor
4: **Input:** $f_\theta$: Trainable MIL classification head
5: **Input:** $\alpha$: Step size (learning rate)
6: **Output:** $f_{\theta^*}$: Trained model

7: **for** each training epoch **do**
8:     **for** $j = 1$ to $n$ **do**
9:         Extract patches $\{\boldsymbol{X}_1^{(j)}, \ldots, \boldsymbol{X}_N^{(j)}\}$ from image $I_j$
10:        Obtain features $\{\boldsymbol{x}_i^{(j)} = \mathsf{ViT}(\boldsymbol{X}_i^{(j)})\}_{i=1}^N$
11:        Compute instance-level scores $q_i^{(j)} = \sigma(f_\theta(\boldsymbol{x}_i^{(j)}))$ for $i = 1, \ldots, N$
12:        Compute bag-level prediction: $p_j = \max(q_{1:N}^{(j)})$ (which is also test time prediction)
13:        Compute loss $\mathcal{L}(y_j \| p_j)$
14:     **end for**
15:     Compute gradient approximation:

$$\nabla_\theta \mathsf{L}_{\mathrm{LSE}} \approx \frac{1}{n} \sum_{j=1}^n \frac{\partial \mathcal{L}}{\partial p_j} \left( Y_j \| \max(q_{1:N}^{(j)}) \right) \cdot \mathsf{Softmax}(q_{1:N}^{(j)}) \cdot J_{q_{1:N}^{(j)}}(\theta)$$

16:     Update model parameters: $\theta \leftarrow \theta - \alpha \cdot \nabla_\theta \mathsf{L}_{\mathrm{LSE}}$
17: **end for**
18: Return trained model: $\theta^* \leftarrow \theta$
19: **return** $f_{\theta^*}$

---

**Algorithm 2** PerSlide Augmentation for WSI Tasks

---

1: **Input:** WSI Dataset $D = \left\{ (I_1, y_1), \ldots, (I_n, y_n) \right\}$, Augmentation functions $\mathcal{T} = \{\tau_1, \ldots, \tau_m\}$, Epochs $E$, Number of instances per data point $N$
2: **Output:** Trained model

3: **for** epoch $t = 1$ to $E$ **do**
4:     Initialize augmented dataset $D_t^{\mathrm{aug}} \leftarrow \emptyset$
5:     **for** each $I_i \in D$ **do**
6:         Sample augmentation $\tau \sim \mathrm{Uniform}(\mathcal{T})$
7:         Apply $\tau$ to the whole image $I_i$ to get $\widehat{I}_i$
8:         Add $(\widehat{I}_i, y_i)$ to $D_t^{\mathrm{aug}}$
9:     **end for**
10:     $\mathtt{TrainOneEpoch}(\, D_t^{\mathrm{aug}}, t \,)$
11: **end for**

---

---

**Algorithm 3** PerPatch Augmentation

---

1: **Input:** WSI Dataset $D = \{(I_1, y_1), \ldots, (I_n, y_n)\}$, Augmentation functions $\mathcal{T} = \{\tau_1, \ldots, \tau_m\}$, Epochs $E$, Number of instances per data point $N$, ViT Frozen Vision Transformer feature extractor

2: **Output:** Trained model

3: **Precompute Stage:**
4: **for** each image $I_i \in D$ **do**
5:     Set first augmented image variant as the original $I_i^{(0)} \leftarrow I_i$
6:     **for** each augmentation $\tau_k \in \mathcal{T}$, where $k = 1, \ldots, m$ **do**
7:         Initialize augmented image variant $I_i^{(k)} \leftarrow \emptyset$
8:         **for** each patch $X_j \in I_i$ where $j = 1, 2, \ldots, N$ **do**
9:             Apply augmentation: $X_j^{(k)} = \tau_k(X_j^i)$
10:            Compute embedding for further optimization: $x_j^{(k)} = \mathsf{ViT}(X_j^{(k)})$
11:            Add $x_j^{(k)}$ to $I_i^{(k)}$
12:         **end for**
13:     **end for**
14: **end for**

15: **Training Stage:**
16: **for** epoch $t = 1$ to $E$ **do**
17:     Initialize augmented dataset $D_t^{\mathrm{aug}} \leftarrow \emptyset$
18:     **for** each image $I_i \in D$ **do**
19:         Initialize empty set $\widehat{I}_i \leftarrow \emptyset$
20:         **for** each patch $X_j \in I_i$ where $j = 1, 2, \ldots, N$ **do**
21:            Sample augmentation index $k \sim \mathrm{Uniform}(\{0, \ldots, m\})$
22:            Retrieve embedding $x_j^{(k)}$ from $I_i^{(k)}$
23:            Add $x_j^{(k)}$ to $\widehat{I}_i$
24:         **end for**
25:         Add $(\hat{I}_i, y_i)$ to $D_t^{\mathrm{aug}}$
26:     **end for**
27:     `TrainOneEpoch(` $D_t^{\mathrm{aug}}, t$ `)`
28: **end for**

---

For cases with multiple classes we simply modify MLP's linear head to have multiple heads for each class (or in the worst case for architectures which differ from MLP, multiple classification head and Maxsoft per class), extending equation 26 for cases with $y = k$. To further facilitate the usage of our Maxsoft aggregation function, we provide a simple Python implementation of it in the paper. It can be easily integrated into any code like other Pytorch [97] layers.

## C  PyTorch Implementation of Maxsoft Pooling

Listing 1 presents the PyTorch-style implementation of Maxsoft pooling.

Listing 1: Pytorch-style implementation for Maxsoft

```python
class MaxSoftmaxSTE(torch.autograd.Function):
    @staticmethod
    def forward(ctx, input, beta):
        # Save the input tensor for backward pass
        ctx.save_for_backward(input)
        ctx.beta = beta

        # Perform the forward pass (select maximum value)
        max_val, _ = torch.max(input, dim=0)

        return max_val

    @staticmethod
    def backward(ctx, grad_output):
        # Retrieve the saved input tensor
        input, = ctx.saved_tensors
        beta = ctx.beta

        # Compute the Softmax over the input for gradient
        Softmax_grad = torch.Softmax(input * beta, dim=0)

        # Multiply the incoming gradient (grad_output) by
        # the Softmax weights
        grad = grad_output * Softmax_grad

        return grad, None
```

## D  PerSlide vs. PerPatch Figure

Figure 5 contains a visual comparison of the difference between PerSlide and PerPatch augmentations.

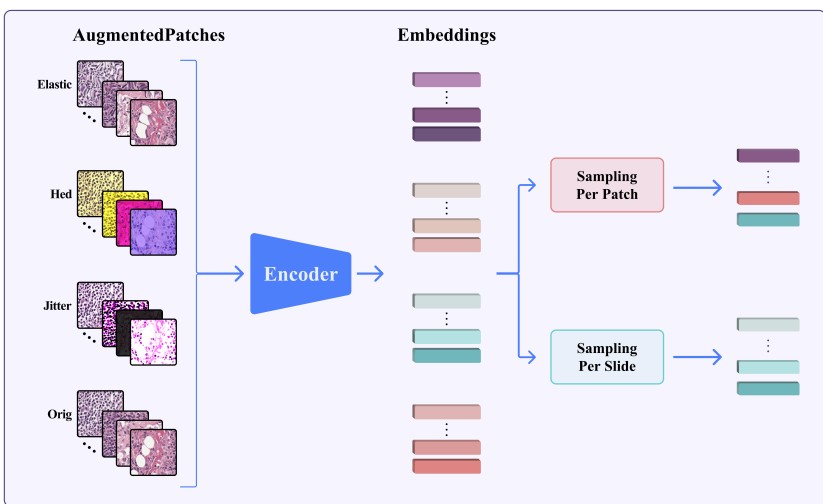

Figure 5: Comparison of PerSlide and PerPatch augmentations: PerSlide applies a single augmentation uniformly across all patches in a WSI, whereas PerPatch independently selects an augmentation for each patch from multiple variants, resulting in substantially higher diversity during MIL pooling training.

# E Datasets

**CAMELYON16 and CAMELYON17** are large-scale WSI datasets introduced for benchmarking automated methods in detecting metastatic breast cancer in lymph nodes. Developed for challenging settings, they serve as standards for evaluating tumor detection algorithms in histopathology. Automating this task can reduce pathologist workload, improve diagnostic consistency, and mitigate subjectivity [78, 35]. CAMELYON16 includes two classes—normal and tumor—while CAMELYON17 includes four: normal, macro, micro, and ITC. The ITC class (isolated tumor cells) is especially challenging due to its sparse, small tumor clusters (<0.2 mm or <200 cells). Both datasets are difficult because tumor regions occupy only a small area in positive WSIs. CAMELYON16 provides an official labeled test split and pixel-level annotations for all WSIs, whereas CAMELYON17 lacks test labels and includes pixel-level annotations for only 100 of its 500 training WSIs.

**TCGA-Lung** is a subset of The Cancer Genome Atlas (TCGA) comprising WSIs from two lung cancer subtypes: Lung Adenocarcinoma (LUAD) and Lung Squamous Cell Carcinoma (LUSC). After filtering out low-quality slides, the dataset includes 1,042 WSIs—530 LUAD and 512 LUSC. A key characteristic is that tumor regions occupy the majority of each slide. Additionally, most patients contribute multiple WSIs [79]. Pixel-level annotations are not available.

**SICAP-MIL** is a publicly available dataset designed to benchmark MIL-based approaches for prostate cancer grading in WSIs. It includes biopsy slides from 271 patients, scanned at $40\times$ and tiled into overlapping $512 \times 512$ patches at $10\times$ resolution. Each slide is globally labeled by expert pathologists with primary and secondary Gleason grades, reflecting the dominant tumor patterns. The dataset also introduces proportional constraints that represent the relative occurrence of each grade, supporting the development of constrained MIL methods that can rival fully supervised models. Slides are labeled as normal or abnormal and further annotated with Gleason grades (GG3, GG4, GG5). In abnormal WSIs, tumor-associated regions are present in a significant portion, though not the majority of the slide. Exact pixel-level annotations are not provided [80].

# F Additional Experimental Setup

**CAMELYON16 and CAMELYON17:** WSIs are tiled into $256 \times 256$ non-overlapping patches at $20\times$ magnification, with background regions excluded following [67]. For CAMELYON16, we use the official test split. As CAMELYON17 lacks an official labeled test set and includes one low-quality slide, we discard that slide and randomly split the remaining data into approximately $60\%$ training, $15\%$ validation, and $25\%$ testing, using the balanced splitting protocol from [78, 35, 98]. Specifically, we ensure each split has a roughly equal number of Normal, Macro, Micro, and ITC samples.

For CAMELYON16, we train on 50 WSIs and evaluate on the full test set. For CAMELYON17, we use the 99 high-quality annotated WSIs, as region-level annotation is clinically most relevant. This limited-WSI setup emulates extreme low-data regimes typical of rare cancers ($\approx$70 WSIs/type [99]). The CAMELYON16 patching yields $\approx$1.5M patches, averaging $\approx$7,900 per WSI. For CAMELYON17, we obtain $\approx$800k patches, about 8,000 per WSI on average. Patches overlapping annotated tumor regions are labeled tumor; all others are labeled normal. Each model is trained from scratch five times, and we report means and standard deviations for all metrics. We binarize CAMELYON17 following [14], treating ITC-labeled WSIs as tumorous. This is the most challenging setting, as some ITC slides contain only one or two tumor patches. For experiments using the complete versions of these two datasets, see Appendix P.

**TCGA-Lung:** WSIs are tiled into non-overlapping $256\times256$ patches at $20\times$ magnification, excluding background regions. This results in approximately 3.5 million patches, averaging around 3,500 patches per WSI. The dataset is roughly split into training ($60\%$), validation ($15\%$), and test ($25\%$) sets and enforce patient-level grouping, whereby all slides from a patient are assigned to a single split (no cross-split leakage) [79].

**SICAP-MIL:** Each $512 \times 512$ patch is divided into four $256 \times 256$ patches, yielding approximately 34,000 patches in total, with each WSI containing around 100 patches on average. Each model is trained from scratch five times independently, and we report the mean and standard deviation for all performance metrics [80].

# G  Calibration Metric ECE

Let $B_m$ denote the set of indices for samples with prediction confidence in the interval $I_m = \left(\frac{m-1}{M}, \frac{m}{M}\right]$. The accuracy within bin $B_m$ is defined as:

$$\text{acc}(B_m) = \frac{1}{|B_m|} \sum_{i \in B_m} 1(\hat{y}_i = y_i), \tag{28}$$

where $\hat{y}_i$ is the predicted label and $y_i$ is the ground-truth label for sample $i$. The average confidence in bin $B_m$ is:

$$\text{conf}(B_m) = \frac{1}{|B_m|} \sum_{i \in B_m} \hat{p}_i, \tag{29}$$

where $\hat{p}_i$ denotes the predicted confidence for sample $i$.

The Expected Calibration Error (ECE) is computed as:

$$ECE = \sum_{m=1}^{M} \frac{|B_m|}{n} \left| \text{acc}(B_m) - \text{conf}(B_m) \right|, \tag{30}$$

where $n$ is the number of samples.

A perfectly calibrated model satisfies $\text{acc}(B_m) = \text{conf}(B_m)$ for all $m$, resulting in an ECE of 0 [100]. For instance, both $\hat{p}_i = 1$ with $\hat{y}_i = y_i$ and $\hat{p}_i = 0$ with $\hat{y}_i \neq y_i$ contribute to lower ECE [15].

# H  Implementation Details

## H.1  MIL Models

For DINO Domain [64] on CAMELYON16, CAMELYON17, and TCGA-Lung [78, 35, 79], we train a ViT-S/16 from scratch on all patches from training WSIs using the default hyperparameters from the official DINO repository [64]. For SICAP-MIL [80], we apply the same default settings for ViT-S/16.

For all MIL methods, including Maxsoft pooling, we tune learning rate, weight decay, and weight initialization using the validation set. The best configuration is selected based on validation performance. No early stopping is applied. All models are trained for 500 epochs using the AdamW optimizer [101] with default parameters unless otherwise specified.

**CAMELYON16.** DINO Natural (LR 0.002, WD 0.05, Xavier-uniform); DINO Domain (LR 0.1, WD 0.05, truncated-normal init); UNI (LR 0.02, WD 0.05, Xavier-uniform); Prov-GigaPath (LR 0.02, WD 0.05, Xavier-uniform).

**CAMELYON17.** DINO Natural (LR 0.02, WD 0.005, Xavier-uniform); DINO Domain (LR 0.1, WD 0.05, truncated-normal); UNI (LR 0.02, WD 0.005, Xavier-uniform); Prov-GigaPath (LR 0.02, WD 0.05, Xavier-uniform).

**TCGA-Lung.** DINO Natural (LR 0.02, WD 0.05, Xavier-uniform); DINO Domain (LR 0.002, WD 0.005, truncated-normal); UNI (LR 0.002, WD 0.05, Xavier-uniform); Prov-GigaPath (LR 0.1, WD 0.05, orthogonal).

**SICAP-MIL.** DINO Natural (LR 0.02, WD 0.05, orthogonal); DINO Domain (LR 0.002, WD 0.005, Xavier-uniform); UNI (LR 0.002, WD 0.05, truncated-normal); Prov-GigaPath (LR 0.1, WD 0.05, truncated-normal).

## H.2  Augmentations

Base PerPatch augmentations comprise Random Rotation, Random Gaussian Blur, and Random Color Jitter, selected per our analyses in Appendix J and Table 4; hyperparameters are dataset-specific: CAMELYON16—LR 0.1, WD 0.05, Xavier-uniform; CAMELYON17—LR 0.1, WD 0.05,

orthogonal; TCGA-Lung—LR 0.1, WD 0.05, orthogonal; SICAP-MIL—LR 0.1, WD 0.05, truncated-normal. All MIL models are trained with bag-level labels only. Experiments use PyTorch 2.1 and scikit-learn on an RTX 4090 [97].

## H.3   Augmentations and Architectures

We further extend our range of experiments to assess the effect of different augmentation methods on various MIL pooling architectures on the CAMELYON17 and SICAP-MIL datasets [80]. As shown in Table 6, our PerPatch augmentation method demonstrates AUC improvements in most setups. Among the previous methods, AugDiff [62] shows performance stability across multiple architectures, whereas MixUp methods [55, 56] exhibit poor performance in some settings.

Table 6: The effect of various augmentation methods across MIL architectures on the Camelyon17 and SICAP datasets.

| | CAMELYON17 | | | | | | | | | | | | | |
|---|---|---|---|---|---|---|---|---|---|---|---|---|---|---|
| Augmentation | max pooling | | mean pooling | | ABMIL | | DSMIL | | Snuffy | | LSE pooling | | Maxsoft pooling | |
| | AUC | ECE | AUC | ECE | AUC | ECE | AUC | ECE | AUC | ECE | AUC | ECE | AUC | ECE |
| ReMix (MixUp) | $0.743_{385}$ | $0.245_{169}$ | $0.823_{006}$ | $0.118_{030}$ | $0.833_{038}$ | $0.193_{038}$ | $0.723_{324}$ | $0.162_{026}$ | $0.688_{176}$ | $0.179_{044}$ | $0.777_{186}$ | $0.212_{046}$ | $0.840_{020}$ | $0.242_{017}$ |
| RankMix (MixUp) | $0.812_{253}$ | $0.144_{157}$ | $0.842_{004}$ | $0.107_{026}$ | $0.823_{045}$ | $0.192_{048}$ | $0.764_{123}$ | $0.092_{018}$ | $0.751_{182}$ | $0.182_{034}$ | $0.833_{170}$ | $0.202_{040}$ | $0.855_{191}$ | $0.197_{156}$ |
| AugDiff | $0.842_{180}$ | $0.163_{093}$ | $0.863_{005}$ | $0.147_{053}$ | $0.855_{035}$ | $0.189_{033}$ | $0.801_{102}$ | $\mathbf{0.067}_{028}$ | $0.886_{041}$ | $0.189_{004}$ | $0.882_{069}$ | $0.222_{058}$ | $0.894_{069}$ | $0.161_{110}$ |
| SSRDL | $0.833_{040}$ | $0.289_{095}$ | $0.767_{040}$ | $0.239_{027}$ | $0.793_{015}$ | $0.263_{015}$ | $0.807_{135}$ | $0.085_{003}$ | $0.818_{028}$ | $0.176_{002}$ | $0.817_{015}$ | $0.237_{048}$ | $0.859_{040}$ | $0.061_{030}$ |
| PerPatch | $\mathbf{0.923}_{076}$ | $0.131_{021}$ | $0.870_{001}$ | $0.126_{000}$ | $\mathbf{0.893}_{110}$ | $0.183_{055}$ | $\mathbf{0.920}_{107}$ | $0.250_{013}$ | $\mathbf{0.980}_{014}$ | $0.174_{024}$ | $\mathbf{0.965}_{024}$ | $0.232_{012}$ | $\mathbf{0.980}_{014}$ | $\mathbf{0.052}_{013}$ |

| | SICAP-MIL | | | | | | | | | | | | | |
|---|---|---|---|---|---|---|---|---|---|---|---|---|---|---|
| Augmentation | max pooling | | mean pooling | | ABMIL | | DSMIL | | Snuffy | | LSE pooling | | Maxsoft pooling | |
| | AUC | ECE | AUC | ECE | AUC | ECE | AUC | ECE | AUC | ECE | AUC | ECE | AUC | ECE |
| ReMix (MixUp) | $0.850_{005}$ | $0.162_{004}$ | $0.787_{002}$ | $0.047_{002}$ | $0.648_{001}$ | $0.388_{001}$ | $0.852_{005}$ | $0.119_{012}$ | $0.845_{003}$ | $0.221_{028}$ | $0.855_{003}$ | $0.201_{006}$ | $\mathbf{0.849}_{008}$ | $0.180_{017}$ |
| RankMix (MixUp) | $0.846_{013}$ | $0.157_{009}$ | $0.808_{002}$ | $0.042_{002}$ | $0.683_{001}$ | $0.381_{001}$ | $0.849_{004}$ | $0.133_{022}$ | $0.855_{002}$ | $0.193_{011}$ | $0.836_{002}$ | $0.220_{022}$ | $0.845_{008}$ | $0.280_{017}$ |
| AugDiff | $0.861_{029}$ | $0.155_{005}$ | $0.809_{008}$ | $\mathbf{0.041}_{003}$ | $0.724_{005}$ | $0.260_{009}$ | $0.856_{005}$ | $0.129_{008}$ | $0.860_{007}$ | $0.191_{004}$ | $0.863_{002}$ | $0.188_{021}$ | $0.818_{019}$ | $0.180_{005}$ |
| SSRDL | $0.806_{008}$ | $0.160_{022}$ | $0.803_{007}$ | $0.224_{023}$ | $0.657_{002}$ | $0.406_{004}$ | $0.840_{004}$ | $0.125_{012}$ | $0.806_{019}$ | $0.259_{008}$ | $0.812_{005}$ | $0.182_{005}$ | $0.838_{009}$ | $\mathbf{0.161}_{005}$ |
| PerPatch | $\mathbf{0.870}_{001}$ | $0.155_{003}$ | $\mathbf{0.810}_{001}$ | $0.142_{002}$ | $\mathbf{0.831}_{010}$ | $0.219_{021}$ | $0.862_{006}$ | $\mathbf{0.118}_{008}$ | $\mathbf{0.866}_{003}$ | $0.186_{003}$ | $\mathbf{0.869}_{000}$ | $0.176_{002}$ | $0.827_{002}$ | $0.223_{018}$ |

# I   Augmentations Descriptions and Samples

Figure 6 presents examples of four CAMELYON17 [35] patches augmented with Random Rotation, Random Elastic Deformation, Random Affine Transformation, Random Gaussian Blurring, Random Color Jitter, and Random Hematoxylin-Eosin-DAB (HED) Jitter [59, 62]. The augmentations are defined as follows: **Random Rotation**: Rotates the image by a random angle within a predefined range. **Random Elastic Deformation**: Applies spatially varying smooth deformations to simulate elastic distortions. **Random Affine Transformation**: Combines translation, rotation, scaling, and shearing. **Random Gaussian Blurring**: Convolves the image with a Gaussian kernel to reduce high-frequency noise. **Random Color Jitter**: Randomly modifies brightness, contrast, saturation, and hue. **Random HED Jitter**: Perturbs the HED color space representation to simulate staining variations [59, 62].

# J   Augmentation Quality and Complexity

In this section, we evaluate the effectiveness of Base augmentations using four key metrics: FID, Density, and Coverage.

**FID (Fréchet Inception Distance)**: FID is a widely used metric that quantitatively compares the distribution of generated images against real images in a deep feature space. In the context of image augmentation, FID helps assess how well the augmented images capture the underlying data distribution of the real dataset. A lower FID score indicates that the augmented images are closer in distribution to the original images, which implies that the generative model produces realistic and coherent augmentations. A good FID score suggests that the additional images maintain the essential visual characteristics of the real-world data, thereby potentially improving the downstream performance of learning algorithms [102].

**Density**: The density metric measures how densely the generated images occupy the feature space relative to the real images. In image augmentation, high density implies that the synthetic images generated by the model are not only realistic but also well-aligned with the clusters of real images in the feature space. This alignment is critical, as it suggests that the augmented images reinforce the intrinsic patterns found in the data rather than creating spurious or outlier representations. Evaluating density helps to understand whether the augmentation process is introducing variations that are

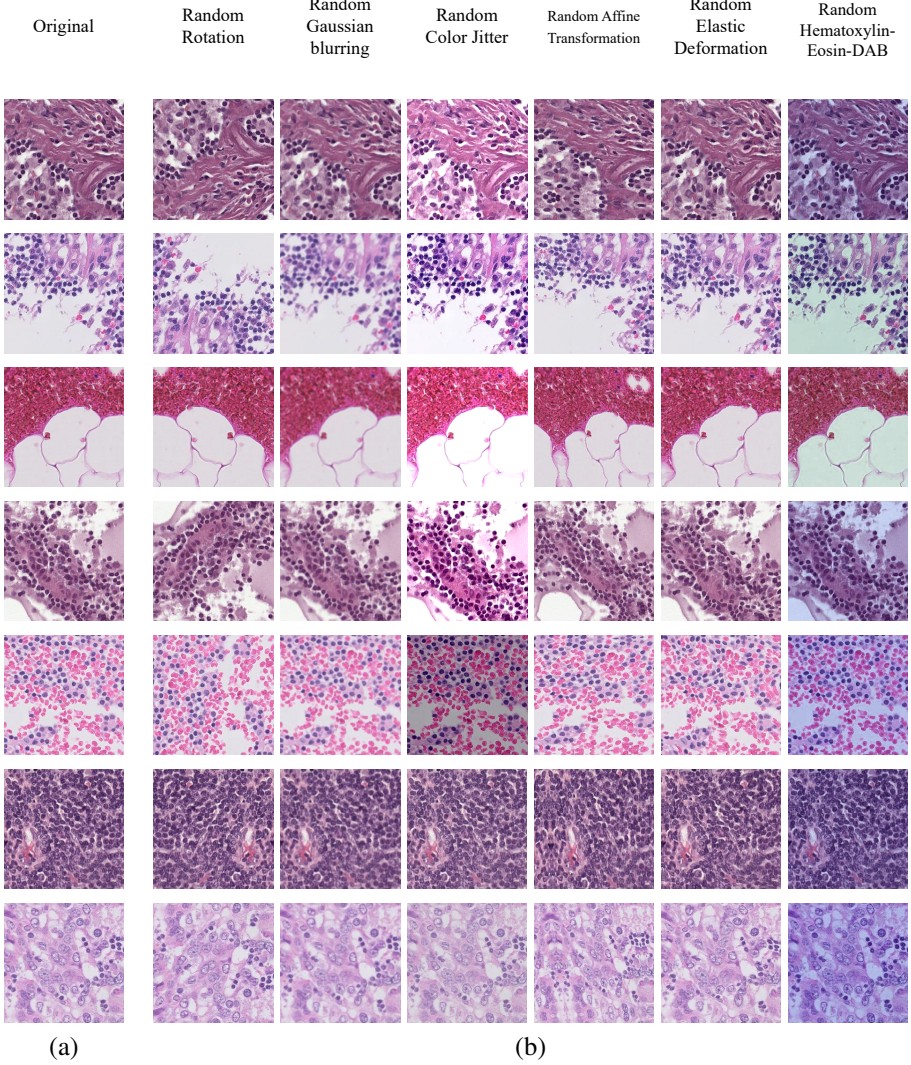

| Original | Random Rotation | Random Gaussian blurring | Random Color Jitter | Random Affine Transformation | Random Elastic Deformation | Random Hematoxylin-Eosin-DAB |

(a)                  (b)

Figure 6: Augmentation schemes (a) sample CAMELYON17 patches (b) augmented versions of the same patches.

plausible and beneficial for training robust models, ensuring that the augmentation enriches the dataset with high-quality, representative samples [103].

**Coverage**: Coverage evaluates the diversity of the generated images by determining the proportion of the real image distribution that is covered by the synthetic samples. In the realm of image augmentation, high coverage indicates that the method is capable of producing a wide range of variations that collectively span the real data manifold. This is particularly important when the goal is to enhance a dataset by introducing new variations that help prevent overfitting and improve generalization in downstream tasks. A model with good coverage ensures that the augmented dataset is not biased toward a narrow subset of the data distribution, thereby providing a more comprehensive training set that captures the full spectrum of variability present in real-world images [103].

Table 7 shows that Random Hematoxylin–Eosin–DAB (HED) Jitter exhibits the lowest Density and Coverage and the highest FID. Despite its pathology-specific design, this augmentation neither reduces slide-level AUC variance nor improves ECE, indicating that the transformations it introduces do not reflect realistic imaging variability. In general, augmentations with higher Density and Coverage and lower FID perform better, as illustrated by the strong results of Random Rotation, Random Gaussian Blur, and Random Color Jitter, and the poor performance of Random Elastic

Deformation and Random Affine Transformation (see Table 5). We attribute the latter two failures to their limited relevance to real-world slide variations. Collectively, these findings indicate that the most effective strategy remains PerPatch combined with Random Rotation, Random Gaussian Blur, and Random Color Jitter.

Table 7: Quality Metrics for different augmentations on the CAMELYON17 dataset. The results are reported in the form of $mean_{.std}$.

| Augmentation | FID | Density | Coverage |
|---|---|---|---|
| Random Rotation | $73.801_{46.799}$ | $0.785_{0.066}$ | $0.999_{0.001}$ |
| Random Elastic Deformation | $274.585_{139.951}$ | $0.353_{0.162}$ | $0.636_{0.185}$ |
| Random Affine Transformation | $135.422_{70.692}$ | $0.434_{0.124}$ | $0.858_{0.108}$ |
| Random Gaussian Blurring | $6.842_{1.197}$ | $0.986_{0.030}$ | $0.999_{0.000}$ |
| Random Color Jitter | $30.085_{7.697}$ | $0.672_{0.053}$ | $0.997_{0.004}$ |
| Random Hematoxylin-Eosin-DAB (HED) Jitter | $2395.841_{286.359}$ | $0.000_{0.001}$ | $0.001_{0.001}$ |

# K UMAP of Patch Embeddings Based on Encoder

Figure 7 presents UMAP visualizations of patch representations obtained from DINO Natural, DINO Domain, UNI, and Prov-GigaPath on four test samples from the CAMELYON17 dataset. In general, as the strength of the representation encoder increases, the embeddings exhibit improved clustering by both label and patient, reflecting higher feature discriminability and domain alignment.

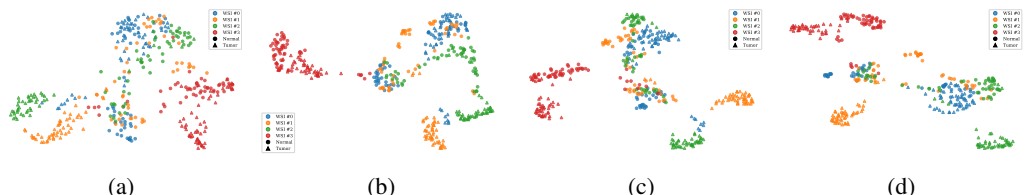

(a)          (b)          (c)          (d)

Figure 7: UMAP of representations. (a) with Dino Natural (b) Dino Domain (c) UNI (d) Prov-GigaPath.

# L Error Analysis

## L.1 Error Analysis on UNI for CAMELYON17

As we can see in Table 1, the results of UNI on the CAMEYON17 dataset are low. This is an exception to our finding that the better the representation encoder (in particular the more data it has been trained on) the better the perofrmance of the MIL pooling. Through an error analysis we found that the problem with UNI comes from the fact that it cannot generalize well in WISs with extremely low tumor regions (mostly ITC and then Micro subgroups). This only happens in CAMELYON17 since only this dataset has such small tumor regions and probably comes from the fact that UNI's autopsy WSIs do not provide such samples. The results can be found at Table 8.

## L.2 Error Analysis for Augmentations and ECE on CAMELYON

When *Per-Patch* augmentation is applied to the sparsely annotated CAMELYON17 WSIs, the network is repeatedly exposed to heavily transformed, minute tumorous regions. This increased morphological diversity strengthens its ability to recognise genuinely positive tissue and, as a result, *raises* accuracy on slides that contain tumour (Table 9). The same shift, however, slightly *erodes* performance on the overwhelmingly abundant tumour-free slides: features that were previously dismissed as benign are now more readily interpreted as malignant, leading to a higher false-positive rate.

From a calibration standpoint, the effect is likewise inverted relative to our original claim. The model becomes better calibrated on the rare, hard positives—confidence scores now align closely with their improved correctness—but it grows over-confident on negatives, which dominate the data distribution

Table 8: Accuracy results for each subgroup in CAMELYON17 on UNI representaton encoder.

| Method | Slide | | | |
|---|---|---|---|---|
| | ITC ACC | Micro ACC | Macro ACC | Negative ACC |
| max pooling | $0.531_{183}$ | $0.604_{433}$ | $0.535_{399}$ | $0.787_{217}$ |
| mean pooling | $0.484_{080}$ | $0.667_{337}$ | $0.514_{168}$ | $0.758_{162}$ |
| ABMIL | $0.417_{358}$ | $0.583_{448}$ | $0.542_{405}$ | $0.681_{350}$ |
| DSMIL | $0.521_{177}$ | $0.604_{422}$ | $0.535_{393}$ | $0.821_{149}$ |
| Maxsoft pooling | $0.713_{184}$ | $0.308_{335}$ | $0.753_{343}$ | $0.816_{129}$ |

yet see a fall in accuracy. Consequently, the overall ECE still increases, though the underlying driver is the miscalibration of negative patches rather than of positives.

In short, *Per-Patch* augmentation sharpens decision boundaries around the scarce tumour class, boosting sensitivity and reducing inter-run variance for positive findings, while sacrificing a portion of specificity on normal tissue. Post-hoc calibration targeted at the negative majority—e.g. temperature scaling on a validation set enriched for benign slides—offers a principled way to retain the newfound robustness to tumour heterogeneity without compromising probabilistic reliability.

Table 9: Accuracy results for each subgroup in CAMELYON17 with DINO Domain representation encoder with and without Augmentation.

| Method | Slide | | | |
|---|---|---|---|---|
| | ITC ACC | Micro ACC | Macro ACC | Negative ACC |
| Maxsoft pooling No Aug | $0.713_{184}$ | $0.308_{335}$ | $0.753_{343}$ | $0.816_{129}$ |
| Maxsoft pooling PerPatch | $1.000_{000}$ | $0.500_{235}$ | $0.500_{235}$ | $0.750_{070}$ |

## M    Additional ROI Detection Images

Figure 8 shows additional examples of patch-level classification on the CAMELYON17 dataset [78, 79]. Consistent with previous results, max pooling and Maxsoft pooling yield the most accurate ROI detections, while mean pooling performs noticeably worse.

## N    Sensitivity to Data Quality

Although our datasets are high quality and processed under stringent protocols, a simple visual inspection can verify their appearance; to test whether the underperformance of Transformer-based models in current data regimes stems from data quality, we reran all CAMELYON17 experiments under varied JPEG compression levels (50; 75—the common setting used in Tables 1; 100—the highest) and with Gaussian blur as a quality corruption. The results in Table 10 replicate prior trends: neither decreasing nor increasing quality alters Transformer behavior, rejecting data quality as the cause of Transformer-based MIL deficiencies in current data regimes.

## O    Experiment on Classical MIL Pooling Functions

We evaluate major classical MIL pooling functions on CAMELYON17 [35] to assess their effectiveness for WSI classification. As shown in Table 11, Maxsoft achieves the best slide- and patch-level metrics (except patch-level on DINO Natural); noisy-or performs poorly [43], ISR excels on patches, and smoothmax lies between LSE and Maxsoft. smoothmax remains without theoretical analysis; intuitively, its weaker performance may arise because its gradient weights instances by deviation from the current smoothmax value rather than Softmax-based importance, potentially hindering credit assignment relative to Maxsoft's direct Softmax path and motivating future theoretical study.

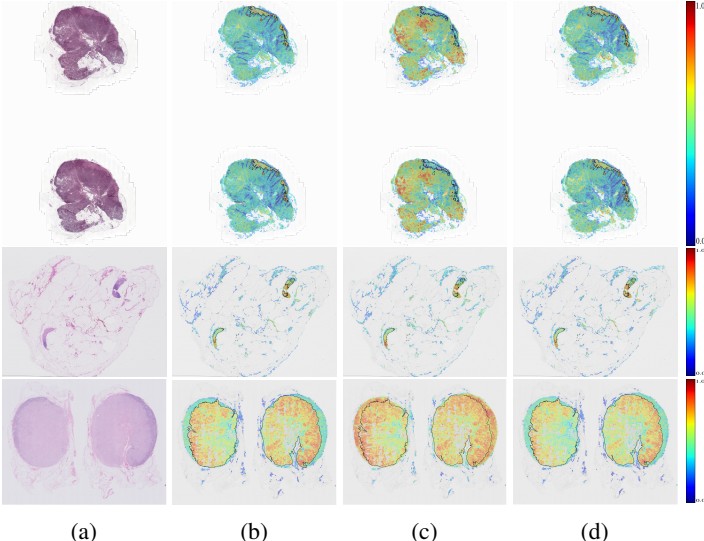

|  (a)  |  (b)  |  (c)  |  (d)  |

Figure 8: Overview of additional ROIs identified on representative WSIs from the CAMELYON17 dataset [35], using (a) max pooling, (b) mean pooling, and (c) Maxsoft pooling.

Table 10: MIL pooling performance under varied JPEG compression levels and Gaussian blur corruptions.

| Method | JPEG 50 | | JPEG 75 | | JPEG 100 | | Gaussian Blur | |
|---|---|---|---|---|---|---|---|---|
| | AUC | ACC | AUC | ACC | AUC | ACC | AUC | ACC |
| max pooling | 0.737 | 0.617 | 0.767 | 0.800 | 0.792 | 0.810 | 0.763 | 0.717 |
| mean pooling | 0.790 | 0.696 | 0.897 | 0.700 | 0.890 | 0.700 | 0.853 | 0.667 |
| ABMIL | 0.697 | 0.600 | 0.750 | 0.683 | 0.805 | 0.750 | 0.737 | 0.650 |
| DSMIL | 0.803 | 0.683 | 0.833 | 0.800 | 0.891 | 0.825 | 0.807 | 0.750 |
| Snuffy | 0.745 | 0.570 | 0.755 | 0.650 | 0.852 | 0.830 | 0.769 | 0.665 |
| LSE pooling | 0.857 | 0.783 | 0.850 | 0.800 | 0.923 | 0.817 | 0.863 | 0.800 |
| Maxsoft pooling | **0.970** | **0.883** | **0.983** | **0.867** | **0.987** | **0.950** | **0.963** | **0.883** |

## P   Full-Data Experiments on CAMELYON

For completeness, we report results from training on the full CAMELYON16 (80% train, 20% validation) and CAMELYON17 (60% train, 20% validation, 20% test) with 5-fold cross-validation, rather than the subsets in Appendix F, across MIL pooling architectures in Table 12 and augmentations in Table 3. Even in this setting, while some gains appear, the overall trend holds, and Maxsoft continues to surpass prior methods. Together with Tables 1 and 3, these results indicate that Maxsoft improves generalization regardless of data availability, consistent with its inductive bias.

## Q   Out-of-Distribution Generalization (CAMELYON17 → CAMELYON16)

While our primary focus is in-distribution generalization, motivated by growing interest in out-of-distribution (OOD) generalization, we report OOD generalization results by training on CAME-LYON17 [35] and evaluating on CAMELYON16 [78] with no exposure to the latter during training. As shown in Table 13, Maxsoft again overall surpasses prior methods, consistent with the established observation that stronger in-distribution accuracy often correlates with improved OOD performance [104–107]. A notable exception is DINO Domain, where mean pooling outperforms all methods—and even its counterparts with other encoders. We hypothesize that because CAMELYON17 contains positive slides with only a single malignant patch, many models (including Maxsoft) become overly

Table 11: Performance of major MIL pooling functions on CAMELYON17.

| Encoder | Method | CAMELYON17 | | | | | |
|---|---|---|---|---|---|---|---|
| | | Slide | | | | Patch | |
| | | AUC | ACC | F1 | ECE | AUC | F1 |
| DINO Natural | LSE pooling | $0.683_{093}$ | $0.600_{087}$ | $0.654_{070}$ | $0.361_{075}$ | $0.168_{291}$ | $0.040_{069}$ |
| | GM pooling | $0.670_{017}$ | $\mathbf{0.700_{000}}$ | $0.700_{000}$ | $0.166_{023}$ | $0.643_{002}$ | $0.034_{000}$ |
| | ISR pooling | $0.620_{030}$ | $0.600_{000}$ | $0.533_{000}$ | $0.213_{022}$ | $0.453_{046}$ | $0.035_{060}$ |
| | noisy-and pooling | $0.667_{005}$ | $0.583_{028}$ | $0.625_{000}$ | $\mathbf{0.095_{003}}$ | $\mathbf{0.655_{002}}$ | $\mathbf{0.049_{000}}$ |
| | noisy-or pooling | $0.500_{000}$ | $0.500_{000}$ | $0.000_{000}$ | $0.500_{000}$ | $0.568_{124}$ | $0.021_{008}$ |
| | smoothmax pooling | $0.643_{040}$ | $0.583_{104}$ | $\mathbf{0.714_{031}}$ | $0.248_{055}$ | $0.505_{015}$ | $0.000_{001}$ |
| | Maxsoft pooling | $\mathbf{0.710_{010}}$ | $0.650_{050}$ | $0.658_{083}$ | $0.345_{040}$ | $0.312_{024}$ | $0.000_{000}$ |
| DINO Domain | LSE pooling | $0.850_{070}$ | $0.800_{050}$ | $0.833_{021}$ | $0.185_{066}$ | $0.836_{026}$ | $\mathbf{0.499_{113}}$ |
| | GM pooling | $0.867_{023}$ | $0.717_{029}$ | $0.822_{003}$ | $0.136_{106}$ | $0.800_{012}$ | $0.303_{021}$ |
| | ISR pooling | $0.863_{031}$ | $0.683_{029}$ | $0.812_{041}$ | $0.198_{018}$ | $0.796_{011}$ | $0.206_{022}$ |
| | noisy-and pooling | $0.873_{006}$ | $0.650_{000}$ | $0.842_{037}$ | $0.169_{028}$ | $0.799_{014}$ | $0.055_{003}$ |
| | noisy-or pooling | $0.500_{000}$ | $0.500_{000}$ | $0.000_{000}$ | $0.500_{000}$ | $0.646_{121}$ | $0.029_{011}$ |
| | smoothmax pooling | $0.803_{035}$ | $0.583_{076}$ | $0.703_{118}$ | $0.302_{066}$ | $0.774_{028}$ | $0.292_{058}$ |
| | Maxsoft pooling | $\mathbf{0.983_{055}}$ | $\mathbf{0.867_{076}}$ | $\mathbf{0.935_{072}}$ | $\mathbf{0.121_{051}}$ | $\mathbf{0.839_{019}}$ | $0.386_{237}$ |
| UNI | LSE pooling | $0.603_{351}$ | $0.617_{247}$ | $0.628_{400}$ | $0.386_{230}$ | $0.709_{213}$ | $0.210_{364}$ |
| | GM pooling | $0.730_{121}$ | $0.667_{144}$ | $0.744_{097}$ | $\mathbf{0.114_{014}}$ | $0.700_{168}$ | $0.030_{012}$ |
| | ISR pooling | $0.607_{235}$ | $0.583_{189}$ | $0.712_{058}$ | $0.397_{176}$ | $\mathbf{0.830_{130}}$ | $0.186_{322}$ |
| | noisy-and pooling | $0.727_{138}$ | $0.633_{126}$ | $0.769_{069}$ | $0.191_{033}$ | $0.616_{154}$ | $0.036_{014}$ |
| | noisy-or pooling | $0.500_{000}$ | $0.500_{000}$ | $0.000_{000}$ | $0.500_{000}$ | $0.642_{121}$ | $0.028_{020}$ |
| | smoothmax pooling | $0.537_{319}$ | $0.533_{104}$ | $0.573_{343}$ | $0.346_{143}$ | $0.650_{263}$ | $0.204_{352}$ |
| | Maxsoft pooling | $\mathbf{0.753_{071}}$ | $\mathbf{0.750_{205}}$ | $\mathbf{0.779_{040}}$ | $0.238_{172}$ | $0.786_{255}$ | $\mathbf{0.476_{429}}$ |
| Prov-GigaPath | LSE pooling | $0.933_{031}$ | $0.900_{050}$ | $0.932_{027}$ | $0.095_{048}$ | $0.943_{004}$ | $0.741_{108}$ |
| | GM pooling | $0.900_{050}$ | $0.767_{058}$ | $0.825_{107}$ | $0.135_{029}$ | $0.915_{011}$ | $0.374_{037}$ |
| | ISR pooling | $0.930_{056}$ | $0.867_{029}$ | $0.893_{053}$ | $0.144_{035}$ | $0.916_{026}$ | $0.658_{165}$ |
| | noisy-and pooling | $0.893_{006}$ | $0.783_{029}$ | $0.881_{033}$ | $0.150_{070}$ | $0.836_{016}$ | $0.138_{008}$ |
| | noisy-or pooling | $0.500_{000}$ | $0.500_{000}$ | $0.000_{000}$ | $0.500_{000}$ | $0.757_{130}$ | $0.301_{021}$ |
| | smoothmax pooling | $0.917_{025}$ | $0.700_{050}$ | $0.889_{000}$ | $0.245_{021}$ | $0.930_{004}$ | $0.704_{035}$ |
| | Maxsoft pooling | $\mathbf{1.000_{000}}$ | $\mathbf{0.933_{029}}$ | $\mathbf{1.000_{000}}$ | $\mathbf{0.062_{022}}$ | $\mathbf{0.948_{009}}$ | $\mathbf{0.744_{034}}$ |

sensitive, whereas mean pooling is less affected. This explains mean pooling's advantage within DINO Domain; its superiority over all encoders, however, warrants further analysis.

# R   MIL Datasets Out of Pathology Context

To evaluate the broader applicability of our approach beyond the pathology domain, we test our proposed method—alongside prior pathology MIL methods—on classical MIL benchmark datasets. These include MUSK1 and MUSK2, which model molecular binding: each molecule is represented by multiple conformations and is labeled positive if at least one conformation binds to a target protein, though the binding instance is not identified.

Animal-based datasets follow a similar MIL assumption. The Elephant dataset comprises 200 bags (100 positive, 100 negative), where each bag contains instances derived from segmented image features. A bag is labeled positive if it contains at least one elephant instance. The Tiger and Fox datasets follow the same structure, with bags labeled positive if they contain at least one instance of a tiger or fox, respectively. Instance-level labels are unavailable in all cases—only bag-level supervision is provided [108, 109].

Following standard protocol [108, 109], we conduct 10-fold cross-validation with five runs per fold and report the mean and standard deviation for each metric.

Table 12: MIL pooling results on the full CAMELYON16 and CAMELYON17 datasets [78, 35].

| Encoder | Method | CAMELYON16 Slide AUC | ACC | F1 | ECE | Patch AUC | F1 | CAMELYON17 Slide AUC | ACC | F1 | ECE |
|---|---|---|---|---|---|---|---|---|---|---|---|
| DINO Natural | max pooling | $0.691_{259}$ | $0.726_{152}$ | $0.533_{363}$ | $0.079_{067}$ | $0.751_{281}$ | $0.248_{215}$ | $0.728_{082}$ | $0.713_{064}$ | $0.631_{036}$ | $0.212_{016}$ |
| | mean pooling | $0.619_{002}$ | $0.713_{000}$ | $0.413_{000}$ | $0.092_{001}$ | $\mathbf{0.939}_{000}$ | $0.328_{001}$ | $0.739_{000}$ | $0.720_{000}$ | $0.669_{005}$ | $\mathbf{0.060}_{000}$ |
| | ABMIL | $0.857_{009}$ | $0.858_{012}$ | $0.789_{039}$ | $0.143_{014}$ | $0.810_{051}$ | $0.383_{054}$ | $0.795_{076}$ | $0.777_{075}$ | $0.723_{086}$ | $0.255_{106}$ |
| | DSMIL | $0.768_{123}$ | $0.762_{063}$ | $0.670_{147}$ | $0.081_{011}$ | $0.545_{143}$ | $0.136_{046}$ | $0.783_{117}$ | $0.767_{042}$ | $0.659_{131}$ | $0.127_{063}$ |
| | Snuffy | $0.792_{088}$ | $0.767_{041}$ | $0.706_{082}$ | $0.083_{020}$ | $0.712_{351}$ | $0.216_{182}$ | $0.798_{058}$ | $0.803_{029}$ | $0.707_{061}$ | $0.111_{039}$ |
| | LSE pooling | $0.884_{004}$ | $0.837_{008}$ | $0.826_{014}$ | $\mathbf{0.049}_{006}$ | $0.894_{004}$ | $\mathbf{0.432}_{007}$ | $0.816_{046}$ | $0.793_{025}$ | $0.707_{013}$ | $0.135_{043}$ |
| | Maxsoft pooling | $\mathbf{0.911}_{018}$ | $\mathbf{0.886}_{004}$ | $\mathbf{0.849}_{013}$ | $0.110_{002}$ | $0.899_{004}$ | $0.363_{018}$ | $\mathbf{0.883}_{011}$ | $\mathbf{0.827}_{015}$ | $\mathbf{0.749}_{008}$ | $0.207_{029}$ |
| DINO Domain | max pooling | $0.986_{001}$ | $0.953_{004}$ | $0.952_{006}$ | $0.038_{003}$ | $0.952_{001}$ | $0.650_{021}$ | $0.918_{015}$ | $0.905_{021}$ | $0.887_{007}$ | $0.053_{012}$ |
| | mean pooling | $0.621_{001}$ | $0.667_{000}$ | $0.413_{000}$ | $\mathbf{0.023}_{005}$ | $0.939_{000}$ | $0.294_{000}$ | $0.741_{001}$ | $0.680_{000}$ | $0.657_{001}$ | $0.102_{004}$ |
| | ABMIL | $0.958_{018}$ | $0.954_{008}$ | $0.939_{017}$ | $0.046_{009}$ | $0.940_{008}$ | $0.487_{215}$ | $0.889_{016}$ | $0.870_{010}$ | $0.853_{007}$ | $0.111_{017}$ |
| | DSMIL | $0.963_{005}$ | $0.948_{012}$ | $0.951_{005}$ | $0.037_{003}$ | $0.465_{046}$ | $0.114_{014}$ | $0.920_{012}$ | $0.883_{015}$ | $0.862_{019}$ | $0.068_{017}$ |
| | Snuffy | $0.827_{023}$ | $0.687_{038}$ | $0.759_{018}$ | $0.125_{041}$ | $0.931_{006}$ | $0.407_{005}$ | $0.903_{008}$ | $0.756_{015}$ | $0.813_{013}$ | $0.125_{019}$ |
| | LSE pooling | $0.978_{000}$ | $0.952_{000}$ | $0.951_{000}$ | $0.038_{004}$ | $0.948_{000}$ | $\mathbf{0.731}_{000}$ | $0.931_{022}$ | $0.897_{011}$ | $0.882_{009}$ | $0.034_{001}$ |
| | Maxsoft pooling | $\mathbf{0.993}_{002}$ | $\mathbf{0.961}_{000}$ | $\mathbf{0.958}_{000}$ | $0.037_{002}$ | $\mathbf{0.953}_{001}$ | $0.664_{030}$ | $\mathbf{0.954}_{013}$ | $\mathbf{0.927}_{011}$ | $\mathbf{0.904}_{009}$ | $\mathbf{0.027}_{006}$ |
| UNI | max pooling | $0.986_{005}$ | $0.974_{012}$ | $0.983_{012}$ | $0.027_{013}$ | $0.966_{007}$ | $\mathbf{0.662}_{087}$ | $0.827_{038}$ | $0.887_{049}$ | $0.864_{047}$ | $0.150_{039}$ |
| | mean pooling | $0.583_{005}$ | $0.594_{004}$ | $0.474_{037}$ | $0.370_{015}$ | $0.821_{003}$ | $0.232_{003}$ | $0.839_{023}$ | $0.800_{026}$ | $0.726_{054}$ | $0.338_{044}$ |
| | ABMIL | $0.970_{015}$ | $0.948_{039}$ | $0.961_{016}$ | $0.051_{041}$ | $0.928_{002}$ | $0.397_{115}$ | $0.804_{224}$ | $0.837_{170}$ | $0.791_{234}$ | $0.143_{206}$ |
| | DSMIL | $0.971_{012}$ | $0.961_{016}$ | $0.956_{030}$ | $0.057_{046}$ | $0.717_{164}$ | $0.177_{090}$ | $0.874_{145}$ | $0.907_{110}$ | $0.840_{202}$ | $0.139_{120}$ |
| | Snuffy | $0.931_{023}$ | $0.819_{049}$ | $0.832_{046}$ | $0.050_{035}$ | $0.972_{001}$ | $0.597_{010}$ | $0.932_{026}$ | $0.906_{025}$ | $0.875_{040}$ | $0.095_{009}$ |
| | LSE pooling | $0.983_{005}$ | $0.969_{008}$ | $0.965_{022}$ | $0.032_{009}$ | $0.967_{003}$ | $0.558_{105}$ | $0.846_{196}$ | $0.840_{176}$ | $0.784_{235}$ | $0.183_{156}$ |
| | Maxsoft pooling | $\mathbf{0.998}_{001}$ | $\mathbf{0.974}_{009}$ | $\mathbf{0.990}_{000}$ | $\mathbf{0.026}_{007}$ | $\mathbf{0.973}_{001}$ | $0.595_{008}$ | $\mathbf{0.982}_{006}$ | $\mathbf{0.957}_{006}$ | $\mathbf{0.943}_{002}$ | $\mathbf{0.038}_{0012}$ |
| Prov-GigaPath | max pooling | $0.980_{006}$ | $0.969_{008}$ | $0.976_{017}$ | $0.027_{005}$ | $0.964_{005}$ | $0.556_{032}$ | $0.958_{003}$ | $0.943_{012}$ | $0.921_{021}$ | $0.057_{018}$ |
| | mean pooling | $0.540_{030}$ | $0.594_{027}$ | $0.379_{074}$ | $0.335_{024}$ | $0.855_{015}$ | $0.231_{008}$ | $0.803_{002}$ | $0.770_{008}$ | $0.676_{000}$ | $0.140_{004}$ |
| | ABMIL | $0.982_{004}$ | $0.964_{024}$ | $0.976_{006}$ | $0.034_{022}$ | $0.963_{002}$ | $\mathbf{0.688}_{134}$ | $0.963_{018}$ | $0.950_{010}$ | $0.937_{017}$ | $0.050_{010}$ |
| | DSMIL | $0.977_{009}$ | $0.954_{021}$ | $0.945_{011}$ | $0.111_{047}$ | $0.500_{046}$ | $0.125_{000}$ | $0.969_{014}$ | $0.947_{012}$ | $0.943_{027}$ | $0.041_{011}$ |
| | Snuffy | $0.962_{005}$ | $0.913_{021}$ | $0.902_{010}$ | $0.121_{001}$ | $0.889_{032}$ | $0.555_{016}$ | $0.942_{011}$ | $0.890_{043}$ | $0.870_{023}$ | $0.076_{032}$ |
| | LSE pooling | $0.980_{011}$ | $0.972_{012}$ | $0.976_{012}$ | $0.028_{011}$ | $0.952_{020}$ | $0.602_{071}$ | $0.968_{007}$ | $0.956_{013}$ | $0.866_{025}$ | $0.271_{006}$ |
| | Maxsoft pooling | $\mathbf{0.988}_{003}$ | $\mathbf{0.974}_{004}$ | $\mathbf{0.979}_{010}$ | $\mathbf{0.025}_{004}$ | $0.965_{003}$ | $0.556_{016}$ | $\mathbf{0.996}_{007}$ | $\mathbf{0.975}_{007}$ | $\mathbf{0.971}_{000}$ | $\mathbf{0.030}_{002}$ |
| IN | R²T-MIL | $0.913_{007}$ | $0.869_{006}$ | $0.852_{020}$ | $0.117_{013}$ | / | / | $0.792_{045}$ | $0.800_{014}$ | $0.733_{013}$ | $0.202_{006}$ |
| UNI PUP | R²T-MIL | $0.946_{011}$ | $0.891_{016}$ | $0.879_{015}$ | $0.095_{020}$ | / | / | $0.896_{006}$ | $0.890_{000}$ | $0.887_{007}$ | $0.108_{004}$ |
| UNI | PANTHER | $0.836_{001}$ | $0.806_{000}$ | $0.776_{002}$ | $0.140_{025}$ | / | / | $0.880_{004}$ | $0.845_{007}$ | $0.825_{009}$ | $0.083_{021}$ |

As shown in Table 14, Maxsoft pooling outperforms all baselines on nearly every dataset, achieving an AUC of 1.0 on MUSK1. The observed gap between AUC and accuracy is likely due to using a fixed decision threshold rather than tuning per dataset. These results demonstrate the versatility and strong generalization of Maxsoft pooling as a broadly applicable MIL aggregation method.

Table 13: OOD generalization results from models trained on CAMELYON17 and evaluated on CAMELYON16.

| Encoder | Method | CAMELYON16 | | | | | |
|---|---|---|---|---|---|---|---|
| | | Slide | | | | Patch | |
| | | AUC | ACC | F1 Score | ECE | AUC | F1 |
| DINO Natural | max | $0.541_{085}$ | $0.587_{065}$ | $0.458_{047}$ | $0.158_{083}$ | $0.379_{083}$ | $0.000_{000}$ |
| | mean | $0.504_{002}$ | $0.442_{000}$ | $0.409_{000}$ | $0.189_{001}$ | $0.388_{002}$ | $0.093_{001}$ |
| | DSMIL | $0.527_{101}$ | $0.550_{047}$ | $0.391_{241}$ | $0.235_{089}$ | $\mathbf{0.577_{127}}$ | $\mathbf{0.154_{050}}$ |
| | LSE | $0.579_{018}$ | $0.558_{013}$ | $\mathbf{0.507_{053}}$ | $\mathbf{0.149_{033}}$ | $0.297_{028}$ | $0.006_{003}$ |
| | Maxsoft | $\mathbf{0.594_{004}}$ | $\mathbf{0.589_{035}}$ | $0.502_{070}$ | $0.187_{115}$ | $0.327_{053}$ | $0.007_{002}$ |
| DINO Domain | max | $0.477_{053}$ | $0.390_{009}$ | $0.413_{204}$ | $\mathbf{0.397_{061}}$ | $0.003_{004}$ | $0.321_{129}$ |
| | mean | $\mathbf{0.702_{000}}$ | $\mathbf{0.628_{000}}$ | $\mathbf{0.624_{000}}$ | $0.574_{000}$ | $\mathbf{0.148_{000}}$ | $0.034_{000}$ |
| | DSMIL | $0.580_{080}$ | $0.491_{100}$ | $0.505_{077}$ | $0.464_{127}$ | $0.102_{046}$ | $0.130_{129}$ |
| | LSE | $0.474_{015}$ | $0.411_{021}$ | $0.530_{018}$ | $0.578_{012}$ | $0.044_{013}$ | $0.269_{048}$ |
| | Maxsoft | $0.530_{074}$ | $0.437_{056}$ | $0.499_{036}$ | $0.407_{048}$ | $0.001_{001}$ | $\mathbf{0.441_{097}}$ |
| UNI | max | $0.653_{204}$ | $0.548_{199}$ | $0.532_{202}$ | $0.165_{122}$ | $0.682_{236}$ | $\mathbf{0.253_{388}}$ |
| | mean | $0.553_{008}$ | $0.496_{008}$ | $0.558_{006}$ | $0.392_{008}$ | $0.582_{010}$ | $0.153_{004}$ |
| | DSMIL | $0.692_{136}$ | $0.636_{113}$ | $0.650_{113}$ | $0.163_{070}$ | $0.500_{000}$ | $0.125_{000}$ |
| | LSE | $0.709_{174}$ | $0.643_{182}$ | $0.617_{174}$ | $0.122_{000}$ | $0.733_{192}$ | $0.218_{279}$ |
| | Maxsoft | $\mathbf{0.778_{180}}$ | $\mathbf{0.765_{126}}$ | $\mathbf{0.700_{203}}$ | $\mathbf{0.120_{000}}$ | $\mathbf{0.869_{119}}$ | $0.157_{147}$ |
| Prov-GigaPath | max | $0.931_{050}$ | $0.827_{126}$ | $0.878_{075}$ | $0.085_{105}$ | $\mathbf{0.958_{003}}$ | $0.495_{012}$ |
| | mean | $0.506_{004}$ | $0.491_{009}$ | $0.537_{003}$ | $0.390_{009}$ | $0.550_{003}$ | $0.141_{001}$ |
| | DSMIL | $0.897_{022}$ | $0.716_{031}$ | $0.793_{017}$ | $0.161_{064}$ | $0.500_{000}$ | $0.125_{000}$ |
| | LSE | $0.969_{009}$ | $0.938_{048}$ | $0.959_{027}$ | $0.110_{052}$ | $0.944_{024}$ | $\mathbf{0.561_{051}}$ |
| | Maxsoft | $\mathbf{0.979_{001}}$ | $\mathbf{0.951_{027}}$ | $\mathbf{0.963_{015}}$ | $\mathbf{0.050_{050}}$ | $0.938_{014}$ | $0.526_{082}$ |

Table 14: Results of MIL pooling on MUSK1, MUSK2, ELEPHANT, TIGER, and FOX datasets.

| Method | MUSK1 | | MUSK2 | | ELEPHANT | | TIGER | | FOX | |
|---|---|---|---|---|---|---|---|---|---|---|
| | AUC | ACC | AUC | ACC | AUC | ACC | AUC | ACC | AUC | ACC |
| max pooling | $0.881_{041}$ | $0.778_{000}$ | $0.861_{048}$ | $0.767_{058}$ | $0.753_{049}$ | $0.700_{050}$ | $0.879_{027}$ | $0.850_{050}$ | $0.675_{089}$ | $0.567_{058}$ |
| mean pooling | $0.762_{083}$ | $0.704_{128}$ | $0.750_{000}$ | $0.700_{000}$ | $\mathbf{0.980_{000}}$ | $0.917_{029}$ | $0.896_{012}$ | $0.817_{058}$ | $0.645_{033}$ | $0.650_{050}$ |
| LSE pooling | $0.976_{041}$ | $0.815_{128}$ | $0.931_{024}$ | $0.750_{116}$ | $0.957_{059}$ | $0.900_{050}$ | $0.909_{027}$ | $0.883_{058}$ | $0.720_{093}$ | $0.625_{106}$ |
| ABMIL | $0.976_{041}$ | $0.870_{111}$ | $0.833_{072}$ | $\mathbf{0.800_{173}}$ | $0.973_{021}$ | $0.933_{029}$ | $0.892_{067}$ | $0.850_{100}$ | $0.698_{104}$ | $0.617_{029}$ |
| DSMIL | $0.929_{000}$ | $0.889_{000}$ | $0.875_{042}$ | $0.733_{058}$ | $0.947_{015}$ | $0.925_{007}$ | $0.919_{030}$ | $0.833_{029}$ | $0.730_{079}$ | $0.617_{029}$ |
| Snuffy | $0.893_{051}$ | $0.833_{079}$ | $0.917_{000}$ | $0.800_{141}$ | $0.955_{007}$ | $0.825_{035}$ | $0.899_{057}$ | $0.850_{071}$ | $0.756_{059}$ | $0.625_{035}$ |
| Maxsoft pooling | $\mathbf{1.000_{000}}$ | $\mathbf{0.889_{000}}$ | $\mathbf{0.954_{024}}$ | $0.767_{058}$ | $0.958_{036}$ | $\mathbf{0.950_{050}}$ | $\mathbf{0.950_{011}}$ | $\mathbf{0.900_{029}}$ | $\mathbf{0.760_{033}}$ | $\mathbf{0.650_{029}}$ |

