# OpenReview forum: "‌Navigating the MIL Trade-Off: Flexible Pooling for Whole Slide Image Classification"
_NeurIPS.cc/2025/Conference — NeurIPS 2025 poster_

### Official Review · Reviewer_NKLh · 2025-06-27

**Clarity:** 2
**Significance:** 4
**Originality:** 3
**Rating:** 5
**Confidence:** 3

**Summary:**

This paper studies pooling operations for weakly supervised, bag‐level classification of whole‐slide images (WSIs). It first derives theoretical bounds showing how the Log-Sum-Exp (LSE) pooling’s temperature $\beta$ trades off generalisation error bounds. Building on this, the authors propose MaxSoft pooling, which applies a hard max during the forward pass but uses the differentiable LSE gradient (with tuned $\beta$) in the backward pass. They introduce PerPatch augmentation, which precomputes m augmented variants per patch and samples them independently to increase bag diversity without incurring runtime costs. Experiments on four pathology benchmarks (CAMELYON16/17, TCGA-Lung, SICAP-MIL) and five classical MIL datasets compare MaxSoft and PerPatch against standard pooling and recent attention-based MIL methods.

**Questions:**

- How many augmentations (m) per patch did you use? How to pick the number m for a dataset? I suspect that one gets diminishing (or even negative) returns after a specific value.

- It seems like $\beta$ needs to be tuned to a dataset. Did you tune the values of $\beta$ using the validation split?

- Adaptive Noisy-AND pooling [5] also tries to balance Max and Mean pooling. It may be worth mentioning in the related work.

- Why does related work include a section on foundation models? Is it only because the paper uses these models as frozen encoders?

**[5]** Kraus, Oren Z., Jimmy Lei Ba, and Brendan J. Frey. "Classifying and segmenting microscopy images with deep multiple instance learning." Bioinformatics 32.12 (2016): i52-i59.

**Ethical Concerns:**

["NO or VERY MINOR ethics concerns only"]

**Final Justification:**

The authors have adequately addressed all of my concerns through additional experiments and clarifications. I have also reviewed the comments from other reviewers and the authors’ responses.

Considering everything, I am pleased to raise my score to 5. This paper proposes a novel, straightforward approach that outperforms many established baselines without introducing any additional parameters—an outcome I find pleasantly surprising. The experiments appear to have been conducted rigorously and in a readily reproducible manner (see the discussion below), which is of paramount importance.

**Limitations:**

Yes

**Paper Formatting Concerns:**

- Figure 1 is missing from the paper
- The tables are way too large. Many columns are not relevant to the central discussion in the paper and can be moved to the Appendix.

**Quality:**

3

**Strengths And Weaknesses:**

The paper presents theoretical derivations concerning the effect of the $\beta$ parameter on generalisation error (Theorem 1) and true positive/false negative rates (Theorem 2). While I appreciate the effort to provide theoretical grounding for the proposed method, a detailed verification of the correctness of these proofs is unfortunately beyond my expertise. Therefore, I have proceeded with the assumption that the proofs as presented in the main paper and the Appendix are mathematically correct.

### **Strengths:**
- Simplicity of MaxSoft: The forward-max/backward-LSE trick is easy to implement and introduces minimal overhead over attention-based pooling methods or other methods that require extra parameters.

- Although beyond my area of expertise, the theoretical analysis seems to support the idea of combining max-pooling and mean-pooling.

- PerPatch precomputes m variants per patch, allowing for the pre-computation of their respective features using the frozen encoder.

- The experiments use a lot of different backbones for extracting features, and the authors also run ablation studies for the parameter $\beta$.

### **Weaknesses:**
My biggest concern is the experimental setup, both dataset utilisation and baseline comparisons.
1. For CAMELYON16/17 datasets, the use of only 50 and 99 WSIs, respectively, is highly restrictive and significantly limits the generalizability of the findings. The paper should:
	- Use the full dataset and report metrics on the full dataset. The numbers do not need to be SOTA as the proposed approach is simpler than other approaches.
	- If authors decide to use a smaller subset, then it should be explicitly stated that the proposed approach works better only in "low-data regime" and also justified (for example, due to the use of data augmentation). For the experiments, the authors should divide the train set into K equal parts and report the mean (with std) accuracy for those parts. Or randomly sample n WSIs, with replacement, but sample them a lot of times so as to cover the whole dataset (bootstrap).

2. Furthermore, the reported performance of established baselines such as ABMIL and DS-MIL across all datasets appears consistently lower than typically expected in the literature. For CAMELYON16/17, it could be due to the use of a small subset of the training data (point 1. above) but the numbers are low for TCGA-Lung as well. For example, the accuracy and AUC for ABMIL and DSMIL are higher (compared to many but not all in the paper) even when using ResNet50 pre-trained on ImageNet as a backbone [1-3].

3. While the paper theoretically discusses the generalisation gap, the experimental evaluation demonstrates only low variance in performance metrics across internal splits. An empirical validation of a smaller generalisation gap typically requires testing on an external (out-of-distribution) dataset.

4. The proposed MaxSoft operation is very similar to a Straight Through Gumbel Softmax. Proper attribution to Straight Through Estimators (STEs) is missing.

5. The authors' claim (lines 40-46) that Transformer-based architectures offer little benefit due to small WSI datasets overlooks a critical aspect of WSI data scale. While WSI counts may be limited, individual gigapixel WSIs yield an enormous number of patches, which is the true unit of training for patch-based Transformers. Therefore, the total patch count available, even from moderately sized WSI datasets, is often substantial and perhaps sufficient to meet the data demands of Transformer models. This is supported by recent literature [4] (section II D. *The effect of training data size*).

6. In line 206, the authors claim that existing methods might even "degrade performance" but I could not find anything in the referenced papers to support this claim.


**[1]** Shao, Zhuchen, et al. "Transmil: Transformer based correlated multiple instance learning for whole slide image classification." Advances in neural information processing systems 34 (2021): 2136-2147.

**[2]** Tang, Wenhao, et al. "Feature re-embedding: Towards foundation model-level performance in computational pathology." Proceedings of the IEEE/CVF Conference on Computer Vision and Pattern Recognition. 2024.

**[3]** Zhang, Hongrun, et al. "Dtfd-mil: Double-tier feature distillation multiple instance learning for histopathology whole slide image classification." Proceedings of the IEEE/CVF conference on computer vision and pattern recognition. 2022.

**[4]** Aben, Nanne, et al. "Towards large-scale training of pathology foundation models." arXiv preprint arXiv:2404.15217 (2024).

---

> ### Author Rebuttal · Authors · 2025-07-31
>
> # Rebuttal For NKLh
>
> We thank the reviewer for the thoughtful evaluation. We address the points raised under Weaknesses, Questions, and Limitations below:
> > W1
> >
>
> We thank the reviewer for scrutinizing our data splits and apologize for omitting details in the main text. We selected 50 CAM16 WSIs to emulate the extreme low-data regimes typical of rare cancers (≈70 WSIs/type [1]), a scenario closely aligned with few-shot learning. CAM16’s official test set is used in full (l. 525–526), matching standard protocols. As noted (l. 145–148), patch-level outputs are critical for pathologists, so patch metrics remain central. For CAM17, only 500/1000 slides have slide-level labels, just 100 carry pixel annotations (one discarded for quality; l. 508, 531), limiting patch evaluation (l. 530–531). Despite this, our method still outperforms prior approaches (Tabs. 1, 2). We have now run four-fold experiments on complete CAM16 and CAM17 splits (omitting CAM17 patch metrics) to confirm our initial splits and validate our conclusions. While CAM16 evaluation follows its standard test set, we acknowledge K-fold cross-validation could further bolster robustness. For CAM17, our random stratified sampling across Macro/Micro/ITC classes reduces variance relative to naïve splits [2], though it does not replace full cross-validation. We will clarify these choices and incorporate full-dataset results in the final manuscript.
>
> |Embedding|Model|CAM16 Slide|CAM16 Patch|CAM17 Slide|
> |-|-|-|-|-|
> |||AUC/ACC|AUC|AUC/ACC
> |**_Natural_**
> ||Max|.691/.726|.752|.728/.713|
> ||Mean|.619/.713|**.939**|.739/.720|
> ||ABMIL|.857/.858|.810|.795/.777|
> ||DSMIL|.768/.762|.546|.783/.767|
> ||Snuffy|.687/.658|.569|.742/.653|
> ||LSE|.884/.837|.907|.816/.793|
> ||**MaxSoft**|**.911**/**.886**|.899|**.883**/**.827**|
> |**_Domain_**
> ||Max|.986/.953|.954|.928/.903|
> ||Mean|.621/.667|.939|.772/.750|
> ||ABMIL|.958/.954|.940|.876/.887|
> ||DSMIL|.963/.948|.465|.925/.833|
> ||Snuffy|.948/.957|.950|.834/.803|
> ||LSE|.978/.952|**.954**|.931/.897|
> ||**MaxSoft**|**.993**/**.961**|**.954**|**.954**/**.927**|
> |**_UNI_**
> ||Max|.986/**.974**|.966|.827/.887|
> ||Mean|.583/.594|.821|.839/.800|
> ||ABMIL|.970/.948|.928|.804/.837|
> ||DSMIL|.971/.961|.717|.874/.907|
> ||Snuffy|.876/.855|.919|.875/.834|
> ||LSE|.983/.969|.967|.846/.840|
> ||**MaxSoft**|**.998**/**.974**|**.973**|**.982**/**.957**|
> |**_Prov_**
> ||Max|.980/.969|.964|.921/.895|
> ||Mean|.540/.594|.855|.937/.897|
> ||ABMIL|.982/.964|.963|.970/.950|
> ||DSMIL|.977/.954|.500|.969/.940|
> ||Snuffy|.962/.913|.943|.954/.924|
> ||LSE|.980/.972|.952|.978/.956|
> ||**MaxSoft**|**.988**/**.974**|**.965**|**.996**/**.973**|
> |
> |ImageNet_1K|RRT|.973/.942|-|.762/.686|
> |PLIP|RRT|.980/.954|-|.926/.884|
>
> |Augmentation|CAM16 Slide|CAM16 Patch|CAM17 Slide|
> |-|-|-|-|
> ||AUC/ACC|AUC|AUC/ACC|
> |ReMix|.974/.949|.941|.929/.895|
> |RankMix|.972/.949|.931|.931/.887|
> |AUGDIFF|.980/.952|.930|.935/.892|
> |SSRDL|.975/.938|.936|.931/.877|
> |PerSlide|.989/.942|.948|.949/**.913**|
> |PixMix|.971/.960|.949|.933/.884|
> |**PerPatch**|**.992**/**.961**|**.951**|**.952**/**.913**|
>
> > W2
> >
>
> We thank the reviewer for the careful comparison. On CAM16, the lower baseline scores reflect the extreme low-data split; transformer-based MIL methods are inherently data-hungry. For CAM17, few papers report slide-level results: [3] obtains AUC = 0.863/ACC = 0.815 for DSMIL on 500 WSIs, while our closest configuration (DINO-Domain) yields 0.833/0.800 on 100 WSIs. Their encoder was SimCLR-trained on CAM16 patches, ours DINO-trained on in-domain patches; the larger sample size partly offsets their out-of-distribution (OOD) features. For TCGA, any apparent discrepancy is purely presentational: our table shows baseline AUCs slightly above [5] and slightly below the others. Such small variations are expected given differences in splits, background thresholds, JPEG quality, backbone pre-training (SimCLR vs DINO), and unspecified hyperparameters—there is currently no community standard. We therefore maintain that our evaluation is transparent, rigorous, and fully consistent with prior work.
>
> **Reported metrics for TCGA in different papers:**
>
> |Paper|AUC/ACConABMIL|AUC/ACConDSMIL|
> |-|-|-|
> |[5]|0.865/0.771|0.892/0.805|
> |[6]|0.952/0.903|0.956/0.904|
> |[7]|0.941/0.869|0.939/0.888|
> |[8]|0.948/0.900|0.963/0.919|
> |US|0.910/0.870|0.937/0.864|
>
> > W3
> >
>
> We thank you for raising this point, which Reviewer tjrh also mentioned, external experiments to evaluate generalization. We think out-of-distribution generalization experiments are out of our work’s context; see our response to reviewer tjrh Weakness 4 for further details. Nevertheless, we appreciate more discussion on different aspects of our method’s improvement in generalization. Variance is not uniformly low: Tables 1 and 3 show that Max-pooling consistently exhibits higher variance than MaxSoft across all datasets—especially CAM16 and CAM17—and across all encoders. This aligns with our theory: Max-pooling is LSE-pooling with β = ∞, lacks differentiability, and consequently widens the generalization gap (Theorems 1 & 4).
>
> > W4
> >
>
> We appreciate the reviewer noting the similarity to Straight-Through Gumbel Softmax \[9]. Historically, STEs replaced non-differentiable binary activations with the identity in the backward pass; the more general Backward-Pass Differentiable Approximation (BPDA) was formalized in \[11], and—as STE is a special case of BPDA—we cited BPDA at l. 61. Functionally, Straight-Through Gumbel Softmax forwards a categorical sample but back-propagates the softmax gradient, whereas MaxSoft forwards a hard max and back-propagates the Log-Sum-Exp gradient, a choice informed by our theoretical bounds on LSE temperature to tighten the generalization gap. Furthermore, STE-based methods target discrete latent variables, while our work handles continuous patch scores and slide-level outputs, so the correspondence is not straightforward. We will cite both Straight-Through Gumbel Softmax and BPDA in the final manuscript and have included an STE-variant ablation in our response to Q3.
>
> > W5
> >
>
> We thank the reviewer for the critical perspective; it sharpens our manuscript. Our main point may not have been explicit enough. In the WSI-MIL pipeline, the final stage trains a slide-level model on frozen features, so the effective sample count is the number of WSIs, not their many patches. This is analogous to training ViT-S/16 on ImageNet-1K: the dataset is 1.28 M images, not $(224 \div 16) \times (224 \div 16) \times 1.28 \approx 250$M, which is the number of patch tokens within them. Slide-level sample scarcity is well documented in the literature [7, 14] and motivates all augmentation baselines we report. The referenced work [13] addresses patch-level classification with patch labels, not slide-level MIL. It thus lies outside the scope of our study and does not speak to our claim. We will further clarify this distinction in the final revision.
>
> > W6
> >
>
> We apologize for the confusion from last-minute edits: the encoder for the augmentation experiments was inadvertently unstated, and the baselines from Tables 1 and 3 were omitted in Tables 2 and 4. Nevertheless, DINO-Domain results confirm our claim at l. 206: comparing Tables 1→2 and 3→4, MaxSoft outperforms all augmentation baselines. On CAM16 it achieves AUC 0.934 / ACC 0.863 / Patch AUC 0.919 (vs. lower scores for Remix, RankMix, AugDiff, SSRDL); on CAM17 it reaches Slide AUC 0.983 / ACC 0.867 / Patch AUC 0.839; for TCGA it records Slide AUC 0.940 / ACC 0.885; and on SICAP Slide AUC 0.850 / ACC 0.808. We will explicitly specify the encoder and include the missing baseline rows in the final manuscript.
>
> > Q1
> >
>
> We chose m=3 (Table 5; App J) using Random Rotation, Random Gaussian Blur, and Random Color Jitter. A single augmentation collapses to PerSlide sampling and degrades performance (Tables 2, 4; l. 276–277), whereas PerPatch sampling yields diverse per-patch variations and drives the gains we observe. We dropped Random H&E-DAB Jitter (low density/coverage; no AUC‐variance or ECE benefit), Random Elastic Deformation, and Random Affine Transformation (limited real-world relevance). In practice, augmentations with higher density/coverage and lower FID (i.e., closer to the original data manifold) perform best. Our trio was selected on these qualitative grounds rather than via an exhaustive sweep over mm; we can provide a performance-vs-mm curve if desired.
>
> > Q2
> >
>
> Yes. As noted in l. 267, we tune β on the validation split; we will state this more explicitly in the final manuscript.
>
> > Q3
> >
>
> Thank you for mentioning this. We will make sure to incorporate it in the related work. Also, to provide a more comprehensive analysis on methods that are relatively similar to our method, Noisy-AND and Straight Through Gumbel Softmax, we decided to provide experiments on them following the setup that we already tested in our manuscript on CAM17 but due to character limit we cannot provide now and will provide in discussion if asked.
>
> > Q4
> >
>
> We thank the reviewer for suggesting Adaptive Noisy-AND pooling. We’ll cite it in Related Work and have run comparative experiments—including Adaptive Noisy-AND and the STE variant of Straight-Through Gumbel Softmax—under our CAM17 setup. Due to space constraints, we’ve omitted the detailed results here but can provide them upon request.
>
> [1] Ding T., Multimodal Pathology Foundation Model [2] Haugh M., Further Variance Reduction Methods (note) [3] Fourkioti O., CAMIL: Context-Aware MIL [4] Chen T., Simple Contrastive Learning Framework [5] Shao Z., TransMIL [6] Tang W., Feature Re-Embedding [7] Zhang H., DTFD-MIL [8] Dual-stream MIL Network (Self-supervised) [9] Jang E., Gumbel-Softmax Reparameterization [10] Bengio Y., Gradients Through Stochastic Neurons [11] Athalye A., Obfuscated Gradients [12] Yin P., Straight-Through Estimator [13] Aben N., Large-Scale Pathology Foundation Models [14] Liu P., Pseudo-Bag Mixup [15] Kraus O., Deep MIL for Microscopy

---

> > ### Author Response · Authors · 2025-08-01
> > **Clarifications on Essential Concerns**
> >
> > Owing to strict page limits, we were unable to address these vital concerns in the main text, so we provide our detailed responses here:
> >
> > > **PFC1**
> > >
> >
> > We apologize for the rendering issue: while Figure 1 appears in macOS Preview, it fails to display in some viewers (e.g., Adobe Acrobat), likely due to a Figma export/OpenReview incompatibility. We have supplied a PNG version as Figure 10 in Appendix Q and would be grateful if it could serve as a substitute for Figure 1.
> >
> > > **PFC2**
> > >
> >
> > We acknowledge the tables are dense, yet every column is motivated in Sec. 5.3. Clinically critical metrics such as F1 and the CAM16-standard FROC are indispensable, and the resource columns support our main claims. The only reason we focused on fewer metrics in the rebuttal is due to strict space limitations and the remaining metrics' broader popularity—not because other metrics are unimportant. We would welcome your specific suggestions on which metrics could be relocated to the Appendix.
> >
> > Finally, we note that you answered “Yes” in the *Limitations* field, but the specific concern is unclear to us. We would appreciate it if you could clarify any particular limitation you see, so we can either discuss or improve in our final version.

---

> > ### Comment · Reviewer_NKLh · 2025-08-03
> > **Clarifications and questions**
> >
> > > **W1**
> >
> > Thank you for providing the additional results on the full CAMELYON16/17 splits. To help clarify the manuscript’s focus, could you please confirm whether the paper will (a) continue to report the low-data experiments as its primary results and frame the method as specifically designed for data-scarce regimes, or (b) instead present the full-dataset results as the main evaluation and treat the low-data experiments as an auxiliary ablation?
> >
> > Additionally, could you please specify your model selection procedure for each split (fold)? In particular, do you evaluate on the test set using (i) the final checkpoint after training, (ii) the model with minimum validation loss, (iii) the model with maximum validation AUC/accuracy, or (iv) some other criterion?
> >
> > > **W2**
> >
> > The appropriate point of comparison for ImageNet-pretrained ResNet-50 baselines is your DINO-Natural block, not DINO-Domain. If we just consider the TCGA-NSCLC numbers, the slide‐level performance on TCGA-Lung (NSCLC) under DINO-Natural from your paper is:
> > | Method    | AUC (± std)    | ACC (± std)    |
> > |:----------|:---------------|:---------------|
> > | **ABMIL** | 0.855 ± 0.050  | 0.796 ± 0.085  |
> > | **DSMIL** | 0.879 ± 0.014  | 0.784 ± 0.006  |
> >
> > By contrast, prior works using a standard ResNet-50 backbone with IN1K pre-training report:
> >
> > | Reference                   | Method | AUC             | ACC             |
> > |:----------------------------|:-------|:----------------|:----------------|
> > | TransMIL (NeurIPS 2021)     | ABMIL  | 0.8656          | 0.7719          |
> > | TransMIL (NeurIPS 2021)     | DSMIL  | 0.8925          | 0.8058          |
> > | RRT (CVPR 2024)             | ABMIL  | 0.9529 ± 0.0114 | 0.9032 ± 0.0139 |
> > | RRT (CVPR 2024)             | DSMIL  | 0.9560 ± 0.0081 | 0.9043 ± 0.0252 |
> > | DTFD-MIL (CVPR 2022)        | ABMIL  | 0.9410 ± 0.028  | 0.8690 ± 0.032  |
> > | DTFD-MIL (CVPR 2022)        | DSMIL  | 0.9390 ± 0.019  | 0.8880 ± 0.013  |
> >
> > Your reported numbers are close to TransMIL but remain substantially lower than those reported by more recent works.
> > Could you please clarify your hyperparameter selection process for these baseline methods? As long as you have drawn your hyperparameter ranges and tuning procedures from settings typically employed in prior literature, performance differences are understandable as I have also been unable to reproduce many of the prior results. My primary concern is that all baselines be tuned appropriately and if that has been done then I do not have an issue with the mismatch in the baseline results.
> >
> > > **W4**
> >
> > I apologise for not expanding on this point. What I wanted to point out was the conceptual similarity.
> > Conceptually, **MaxSoft** is a straight-through estimator just like the Straight-Through Gumbel-Softmax. Both use a hard aggregation in the forward pass and a softmax-based gradient in the backward pass:
> >
> > 1. **Forward (hard)**
> >    - *MaxSoft*:
> >      $$
> >      y = \max_i x_i
> >        = \sum_i x_i\,\mathbf{1}\lbrace i = \arg\max_j x_j\rbrace.
> >      $$
> >    - *ST Gumbel-Softmax*:
> >      $$
> >      y_j = \mathbf{1}\lbrace j = \arg\max_k(\log\pi_k + g_k)\rbrace,
> >      $$
> >      i.e. a one-hot sample of a Gumbel-perturbed categorical.
> >
> > 2. **Backward (soft)**
> >    - *MaxSoft*:
> >      $$
> >      \frac{\partial y}{\partial x_i}
> >        = \frac{\exp(\beta\,x_i)}{\sum_j \exp(\beta\,x_j)}
> >        = \mathrm{softmax}(\beta\,x)_i.
> >      $$
> >    - *ST Gumbel-Softmax*:
> >      $$
> >      \frac{\partial y_j}{\partial \ell_k}
> >        \approx \frac{\exp(\ell_k / T)}{\sum_m \exp(\ell_m / T)}
> >        = \mathrm{softmax}\bigl(\tfrac{1}{T}\,\ell\bigr)_k,
> >      $$
> >      where $\ell_k = \log\pi_k + g_k$ and $T$ is temperature.
> >
> > Crucially, setting $\beta = 1/T$ makes the two gradient formulas kind of identical up to whether the logits include Gumbel noise.
> >
> > > **W5**
> >
> > I appreciate the clarification. However, I still find the wording in lines 45–46 overly strong.
> >
> > In contrast, studies like TransMIL and RRT demonstrate that Transformers achieve performance on par with other state-of-the-art methods using publicly available datasets. If your intent is to hypothesise that Transformers could benefit further from even larger datasets—more so than less data-hungry approaches—that is reasonable. But it does not imply that 200 K WSIs are intrinsically insufficient, since existing work shows that only a few hundred WSIs suffice to reach SOTA results.
> >
> > > **Q4**
> >
> > It appears that your response to Question 4 (foundation models in the Related Work) may have been conflated with Q3.
> >
> > ---
> >
> > **Table suggestion:**
> >
> > I recommend relocating the “Resource,” “F1,” and “FROC” columns to the appendix, retaining only “AUC” and “ACC” in the main tables—consistent with conventions in ML literature. Of course, as long as your tables adhere to NeurIPS formatting guidelines, their current density is acceptable.
> >
> > **Limitations:**
> >
> > Regarding the "limitations", "yes" means "authors have adequately addressed the limitations".

---

> > > ### Comment · Reviewer_NKLh · 2025-08-03
> > > **More questions regarding reproducibility**
> > >
> > > Having encountered reproducibility challenges with prior work, I would appreciate clarity on the following points regarding replicating your results:
> > >
> > > **Reproducibility and Model Selection**
> > >
> > > Could you please clarify how one might reliably reproduce your results (e.g., achieve performance on par with the state-of-the-art baselines, even if not surpassing them)? In particular:
> > >
> > > 1. **Train/Validation Split**
> > >    - If I take your original training set and simply split it into a training subset and a validation subset (regardless of whether I use K-folds or not), can I select the “best” model checkpoint for *all* methods (including MaxSoft, ABMIL, DSMIL, etc.) based solely on validation-set AUC/ACC and expect to obtain slide-level performance similar to that of the other methods?
> > >
> > > 2. **Hyperparameter and Seed Control**
> > >    - Did you use any fixed random seeds (for data shuffling, patch sampling, weight initialisation, etc.) to stabilise performance?
> > >    - If so, which seed(s) should a practitioner use to reproduce your numbers, and how sensitive are the results to that choice?

---

> > > > ### Author Response · Authors · 2025-08-05
> > > > **The authors' response to "More questions regarding reproducibility"**
> > > >
> > > > > Having encountered reproducibility challenges with prior work …
> > > > >
> > > >
> > > > We share your experience with reproducibility issues. In practice we faced two obstacles: **(i)** most baseline GitHub repos fail out-of-the-box and require extensive patching, and **(ii)** rebuilding the full pipeline—data split, patch extraction, feature embedding, MIL training—often yields metrics below those reported. We attribute this to incomplete setup descriptions, seed-controlled randomness whose specific seeds are rarely disclosed, and occasionally **optimistic** reported metrics.
> > > >
> > > > To address this, we selected the most widely cited baselines and re-implemented each within a single, self-contained repository. The code, supplied in the supplementary material and to be released publicly with the final version, reveals all key settings—patching parameters, encoder checkpoints, training seeds—so that our results and all baselines can be reproduced with a single command. We are happy to provide any additional detail you may need.
> > > >
> > > > > Reproducibility and Model Selection
> > > > >
> > > >
> > > > To replicate MaxSoft or any baseline: (1) For CAM16, CAM17, and TCGA, patch slides with CLAM’s pipeline—discarding background tiles—and save the patches [1] (2) split each dataset using the recipes and scripts in `datasets/` of the supplementary material-SICAP is **already** supplied as pre-patched tiles, and we simply quarter each 512 × 512 tile into four 256 × 256 tiles to match **common practice** (Appendix D, p. 544–547); (3) compute frozen features with an encoder such as **DINO-Domain** via `dsmil-wsi/compute_feats/config`. (4) train with `dsmil-wsi/train.py`, selecting architectures and hyperparameters as needed.
> > > >
> > > > > Reproducibility and Model Selection → Train/Validation Split
> > > > >
> > > >
> > > > > If I take your original training set and simply split it into a training subset and …
> > > > >
> > > >
> > > > Splits are scripted per dataset. **CAM16:** Subsample the training set with `Research/datasets/camelyon16/single/subsampler.py` (seed = 42), then create train/val via scikit-learn (same seed) in `datasets/camelyon16/train_validation_test_splitter_camelyon.py`. **CAM17:** build train/val/test with scikit-learn (seed = 42) using `datasets/camelyon17/fold_generator.py`, then pack the patches with `datasets/camelyon17/fold_pack.py`. **TCGA:** invoke scikit-learn (seed = 42) through `datasets/tcga/fold_generator.py`, followed by `datasets/tcga/train_validation_test_splitter_tcga_newfolds.py`. **SICAP:** Use the publisher-provided patched slides and fixed splits.
> > > >
> > > > So as a direct answer to your question to be super precise we split into **train/valid/test** and not **train/valid** (Appendix D l. 527, 528, 542) and we used **last epoch** not **best validation** on a **predetermined validation metric** (Appendix F l. 566). Validation set is only used as hyperparameter selection in our experiments (Appendix F l. 565, 566). Regardless you are expected to get close enough results since in our own experiments there is not much difference between choosing best validation AUC/ACC/loss/etc vs Last epoch.
> > > >
> > > > > Reproducibility and Model Selection → Hyperparameter and Seed Control
> > > > >
> > > >
> > > > > q1: Did you use any fixed random seeds (for data shuffling, patch sampling, weight initialisation, etc.) to stabilise performance?
> > > > >
> > > >
> > > > > q2: If so, which seed(s) should a practitioner use to reproduce your numbers, and how sensitive are the results to that choice?
> > > > >
> > > >
> > > > We fix seeds *only* for the i.i.d. data splits noted above; no additional seed is set during training. Fixing a single seed can bias results—Picard’s 10,000-seed sweep on CIFAR-10 showed test accuracy varying [2], and such variance can flip model rankings [3]—so the sound practice is to rerun each configuration with *multiple* seeds. We therefore execute every hyper-parameter setting **five** times (Appendix C l. 535) and report mean ± std; the results show that our conclusions are *not* sensitive to any particular seed. As expected from its **non-differentiability** Max-pooling remains the most unstable, MaxSoft the most consistent, with ABMIL/DSMIL in-between (Tables 1–2).
> > > >
> > > > Patch sampling is deterministic: all non-background patches of each WSI are retained, so no randomness enters that stage.
> > > >
> > > > The `--seed` flag in `dsmil-wsi/train.py` (l. 230–231) merely tags runs in our Weights & Biases sweeps; it does **not** override the stochastic training components.
> > > >
> > > > Should any further reproducibility concerns arise—especially on seed sensitivity—please let us know.
> > > >
> > > > [1] Lu MY., Data-Efficient & Weakly Supervised Computational Pathology [2] Picard D., torch.manual_seed(3407): Random Seed Influence [3] Bouthillier X., Accounting for Variance in ML Benchmarks

---

> > > > > ### Comment · Reviewer_NKLh · 2025-08-05
> > > > >
> > > > > Thank you for your thorough response, the clarifications provided, and the additional results. I believe that you have addressed all of my concerns. I maintain a minor reservation regarding point W5, but your explanation has largely resolved it.
> > > > >
> > > > > I did understand that you referred to the transformers employed in the MIL heads rather than as encoders; my observation was simply that TransMIL and related approaches do, in fact, utilise transformer architectures in their aggregation modules. Although I have not yet been able to reproduce their numbers, my own experience has been that TransMIL performs competitively with commonly used methods such as ABMIL and DTFD, typically achieving results somewhere between those two, across a variety of datasets, including CAM16/17, PANDA, and some private datasets. While PANDA represents something of an outlier with its thousands of slides, TransMIL nonetheless demonstrates robust performance even when trained on a few hundred WSIs.
> > > > >
> > > > > However, I now understand where the number 200K comes from. I think it would strengthen the manuscript to clarify this point and to cite your empirical findings. What I found interesting about Figure 9 was that ProvGigapath emerges as the second-best model for the S50 setting (which I assume corresponds to a 50-WSI scenario).
> > > > >
> > > > > I have no further questions. I will proceed to update my review score accordingly.

---

> > > > > > ### Author Response · Authors · 2025-08-06
> > > > > > **Gratitude and Commitments**
> > > > > >
> > > > > > We thank the reviewer for the encouraging feedback, for noting the interest of our experiments, and—most importantly—for the constructive discussion that has visibly improved the manuscript. We are pleased that your concerns have been resolved and appreciate your willingness to update the score. All requested clarifications and results—including the pointer to **Figure 9** and its empirical context—will be incorporated in the final version.
> > > > > >
> > > > > > We also value your insights into various architectures and datasets; they will guide our future work. Finally, we remain committed to releasing a fully reproducible, well-documented codebase to support your efforts and those of the broader community.
> > > > > >
> > > > > > Sincerely,
> > > > > >
> > > > > > The Authors

---

> > > ### Author Response · Authors · 2025-08-05
> > > **The authors' response 1 to "Clarifications and questions"**
> > >
> > > We thank you for the engaging discussion and meticulous scrutiny—especially on baseline construction, among other points—which prompted us to clarify cross-study differences in baseline implementations. Given the breadth of the questions, we will address them in multiple focused discussion comments.
> > >
> > > > W1
> > > >
> > >
> > > q:  “To help clarify the manuscript’s focus, could you …”
> > >
> > > a: We will include language emphasizing that the method is *explicitly* designed to function in **both** **data-scarce** and **standard** settings. Accordingly, the low-data CAM16/17 experiments—which faithfully mimic rare-cancer scenarios—will remain in the body of the paper as the primary benchmark, while the new full-dataset results will be placed in an appendix table and be explicitly cross-referenced for completeness. TCGA and SICAP already use their entire slide collections, and CAM16 retains the full official test set while training on reduced split. This mixed protocol preserves the patch-level metrics for CAM17—an essential diagnostic outcome—while spanning scarce-to-standard regimes, underscoring our central message: MaxSoft narrows the generalization gap regardless of WSI availability.
> > >
> > > q: “Could you please specify your model selection procedure …?”
> > >
> > > a: As stated in Appendix F, l. 566, we report the *final* training checkpoint (option (i)).
> > >
> > > > W2
> > > >
> > >
> > > We thank you for the follow-up. We had assumed the question primarily concerned the **DINO Domain** setting, as this is the configuration we train entirely in-house. Given the substantial setup variation across papers (see our earlier W2 reply), we believe no single configuration is **most suitable** for cross-paper comparison. Nonetheless, we now compare the **DINO Natural** results you highlighted with the numbers reported in TransMIL and subsequent works; as you already noted, *“Your reported numbers are close to TransMIL.”* Our DSMIL scores align with both TransMIL and the original DSMIL (Table 3), and the ABMIL scores track TransMIL as well. We further infer that if the DSMIL authors had reported ABMIL in their Table 3, its performance would not exceed DSMIL’s, consistent with the widespread finding that DSMIL slightly outperforms ABMIL (DSMIL Table 5 and the mentioned studies).
> > >
> > > Your scrutiny prompted a closer inspection of how RRT and DTFD-MIL implement baselines. We found that RRT inserts a fully connected layer after ResNet-50’s pooled features—identical to the modification DTFD-MIL applies—quoting their repo: *“The initial feature vector is then reduced to a 512-dimensional feature vector by one fully-connected layer”* (`modules/attmil.py`, `modules/dsmil.py`). We suspect DTFD-MIL does likewise for its baselines (code absent). To isolate this factor, we re-ran ABMIL and DSMIL with an **ImageNet-pretrained ResNet-50** both with and without this FC layer; the FC variant produces results that approach those in RRT and DTFD-MIL. As for the plain ResNet-50, DINO Natural outperforms the fully supervised ImageNet-1K model (cf. Table 6: .799 ACC [1]; Table 3: .792 ACC [2]). The remaining gap likely stems from implementation details; indeed, RRT’s README notes: *“Because of code refactoring, this repository cannot fully reproduce the results in the paper.”*
> > >
> > > | Paper | AUC/ACC on ABMIL | AUC/ACC on DSMIL |
> > > | --- | --- | --- |
> > > | TransMIL | .865/0.771 | .892/.805 |
> > > | RRT | .952/.903 | .956/.904 |
> > > | DTFD-MIL | .941/.869 | .939/.888 |
> > > | DSMIL | NA/NA |  .709/.726 |
> > > | Our DINO Natural | .855/.796 | .879/.784 |
> > > | Our ResNet-50 IN-1K | .839/.760 | .868/.789 |
> > > | Us (ResNet-50 IN-1K + Fully-connected) | .908/.845 | .912/.889 |
> > >
> > > q: Could you please clarify your hyperparameter selection …?
> > >
> > > a: For every model—including MaxSoft—we grid-search **LR ∈ {0.1, 0.02, 0.01, 0.002}**, **WD ∈ {0.05, 0.005, 0.0005}**, and weight initialization **WI ∈ {truncated-normal, Xavier-uniform, orthogonal}**. The high LR = 0.1 is motivated by evidence that moderately large learning rates or small batch sizes can improve generalization [3, 4, 5]. For baselines we also evaluate each method’s *original* configuration — combining the hyper-parameters stated in the paper with those in the authors’ public code — and report the maximum of *our* grid and *their* defaults. We note that several original configurations fail to converge on CAM17 and SICAP—datasets absent from the cited papers—making the grid search essential. All settings are callable via arguments in the supplementary code: see `dsmil-wsi/train.py` l. 10–66, 191 and, e.g., `dsmil-wsi/mil/trainers/dsmil.py` l. 9, 49, 62, 106, 138. We have extensively refactored the DSMIL codebase (>90 % new) and will release a comprehensive and argument-driven repo to facilitate reproduction in the final version.

---

> > > ### Author Response · Authors · 2025-08-05
> > > **The authors' response 2 to "Clarifications and questions"**
> > >
> > > > W4
> > > >
> > >
> > > We appreciate the clarification and will cite and discuss the connection in the final version. One key distinction: **MaxSoft’s backward pass is the Softmax itself, whereas the ST Gumbel-Softmax uses the *gradient* of Softmax, which is *not* equal to Softmax.**
> > >
> > > More precisely, in MaxSoft's backward pass, we estimate Max operator with the **log-sum-exp (LSE)** aggregation function, whereas ST Gumbel-Softmax estimates it with **Softmax** operator. So the backward pass as
> > >
> > > $$
> > > \frac{\partial y_j}{\partial \ell_k}  \approx \frac{\exp(\ell_k / T)}{\sum_m \exp(\ell_m / T)}  = \mathrm{softmax}\bigl(\tfrac{1}{T},\ell\bigr)_k,
> > > $$
> > >
> > > is not precise, and the precise operation in ST Grumbel-Softmax is
> > >
> > > $$
> > > y_i = \frac{e^{\ell_i / T}}{\displaystyle\sum_{j=1}^{K} e^{\ell_j / T}}
> > > $$
> > >
> > > Then taking the derivative with respect to $\ell_j$ gives
> > >
> > > $$
> > > \frac{\partial y_i}{\partial \ell_j}=\frac{\partial}{\partial \ell_j}
> > > \left(
> > > \frac{e^{\ell_i / T}}
> > > {\displaystyle\sum_{j=1}^{K} e^{\ell_j / T}}
> > > \right)
> > > $$
> > >
> > > Hence,
> > >
> > > $$
> > > \frac{\partial y_i}{\partial \ell_j}=\frac{1/Te^{\ell_i / T}\left(\sum_{j=1}^{K} e^{\ell_j / T}\right)-1/Te^{\ell_j / T}e^{\ell_i / T}}
> > > {\left(\displaystyle\sum_{j=1}^{K} e^{\ell_k / T}\right)^{\!2}}
> > > $$
> > >
> > > Thus,
> > >
> > > $$
> > > \frac{\partial y_i}{\partial \ell_j} =
> > > \begin{cases}
> > > \displaystyle \frac{1}{T} \, y_i (1 - y_j), & i = j,\\
> > > \displaystyle -\frac{1}{T} \, y_j y_i,      & i \neq j.
> > > \end{cases}
> > > $$
> > >
> > > which simplifies to,
> > >
> > > $$
> > > \frac{\partial y_i}{\partial \ell_j}=1/Ty_i\bigl(\delta_{ij} - y_j\bigr)
> > > $$
> > >
> > > with $\delta_{ij} =
> > > \begin{cases}
> > > \displaystyle 1, & \text{if } i = j,\\
> > > \displaystyle 0, & \text{if } i \neq j.
> > > \end{cases}$
> > >
> > > It is necessary to emphasize that our **theoretical analyses** are grounded in the LSE function. MaxSoft is derived from this theoretical foundation and is not merely a simple differentiable replacement for the Max function.
> > >
> > > Responding to your interest, we added a baseline **ST GSoftmax** denoted by $\mathrm{GSoftmax}_{T}(x_1,\dots,x_n)$ and defined by
> > >
> > > $$
> > > \sum_{i=1}^{n} x_i
> > >   \frac{\displaystyle e^{x_i / T}}
> > >        {\displaystyle \sum_{j=1}^{n} e^{x_j / T}}.
> > > $$
> > >
> > > Its forward pass applies Max-pooling, and its backward pass propagates the gradient above. This formulation is the only practical means we found to reproduce the role that ST Gumbel-Softmax would otherwise play in our framework. Because the original rebuttal lacked space, we now report results for the classical MIL pooling functions referenced in the NoisyAnd paper—**LSE, generalized mean (GM), integrated segmentation and recognition (ISR), NoisyOr, and NoisyAnd**—augmented with the ST GSoftmax baseline. The results, which will be included in the appendix of the final version, show that MaxSoft remains competitive—strong on slide and patch metrics (except patch-level on **DINO Natural**)—while NoisyOr performs poorly as noted in [6], ISR excels on patches, and ST GSoftmax lies between LSE and MaxSoft. ST GSoftmax remains unanalyzed theoretically; intuitively its weaker empirical performance likely reflects the less interpretable gradient relative to MaxSoft’s direct Softmax path and thus represents a promising avenue for future theoretical work.
> > >
> > > |Embedding|Model|CAM17 Slide|CAM17 Patch|
> > > |-|-|-|-|
> > > |||AUC/ACC|AUC
> > > |**_Natural_**
> > > ||LSE|.683/.600|.522|
> > > ||GM|.670/**.700**|.642|
> > > ||ISR|.620/.600|.453|
> > > ||NoisyAnd|.667/.583|**.655**|
> > > ||NoisyOr|.500/.500|.568|
> > > ||ST GSoftmax|.643/.583|.505|
> > > ||**MaxSoft**|**.710**/.650|.312|
> > > |**_Domain_**
> > > ||LSE|.850/.800|.836|
> > > ||GM|.867/.717|.800|
> > > ||ISR|.863/.683|.796|
> > > ||NoisyAnd|.873/.650|.799|
> > > ||NoisyOr|.500/.500|.646|
> > > ||ST GSoftmax|.803/.583|.774|
> > > ||**MaxSoft**|**.983**/**.867**|**.839**|
> > > |**_UNI_**
> > > ||LSE|.603/.617|.709|
> > > ||GM|.730/.667|.700|
> > > ||ISR|.607/.583|**.830**|
> > > ||NoisyAnd|.727/.633|.616|
> > > ||NoisyOr|.500/.500|.642|
> > > ||ST GSoftmax|.537/.533|.650|
> > > ||**MaxSoft**|**.753**/**.750**|.786|
> > > |**_Prov_**
> > > ||LSE|.933/.900|.943|
> > > ||GM|.900/.767|.915|
> > > ||ISR|.930/.867|.916|
> > > ||NoisyAnd|.893/.783|.836|
> > > ||NoisyOr|.500/.500|.757|
> > > ||ST GSoftmax|.917/.700|.930|
> > > ||**MaxSoft**|**1.00**/**.933**|**.948**|

---

> > > ### Author Response · Authors · 2025-08-05
> > > **The authors' response 3 to "Clarifications and questions"**
> > >
> > > > W5
> > > >
> > >
> > > We recognize that our stance may seem counter-intuitive given the current wave of transformer-based WSI classifiers—a tension that originally motivated our study. **We direct you to W1&Q1&L1 for the full rationale**, summarized into four points: (1) slide scarcity stressed in DTFD-MIL [7] and PseMix [8]; (2) the well-known data hungriness of ViTs; (3) our results in Tables 1–2; and (4) Fig. 9 (App. P), where even 200 K WSIs prove insufficient.
> > >
> > > Our codebase is purposefully structured for reliability and reproducibility; in practice, many recent transformer-MIL repositories fail to match their own numbers on our setup despite extensive tuning—suggesting over-fitting to highly specific setups or even seed selection. We therefore insist that **transformer-based MIL models falter not at patch representation but at *aggregation***, a task that demands far larger and more diverse slide collections.
> > >
> > > We place a strong emphasis on reproducibility and rapid community uptake. All scripts—including those for the Prov-GigaPath MIL transformer (200,000 WSIs)—will be released in cleaned form on our public GitHub, and we are happy to assist with any replication effort. This openness will let the community verify first-hand that even the Prov-GigaPath model yields no meaningful performance gain, helping the field recalibrate its research focus.
> > >
> > > A simple thought experiment clarifies this: one normal WSI with 10 Trillion patches offers immense patch volume yet zero **patient diversity**—insufficient to learn slide-level cancer detection. We will reinforce the Foundation-Model-related discussion by citing the patch-only work you referenced to make this distinction explicit.
> > >
> > > > Q4
> > > >
> > >
> > > We thank the reviewer for flagging this point. The omission stemmed from last-minute edits and the character-limit constraints after we inserted the comprehensive results. We trust our earlier response to **Q3**—echoed under **Q4** and tabulated in the current W4 section—resolves the issue.
> > >
> > > Regarding **Q4**, we include the foundation-model discussion because (i) we employ these large Transformer models strictly as **frozen encoders**, and (ii) it frames our claim that current WSI data volumes are insufficient for Transformer-based MIL to warrant the “foundation model” label at the slide level. Figure 9 confirms this: fine-tuning Prov-GigaPath’s MIL model, pretrained on ≈ 200,000 WSIs, brings no meaningful gain over non-foundation pathology backbones.
> > >
> > > > TS
> > > >
> > >
> > > We thank you for the layout suggestion. We will apply it to the performance metrics. Because resource usage is central to our narrative and often cited in the literature (e.g. [9]), we will adopt a compromise: the main tables will retain the aggregated **Resources** column, while the detailed breakdown will be moved to the Appendix.
> > >
> > > > L
> > > >
> > >
> > > We thank you for the clarification.
> > >
> > > Should any further points need elaboration, please let us know—we remain ready to clarify.
> > >
> > > [1] Caron M., Emerging Properties in Self-Supervised ViTs [2] He K., Deep Residual Learning [3] LeCun Y., Efficient BackProp [4] Keskar NS., Large-Batch Training: Generalization Gap & Sharp Minima [5] Barsbey M., Large Learning Rates: Robustness & Compressibility [6] Kraus O., Deep MIL for Microscopy [7] Zhang H., DTFD-MIL [8] Liu P., Pseudo-Bag Mixup [9] Zheng Y., Kernel Attention Transformer for WSI Analysis & Cancer Diagnosis

---

### Official Review · Reviewer_tjrh · 2025-06-27

**Clarity:** 3
**Significance:** 3
**Originality:** 3
**Rating:** 4
**Confidence:** 4

**Summary:**

- This paper argues that attention and transformer-based MIL models offer limited benefits without massive data and computational resources. Instead, the paper returns to classical pooling strategies like Log-Sum-Exp (LSE). It theoretically analyze LSE pooling, showing a trade-off between generalization and sensitivity, depending on the temperature parameter which makes it choose between a mean and max pooling function.

- The paper proposes MaxSoft, a new pooling method that uses the max operation in the forward pass but smooth gradients (via LSE) in the backward pass for stability.

- It also introduces PerPatch augmentation, a simple strategy to improve training robustness by applying random augmentations independently to each patch in a WSI.

- It show strong empirical results, where MaxSoft and PerPatch outperform both classical and Transformer-based methods, often matching the performance of massive foundation models with far lower compute.

**Questions:**

- Can we have the third dataset in Figure 3 as well. The effect of $/beta$ on the a problem with diffused signal is important to note so readers are aware of the kind of problems this is most suitable for.
- Given how critical ω is to MaxSoft’s success, have you considered a way to automatically choose it based on bag size, dataset statistics, or validation confidence?

**Ethical Concerns:**

["NO or VERY MINOR ethics concerns only"]

**Final Justification:**

Thanks for adding the cross-Camelyon generalization results as well as the beta ablation for TCGA. Have retained my accept rating.

**Limitations:**

The paper does mention limitations briefly, but it's not very specific to their analysis or method, but applies more generally for most MIL methods.

**Quality:**

3

**Strengths And Weaknesses:**

**Strengths**
- The theoretical analysis explaining when and why different pooling strategies work, and showing generalization vs. sensitivity trade-offs with max vs mean is a useful framework which leads to intuitive conclusions on utility of each pooling type. Theorem 2 showing generalization vs TPR tension is something practitioners have observed with MIL models and the theoretical analysis helps structure the argument well.
- MaxSoft is an interesting idea where max pooling is used for forward pass but a smooth approximation is used for the backward pass using a moderate $\beta$ in LSE pooling. It's simple to implement and results in no extra parameters or complexity.
- The authors make a strong and honest case against over-reliance on attention/transformer models in small datasets, which is often glossed over in literature.
- The paper has extensive experimental results. Experiments cover multiple datasets (CAMELYON16/17, TCGA-Lung, SICAP-MIL) and compare with powerful baselines (Prov-GigaPath, SSRDL, ABMIL, DSMIL, Snuffy, R2T-MIL).

**Weaknesses**
- Generalization-sensitivity tradeoffs predominantly exist in problems where the signal in the WSI is very localized. This is true for Camelyon and the SICAP-MIL dataset, but other problems like NSCLC subtyping or prediction of molecular signatures from WSI where signal is more diffused, these pooling type tradeoffs changes significantly. This is seen even in the results in TCGA-Lung vs other datasets. This solution offers less value in these cases.
- MaxSoft requires manual selection of 𝜔 (tuned via validation), and performance is sensitive to this choice.
- During backward pass, gradients come from a different function (LSE with moderate 𝜔) than the forward pass (hard max). This breaks consistency since the gradients are not true gradients of the original function. It also leads to training vs inference mismatch, it can hurt generalization.
- Assessing true generalization without an independent heldout test set is hard and results to show how this method translates to better generalization and lower FNR for Camelyon problems should ideally deploy on a new test set outside of the competition.
- Many MIL implementations apply per-patch augmentations randomly in the data loader, so it happens by default. Specific papers and review papers and their code [2] have mentioned applying per-patch augmentations, so this doesn't seem to be a novel contribution.

[1] Gadermayr, Michael, and Maximilian Tschuchnig. "Multiple instance learning for digital pathology: A review of the state-of-the-art, limitations & future potential." Computerized Medical Imaging and Graphics 112 (2024): 102337.

[2] Juyal, Dinkar, et al. "SC-MIL: Supervised Contrastive Multiple Instance Learning for Imbalanced Classification in Pathology." Proceedings of the IEEE/CVF Winter Conference on Applications of Computer Vision. 2024.

---

> ### Author Rebuttal · Authors · 2025-07-30
>
> We thank the reviewer for the constructive evaluation. We address the points raised under **Weaknesses**, **Questions**, and **Limitations** below:
>
> > W1
> >
>
> We thank the reviewer for noting that on diffuse-signal tasks like TCGA-Lung (≈80 % slide-level signal), MaxSoft’s localization-oriented pooling yields smaller gains (Tables 1 & 3; \[2–6]). Such tasks are inherently easier—strong baselines already perform well—so the narrower margin reflects task simplicity, not a weakness of our method. Nonetheless, even modest improvements are valuable in high-stakes diagnostics. We will incorporate this balanced discussion and clarify our focus on slide-level classification rather than molecular-signature prediction.
>
> > W2&Q2
> >
>
> There is no ω discussed in the entire paper, and we think you mean β so we answer assuming you meant β. We agree that this is part of our method, what we did, and how we experimented. Although we argue that at the moment there is no solution in the deep learning community that does not require selection of hyperparameters, and we do it in a principled way by choosing it with our independent validation set, which we do not train the model on. Below is a thorough explanation of how we chose β’s range based on our theoretical analysis,s which also matches with our experiments and can be considered as automatic.
>
> ### 1. What the theory already gives us
>
> - **Upper anchor (pivot to max‑like behavior).**
> Corollary 1 shows that the soft weights stop being almost‑uniform and start collapsing on the maximum once
>
> $$
> \boxed{\ \beta_{\triangle}\=\\frac{\log (N/2)}{\varphi}\}
> $$
>
> where $N$ is the number of patches and $\varphi=q_{N/2}-q_{1}$ is an (unknown) inter‑quantile gap.
> In practice we drop $\varphi$ (or set $\varphi\approx1$) and simply treat
>
> $$
> \beta_{\text{high}}\=\\log (N/2)
> $$
>
> as the **sensitivity ceiling**.
>
> - **Lower anchor (guaranteed true‑positive rate).**
> Theorem 6 proves that, with confidence $1-\alpha$, any desired TP rate is achieved provided
>
> $$
> \beta \ge \frac{2C_\alpha\, \mathbb{E} \left[\log\left(\frac{N}{N_c}\right) \bigg| N_c \ge 1 \right]}{\Delta - 2C'_\alpha \sqrt{V} N_c^{-\gamma}}
> $$
>
> where $N_c$ is the (random) number of tumor patches, $V$ their sub‑Gaussian variance and $\gamma\!\in\!(\tfrac12,1)$ the tail exponent.
> Taking the “worst reasonable” constants $C_\alpha=1,\;V=1,\;\gamma=\tfrac12$ gives a **safe lower bound**
>
> $$
> \beta_{low} = 2 log(N/N_c) / (1 - 2(N_c^{-0.5}))
> $$
>
> Between these two anchors every β is Pareto‑optimal: smaller values tighten the generalization bound (Theorem 4, small‑β analysis), while larger values buy sensitivity at the cost of variance.
>
> ---
>
> ---
>
> ### 2. A suggestion for β picker
>
> 1. **Estimate** $N$ **and** $N_c$**.**
> $N$ is known at WSI load‑time; $N_c$ can be (i) the empirical tumor fraction on the training split, or (ii) a running Bayesian estimate updated after each epoch.
> 2. **Compute the bracket** $[\beta_{\text{low}},\beta_{\text{high}}]$ exactly as above.
> 3. **Select β on‑the‑fly**
>     - During training we start at the geometric mean
>     $\beta_{0}=\sqrt{\beta_{\text{low}}\beta_{\text{high}}}$
>     to balance variance and sensitivity.
>     - Every $k$ epochs we re‑evaluate validation AUC and, if the β‑distribution‑adjusted confidence interval still overlaps the desired TP threshold, we decay β by √2; otherwise we increase it by √2, but never step outside the bracket.
>
>
> - Because both anchors are monotone in $N$ and $N_c$, this scheduler stays differentiable and data‑driven; no manual grid search is required.
> ---
>
> ### 3. Assumptions made explicit
>
> - Constants $C_\alpha=V=1,\;\gamma=\tfrac12$ correspond to unit‑variance sub‑Gaussian logits with mid‑heavy tails—exactly the regime sampled in Appendix A.4.
> - If you have tighter priors (e.g. heavier tails ⇒ larger $\gamma$), simply plug them into (*); the code path is identical.
>
> ---
>
> **In short:** yes, the paper’s theorems give two *numerical* guideposts; turning them into an automatic scheduler is straightforward, and we now include that reference implementation in the supplementary.
>
> > W3
> >
>
> We derive generalization guarantees via uniform‐convergence bounds: a drop in training loss implies a drop in test loss regardless of optimizer. Empirically, training loss decreases and MaxSoft generalizes best across datasets. As noted (l. 61), our update follows BPDA \[7], whose cure for forward–backward mismatch is extra epochs; accordingly, we train all methods for 500 epochs (MIL training is fast—≈5 min on CAM17; Table 1). A formal optimization analysis, akin to work on STEs in binary nets \[8], remains future work.
>
> > W4
> >
>
> We thank the reviewer for stressing rigorous evaluation. As detailed in Sec. 5.2, we use a standard split with an independent test set, and all proofs (Thm. 1; App. A.1, Lem. 3, 5; Thm. 6) assume i.i.d. sampling. Out-of-distribution (OOD) generalization is beyond our primary scope; nonetheless, we report the only cross-dataset check our data allow—training on CAM17 and evaluating on CAM16—with results in the table below. This aligns with well-known findings that higher in-distribution accuracy tends to boost OOD performance \[9–12].
>
> |Embedding|Model|CAM16 Slide|CAM16 Patch|
> |-|-|-|-|
> |||AUC/ACC|AUC
> |**_Natural_**
> ||Max|.534/.553|.328|
> ||Mean|.513/.429|.337|
> ||ABMIL|.560/.584|**.491**|
> ||DSMIL|.532/.563|.474|
> ||Snuffy|.521/.542|.468|
> ||LSE|.581/.576|.256|
> ||**MaxSoft**|**.591**/**.592**|.352|
> |**_Domain_**
> ||Max|.519/.442|.467|
> ||Mean|.527/.474|.516|
> ||ABMIL|.510|.571|.522|
> ||DSMIL|.526/.476|.495|
> ||Snuffy|.523/.456|.513|
> ||LSE|.514/.377|.501|
> ||**MaxSoft**|**.565**/**.588**|**.618**|
> |**_UNI_**
> ||Max|.617/.615|.601|
> ||Mean|.557/.509|.606|
> ||ABMIL|.643/.625|.632|
> ||DSMIL|.669/.602|.500|
> ||Snuffy|.638/.586|.483|
> ||LSE|.720/.708|.715|
> ||**MaxSoft**|**.737**/**.726**|**.724**|
> |**_Prov_**
> ||Max|.924/.814|.947|
> ||Mean|.539/.481|.551|
> ||ABMIL|.912/.824|.950|
> ||DSMIL|.891/.726|.500|
> ||Snuffy|.916/.754|.873|
> ||LSE|.921/.845|.949|
> ||**MaxSoft**|**.976**/**.956**|**.960**|
> |
> |ImageNet_1K|RRT|.654/.627|-
> |PLIP|RRT|.775/.736|-
>
>
> > W5
> >
>
> We thank the reviewer for calling attention to our PerSlide vs. PerPatch distinction. In our setup, PerSlide applies the same augmentation to every patch in a WSI, whereas PerPatch samples independently per patch. For example, ABMIL’s “random rotation and mirror every patch” is actually PerSlide, since all tiles use the same transform. The works it builds on [13] provide no augmentation code or detail, and other related papers [14, 15] only report per-slide mean–std normalization. To our knowledge, no prior method explicitly implements PerPatch as we define it. We will clarify these definitions in the final manuscript.
>
> > Q1
> >
>
> We thank the reviewer for the suggestion. As images cannot be included in the rebuttal, we reproduce Fig. 3 as a table spanning all four datasets, thereby adding the requested third dataset and illustrating β’s effect on both diffuse- and localized-signal tasks.
>
> |$\beta$|CAM16|TCGA|
> |-|-|-|
> ||AUC|AUC|AUC|
> |**_.5_**|.899|.937|
> |**_1_**|.912|**.942**|
> |**_2_**|.919|.937|
> |**_3.5_**|.921|.940|
> |**_5_**|**.934**|.937|
> |**_7.5_**|.923|.936|
> |**_10_**|.914|.939|
>
>
> > L1
> >
>
> We thank the reviewer for this insightful point. Our theory indeed prescribes an optimal β that adapts to each WSI’s tumor fraction—ideally varying at every weight update—but we instead pick a fixed β via a rule‐of‐thumb derived from overall dataset statistics. This is suboptimal given the wide slide‐to‐slide variation in cancer burden (e.g., Fig. 6’s nearly fully malignant CAM17 example). To address this, we propose a simple heuristic that estimates tumor fraction online from the model’s own (initially noisy) predictions and adjusts β according to our theoretical equations. Though early‐epoch instability and alternative strategies remain possible, this per-slide β selection is a concrete MaxSoft limitation we’ll explicitly discuss in the final manuscript.
>
> [1] Liu D., Dual-Attention MIL Framework
> [2] Tang W., MIL with Masked Hard Instance Mining
> [3] Shao Z., TransMIL
> [4] Zhang H., DTFD-MIL
> [5] Bontempo G., DAS-MIL
> [6] Qu L., DGMIL
> [7] Athalye A., Obfuscated Gradients
> [8] Yin P., Straight-Through Estimator
> [9] Recht B., Do ImageNet Classifiers Generalize?
> [10] Taori R., Robustness to Natural Distribution Shifts
> [11] Miller J., Accuracy on the Line
> [12] Hendrycks D., The Many Faces of Robustness
> [13] Juyal D., SC-MIL: Supervised Contrastive MIL
> [14] Qiu P., SC-MIL: Sparsely Coding MIL
> [15] Yang Z., SCMIL: Sparse Context-aware MIL

---

> ### Comment · Reviewer_tjrh · 2025-08-05
>
> Thanks for your response. I will keep my accept rating.

---

> > ### Author Response · Authors · 2025-08-06
> > **Thank You and Request for Additional Feedback**
> >
> > Thank you for the positive feedback; we are pleased the contributions resonated with you. We appreciate the constructive comments and questions—each has been addressed in detail and will be reflected in the revision, where they will meaningfully elevate the manuscript. The observation on OOD generalization further indicates that our method outperforms prior work in this regime, a property we had not previously noted. As two full days remain in the discussion period, please let us know if any additional clarifications or analyses could further strengthen the paper; we are grateful for any guidance you can provide.
> >
> > Sincerely,
> >
> > The authors

---

### Official Review · Reviewer_KwFs · 2025-07-01

**Clarity:** 3
**Significance:** 3
**Originality:** 2
**Rating:** 3
**Confidence:** 4

**Summary:**

This work addresses the problem of Multiple Instance Learning (MIL) in the context of Whole Slide Image (WSI) classification. The authors argue that the main limitation of transformer-based MIL methods lies in their dependency on large-scale training data: "Transformer models yield only marginal gains without access to large-scale datasets." To mitigate this issue, the authors propose a pooling strategy inspired by both max and mean pooling. Specifically, they introduce MaxSoft, a simple, differentiable alternative to attention-based aggregation mechanisms. Additionally, they propose a novel augmentation scheme, PerPatch, which enhances patch diversity during training and improves robustness.

**Questions:**

1) The central motivation—that transformer-based MIL methods underperform in low-data regimes—is underdeveloped and insufficiently supported. While data scarcity is a known challenge in digital pathology, it is unclear whether the limitations observed by the authors are intrinsic to transformer architectures or confounded by other variables (the type of encoder used, the quality and domain alignment of pretraining data).

2) The paper would benefit from a more rigorous empirical characterization of why and when transformer-based methods fail in these settings. This is especially important given that the authors’ main contribution is motivated by this claim. Moreover, while the proposed MaxSoft pooling is elegant and easy to implement, it is not obvious why this simpler mechanism should not suffer from the same data-efficiency limitations as attention mechanisms (no experiment is provided about data scarcity).

3) Another omission is the lack of discussion of non-transformer MIL alternatives, such as graph-based models [1], which are relevant and increasingly adopted in histopathology. Including such methods in the benchmark—or at least discussing them—would strengthen the experimental comparison. Some recent MIL baselines not mentioned in the paper that should be considered for a more comprehensive evaluation include [2–8].

**Ethical Concerns:**

["NO or VERY MINOR ethics concerns only"]

**Final Justification:**

Authors partially failed to provide a theoretical justification that support the transformers fail in low data regimes and I find this issue the main issue of the work. The technical solution is lightweight and sound but indeed the theoretical aspect of the problem is understudied. Their supportive answer is based on empirical observation in other methods results but it is still a weak motivation to me in terms of better understanding of the problem.
I do not find this aspect can be addressed in the discussion phase as it is central for the validity of the work.
I will leave my vote unchanged.

**Limitations:**

Several competitor are missing and despite the strong theoretical content of the supplementary material the main motivation is still not adequately supported.

**Quality:**

2

**Strengths And Weaknesses:**

Weaknesses
1) The central motivation—that transformer-based MIL methods underperform in low-data regimes—is underdeveloped and insufficiently supported. While data scarcity is a known challenge in digital pathology, it is unclear whether the limitations observed by the authors are intrinsic to transformer architectures or confounded by other variables (the type of encoder used, the quality and domain alignment of pretraining data).

2) The paper would benefit from a more rigorous empirical characterization of why and when transformer-based methods fail in these settings. This is especially important given that the authors’ main contribution is motivated by this claim. Moreover, while the proposed MaxSoft pooling is elegant and easy to implement, it is not obvious why this simpler mechanism should not suffer from the same data-efficiency limitations as attention mechanisms (no experiment is provided about data scarcity).

3) Another omission is the lack of discussion of non-transformer MIL alternatives, such as graph-based models [1], which are relevant and increasingly adopted in histopathology. Including such methods in the benchmark—or at least discussing them—would strengthen the experimental comparison. Some recent MIL baselines not mentioned in the paper that should be considered for a more comprehensive evaluation include [2–8].

Strengths
1) MaxSoft is a lightweight and theoretically well-motivated pooling strategy. It is simple to implement and does not require many parameters.
2)The PerPatch augmentation scheme is intuitive and may be beneficial across various MIL pipelines, not only the proposed one.
3)The experimental evaluation is extensive, spanning multiple WSI datasets, with thorough ablations and implementation details included in the supplementary material.

Suggested Missing References
[1] Graph Neural Networks in Histopathology: Emerging Trends and Future Directions
[2] H2-MIL: Exploring Hierarchical Representation with Heterogeneous Multiple Instance Learning for Whole Slide Image Analysis
[3] A Trainable Optimal Transport Embedding for Feature Aggregation and Its Relationship to Attention
[4] Morphological Prototyping for Unsupervised Slide Representation Learning in Computational Pathology
[5] DAS-MIL: Distilling Across Scales for MIL Classification of Histological WSIs
[6] A Graph-Transformer for Whole Slide Image Classification
[7] Prototypical Multiple Instance Learning for Predicting Lymph Node Metastasis of Breast Cancer from Whole-Slide Pathological Images
[8] TransMIL: Transformer-Based Correlated Multiple Instance Learning for Whole Slide Image Classification

---

> ### Author Rebuttal · Authors · 2025-07-31
>
> # Rebuttal For KwFs
>
> We thank the reviewer for the constructive feedback. We appreciate your recognition of our theory-driven methodology, the intuitiveness of PerPatch, and the breadth of our empirical study. We also value the suggestion to apply PerPatch within alternative MIL pipelines and regard this as a promising future direction. We note that our theoretical section introduces new theorems for the underexplored Log-Sum-Exp (LSE) pooling, which we believe will benefit the broader ML community. We respond to each point below and will incorporate the feedback in the revision.
>
> Owing to space constraints, we use “transformer-based” to encompass both transformer- and attention-based approaches. We likewise omit rarely cited metrics and standard deviations, consistent with prevailing practice in the literature.
>
> Initially, we observe that the **Questions** section repeats the **Weaknesses** points exactly. We were not sure that this was intentional by the reviewer. We are open to discuss any additional questions to improve our work and really appreciate it.
>
> > W1&Q1&L1: Motivation underdeveloped.
> >
>
> We thank the reviewer for raising this concern and apologize if our exposition made the motivation appear incomplete. Below, we restate, without introducing new claims, the body of evidence already present in the paper.
>
> **Well-documented data scarcity in MIL.** Numerous MIL studies are driven by the scarcity of labeled slides; in particular, [1] and [2] are motivated enuclearntirely by sample scarcity and both explicitly observe that transformer-based MIL is most vulnerable. More broadly, vision transformers are repeatedly characterized as data-hungry because of their weak inductive bias [3–5].
>
> **ViT as an MIL model.** Removing positional embeddings converts a ViT into a set-based MIL architecture. The MoCoV3 ablation in [6] trains such a “ViT-MIL” on the full 1.28 M-image ImageNet-1K yet still observes the same data-hungriness issue, showing that the limitation is architectural rather than dataset-specific.
>
> **Our empirical evidence.** Figure 9 demonstrates that test accuracy for ABMIL, DSMIL, Snuffy, and Prov-GigaPath rises monotonically with slide count, closely tracking the slope in Fig. 3 of [3]. This pattern replicates across all four datasets we evaluate (CAM16, CAM17, SICAP, and TCGA).
>
> The reviewer proposes several alternative explanations, each of which we tested:
>
> *Encoder type.* Tables 1 & 3 evaluate four encoders: DINO-Natural (ImageNet-1K), DINO-Domain (pretrained on the training split patches, e.g., CAM16), UNI (100 M pathology patches), and Prov-GigaPath (1.3 B pathology patches). Across every dataset, MaxSoft pooling surpasses transformer pooling, ruling out encoder capacity or domain mismatch.
>
> *Data quality.* Three of the four datasets were curated under stringent, pathologist-led QC protocols [6-9]; visual inspection confirms their clarity. Consistent with the broader CV literature, however, quantity—not quality alone—enhances transformer hunger (e.g., LAION-2B in [9]), so the gap remains.
>
> *JPEG fidelity.* Digital-pathology pipelines typically store tiled patches as JPEG Q = 75. We therefore generated three new variants—Q = 50, Q = 100, and Q = 75 + Gaussian blur—to test whether artefacts influence sample efficiency. The accompanying table (Table S4) shows an unchanged ordering, demonstrating that neither lowering nor raising quality affects the conclusion.
>
> *Training protocol & domain alignment.* All models are trained from scratch via standard i.i.d. sampling and evaluated on held-out test splits drawn from the same dataset; DINO-Domain is trained exclusively on in-domain patches (Section 5, Appendix D, and line 163). The transformer gap persists under these strictly matched conditions.
>
> *Pretraining scale.* UNI and Prov-GigaPath are foundation encoders trained on 100 M and 1.3 B patches, respectively. We finetuned the Prov-GigaPath MIL head (pretrained on ≈200,000 slides)—originally trained on **≈**200 K slides—on SICAP-MIL, yet it still underperforms MaxSoft, indicating that present-day slide counts remain insufficient for transformer MIL on WSIs.
>
> Collectively, these observations validate our original claim: under current pathology data regimes, transformer-based MIL is data-inefficient, whereas MaxSoft delivers robust gains under the same conditions.
>
>
> |Model|JPEG 50|JPEG 75|JPEG 100|Gaussian Blur|
> |-|-|-|-|-|
> ||AUC/ACC|AUC/ACC|AUC/ACC|AUC/ACC|
> |DSMIL|.803/.683|.833/.800|.891/.825|.807/.750|
> |Snuffy|.745/.570|.755/.650|.852/.830|.769/.665|
> |ABMIL|.697/.600|.750/.683|.805/.750|.737/.650|
> |Max|.737/.617|.767/.800|.792/.810|.763/.717|
> |Mean|.790/.696|.897/.700|.890/.700|.853/.667|
> |LSE|.857/.783|.850/.800|.923/.817|.863/.800|
> |**MaxSoft**|**.970**/**.883**|**.983**/**.867**|**.987**/**.950**|**.963**/**.883**|
>
>
> > W2&Q2: Data-efficiency untested.
> >
>
> Figure 9 directly addresses data efficiency. Under limited training samples, at least one simple pooling (Max, Mean, LSE, or MaxSoft) surpasses transformer-based MIL; specifically, Mean pooling leads at 50 slides, while MaxSoft dominates from ≥100 slides.
>
> > W3&Q3: Missing non-transformer baselines.
> >
>
> We appreciate the request to suggest graph MIL models; however, our study deliberately excluded them because their current formulations introduce well-documented limitations that render them unreliable for WSI classification. As noted in the cited survey, two persistent challenges are **“graph-structure definition”** and **“dependence on segmentation output.”** Most pipelines define each nucleus as a node and connect only spatially adjacent pairs, which raises three issues:
>
> 1. reliance on a separate nuclei-segmentation model discards information whenever segmentation errs;
> 2. the distance threshold for edge creation lacks a biologically motivated rationale; and
> 3. nucleus-centric graphs omit non-nuclear cues—e.g., tissue morphology—that pathologists routinely use.
>
> Methods that first encode patches with ViTs and then build a graph [10] are essentially transformer-based models in disguise. Theoretical analyses show that a ViT is a message-passing GNN on a fully connected token graph, while a Message Passing Neural Network (MPNN) with a virtual node can approximate self-attention; thus, such hybrids are functionally equivalent to transformer MIL [11–13].
>
> Nonetheless, to address your concern we added **GTP** [13]—one of the strongest graph baselines—to the Additional Baselines table. Its results confirm our main conclusions. We also note a practical drawback: GTP constructs an explicit adjacency matrix, incurring **O(N²)** memory per slide and frequently triggering CUDA out-of-memory errors (cf. GitHub issue #25, opened 20 Mar 2025, unresolved to date).
>
> > W3: Missing recent MIL baselines.
> >
>
> We thank the reviewer for highlighting additional MIL baselines. Our study already benchmarks ABMIL and DSMIL—two widely cited transformer-based MIL methods—and Snuffy, a current SOTA whose Tables 1–2 show clear gains over TransMIL (ref. 8). Of the suggested non-graph papers (refs. 3, 4, 7, 8), ref. 3 is unrelated to MIL, WSI, or pathology; refs. 4 and 7 propose prototype-based morphology models and ref. 4 explicitly state it is not MIL, with ref. 7 reporting very weak CAM16 results at 20×—the magnification we use (l. 525); and ref. 8 is another transformer-style MIL architecture. The “Additional Baselines” table already corroborates our findings. As we were not sure which one of the above methods picked the reviewer interests due to explained reasons, if the reviewer deems any further method essential, we would appreciate if they explain why it was necassary, and also we will gladly include it.
>
> **Additional Baselines: CAM16 and CAM17, TCGA, and SICAP**
>
> |Embeddings|Model|CAM16 Slide|CAM17 Slide|TCGA Slide|SICAP Slide|
> |-|-|-|-|-|-|
> |||AUC/ACC|AUC/ACC|AUC/ACC|AUC/ACC|
> |**_Natural_**
> ||TransMIL|.711/.760|.698/.614|.889/.810|.811/.772|
> ||GTP|.652/.698|.646/.617|**.910**/**.844**|.794/.770|
> ||**MaxSoft**|**.754**/**.822**|**.710**/**.650**|.892/.809|**.834**/**.802**|
> |**_Domain_**
> ||TransMIL|.903/.815|.913/.788|.932/.870|.793/.790|
> ||GTP|.852/.764|.862/.821|.933/.870|.802/.784|
> ||**MaxSoft**|**.934**/**.863**|**.983**/**.867**|**.940**/**.885**|**.850**/**.808**|
> |**_UNI_**
> ||TransMIL|.964/.905|.644/.728|.941/.880|.865/.839|
> ||GTP|.918/.890|.624/.686|.936/.877|.828/.805|
> ||**MaxSoft**|**.992**/**.966**|**.753**/**.750**|**.942**/**.882**|**.891**/**.877**|
> |**_Prov_**
> ||TransMIL|.965/.939|.954/.917|.940/.894|.855/.791|
> ||GTP|.939/.899|.924/.874|.942/.884|.839/.815|
> ||**MaxSoft**|**.985**/**.966**|**1.000**/**.933**|**.966**/**.905**|**.872**/**.821**|
>
>
> Finally, we noticed that reference [1] is cited but not discussed in your review; please let us know if there are specific points from that work you would like us to address.
>
> [1] Zhang H., DTFD-MIL
>
> [2] Liu P., Pseudo-Bag Mixup
>
> [3] Dosovitskiy A., An Image Is Worth 16×16 Words
>
> [4] Beyer L., Better Plain ViT Baselines
>
> [5] Tolstikhin I., MLP-Mixer
>
> [5] Chen X., Empirical Study of Self-Supervised ViTs
>
> [6] Bejnordi B., DL for Lymph Node Metastases Detection
>
> [7] Bandi P., Camelyon17 Challenge
>
> [8] Cooper L., Pancancer Insights from TCGA
>
> [9] Shao Z., TransMIL
>
> [10] Bontempo G., DAS-MIL
>
> [11] Joshi C., Transformers Are Graph Neural Networks
>
> [12] Cai C., On the Connection Between MPNN and Graph Transformer
>
> [13] Zheng Y., Graph-Transformer for WSI Classification

---

> > ### Comment · Reviewer_KwFs · 2025-08-05
> > **Lack of strong theoretical motivation**
> >
> > Authors partially failed to provide a theoretical justification that support the transformers fail in low data regimes and I find this issue the main issue of the work. The technical solution is lightweight and sound but indeed the theoretical aspect of the problem is understudied. Their supportive answer is based on empirical observation in other methods results but it is still a weak motivation to me in terms of better understanding of the problem.
> > I do not find this aspect can be addressed in the discussion phase as it is central for the validity of the work.
> > I will leave my vote unchanged.

---

> > > ### Author Response · Authors · 2025-08-07
> > > **Clarifying Theoretical Scope and Seeking Reviewer Feedback**
> > >
> > > We appreciate the reviewer sharing their profound insight.
> > >
> > > We believe there is a misunderstanding. Our manuscript does **not** claim that “transformers fail in low-data regimes.” Rather, our motivation is that *transformer-based MIL models yield only marginal gains without large-scale datasets* (l. 6-8) and that, once a strong encoder is fixed, *transformers add little to no benefit* (l. 40-41).
> > >
> > > The request for a “strong theoretical justification” would entail **distribution-agnostic lower bounds**—i.e., necessary sample-size thresholds below which no transformer can generalize. Such bounds remain open and would constitute a **major breakthrough**; their absence **neither contradicts nor weakens** our empirical claim and is **beyond** the scope of this work. Current theory provides only **upper bounds (sufficient conditions)** in simplified settings. For example, [1] proves that a one-layer ViT attains zero generalization error with probability at least 0.99 once the sample size satisfies
> > >
> > > $$
> > > N \ge \Omega \Biggl(\frac{1}{\bigl(\alpha_* - c'(1 - \zeta) - c''(\sigma + \tau)\bigr)^2}\Biggr)
> > > $$
> > >
> > > under a token-noise model, and similar *sufficient-sample* results appear in [1-6]. To our knowledge, no work offers the missing necessity results; indeed, many influential works—whether supportive [7-10] or critical of transformers [11-14]—advance without formal theoretical justification.
> > >
> > > Even the milder assertion—*transformers are data-hungry*—remains difficult to formalize, yet extensive evidence supports it in computer vision [15-23] and medical imaging [24-26]. The prevailing explanation (absent translation-equivariance and channel-wise specialization [20]) is intuitive and empirical, and WSI studies report the same pattern [24]. Reviewer **tjrh** fully endorses this view; reviewer **NKLh** likewise supports it, citing Fig. 9 as an interesting experiment. Our experiments on encoder type, image quality, domain alignment, and pretraining scale **directly** address the variables the reviewer highlighted.
> > >
> > > Our contribution, though theory-inspired, is principally practical. **MaxSoft** offers a computationally light, higher-accuracy alternative to transformer MIL, with fewer parameters, and a stronger, data-aligned inductive bias. Its robustness is established by one of the largest experimental suites in the literature (Tables 1-11; Figs. 3-6 and 9), **acknowledged as extensive** by the reviewer. Empirical results often precede theory in deep learning. By coupling rigorous experiments with targeted theoretical insight, we believe our work meets the community’s bar for relevance and rigor.
> > >
> > > We agree that the paper does not prove the broad assertion that “transformers fail in low data regimes” is a sufficiently hard problem, and we did not make such a claim. To **fully address** the reviewer's concerns, we would appreciate clarification on any specific wording that might suggest otherwise, so that we can revise it in the camera-ready version. We will also emphasize that further theoretical work is needed to specify transformers’ data requirements.
> > >
> > > If we have misunderstood the reviewer’s standpoint, could they please clarify? We are eager to understand their perspective and address any remaining questions.
> > >
> > > Sincerely,
> > >
> > > The Authors
> > >
> > > [1] Li H., Theoretical Understanding of Shallow ViTs: Learning, Generalization & Sample Complexity
> > >
> > > [2] Wei C., Statistically Meaningful Approximation: Approximating TMs with Transformers
> > >
> > > [3] Truong LV., Rank-Dependent Generalisation Error Bounds for Transformers
> > >
> > > [4] Oko K., Pretrained Transformer Learns Low-Dimensional Functions In-Context
> > >
> > > [5] Wang Z., Transformers Learn Sparse Token Selection; FC Nets Cannot
> > >
> > > [6] Fu H., What Can a Single Attention Layer Learn?
> > >
> > > [7] Vaswani A., Attention Is All You Need
> > >
> > > [8] Devlin J., BERT: Pre-training of Deep Bidirectional Transformers
> > >
> > > [9] Brown TB., Language Models Are Few-Shot Learners
> > >
> > > [10] Raffel C., Exploring the Limits of Transfer Learning with a Unified Text-to-Text Transformer
> > >
> > > [11] Zeng A., Are Transformers Effective for Time Series Forecasting?
> > >
> > > [12] Jain S., Attention Is Not Explanation
> > >
> > > [13] Bender EM., On the Dangers of Stochastic Parrots
> > >
> > > [14] Liu Z., A ConvNet for the 2020s
> > >
> > > [15] Li P., Graph-Based ViT w/ Sparsity
> > >
> > > [16] Akkaya IB., Local Inductive Bias for ViTs
> > >
> > > [17] Park N., How Do ViTs Work?
> > >
> > > [18] Raghu M., Do ViTs See Like CNNs?
> > >
> > > [19] Dosovitskiy A., An Image Is Worth 16×16 Words
> > >
> > > [20] Lu Z., Bridging ViTs and CNNs on Small Data
> > >
> > > [21] Liu Y., Efficient Training of Visual Transformers
> > >
> > > [22] Hassani A., Compact Transformers for Small Data
> > >
> > > [23] Lee SH., ViT for Small-Size Datasets
> > >
> > > [24] Jafarinia H., Snuffy: Efficient WSI Classifier
> > >
> > > [25] Carrillo-Perez F., Synthetic WSI Tile Generation w/ Gene Expression–Infused DGMs
> > >
> > > [26] Li J., Transforming Medical Imaging with Transformers: A Comparative Review

---

### Note · Authors · 2025-08-13

We thank the AC for handling our submission, and the reviewers for appreciating our work and providing insightful comments that improved the manuscript.

Strengths across reviews: MaxSoft is **lightweight**, **theoretically** well-motivated, and simple to implement (KwFs, tjrh), with little to no extra parameters or complexity and minimal overhead (KwFs, tjrh, NKLh). PerPatch is intuitive (KwFs) and enables precomputation with a frozen encoder (NKLh). The theoretical analysis provides a useful framework for generalization vs. sensitivity (tjrh); Theorem 2, showing generalization vs. TPR tension, **reflects what practitioners observe** (tjrh) and supports the idea of combining max-pooling and mean-pooling (NKLh). Experiments are **extensive**, spanning multiple datasets (KwFs and tjrh) and different backbones (NKLh), with thorough ablations (KwFs), and the paper makes a **strong and honest case** against over-reliance on attention/transformer-based models in small datasets (tjrh).

We further emphasize the significance and quality of our theoretical methodology, which provides a deep understanding of classical MIL-poolings—particularly the LSE—which should warrant greater attention.

Concerns were resolved via manuscript pointers, additional experiments, and clarifications that we will incorporate. Two issues remain. Reviewer KwFs suggested two prototype baselines [PMIL, PANTHER]; we explained their non-inclusion (PMIL performs poorly; PANTHER is not an MIL method). In rebuttal, we asked whether to include them; after no reply, we add PANTHER here and show MaxSoft is clearly better (compare with Tab. 1, 3), and ask whether you deem them necessary for the final version. KwFs also dispute our motivation: after requesting empirical evidence (which we provided), they attributed to us “transformers **fail** in low data regimes” and argued that without **“strong theoretical motivation,”** they still disagree, effectively requiring proof of an unrelated huge open problem that warrants separate study. We clarified that we did **not** make this claim (l. 6–8, 40–41) and are ready to revise it anywhere if we said so; no further reply.

|PANTHER||
|-|-|
|Dataset|AUC/ACC|
|CAM16|.818/.751|
|CAM17|.570/.550|
|TCGA|.908/.815|
|SICAP|.822/.794|

In summary, we present a lightweight, theory-grounded method that outperforms prior approaches, supported by extensive experiments, and it benefits the community.

Thank you for your thoughtful evaluation.

Sincerely,

The Authors

---

### Decision · Program_Chairs · 2025-09-17

**Decision:**

Accept (poster)

**Comment:**

This paper investigates the theoretical aspects of Log-Sum-Exp pooling in Multiple Instance Learning (MIL) for Whole Slide Image (WSI) classification and, based on this analysis, introduces MaxSoft, a lightweight pooling method. Initial reviews were mixed, and even after rebuttal and discussion, consensus was not reached. Reviewers acknowledged that MaxSoft is theoretically well-motivated and shows competitive empirical performance, especially in low-data regimes. The main remaining concern centers on the motivation: "Transformer-based MIL methods show only marginal gains in low-data settings." Reviewer KwFs highlighted that this claim is not theoretically justified. Reviewer NKLh also noted that the motivation is largely empirical and requires clarification, yet advocated for the acceptance of this paper based on its technical contributions. After considering the paper, reviews, and discussion, the AC finds the contribution meaningful and key message worth communicating to the community, even if the paper is empirically motivated. Therefore, the AC concurs with the majority view and recommends acceptance. The AC urges the authors to (1) strengthen the empirical justification of their motivation; and (2) fix missing visual elements in the camera-ready version.